# Can LLMs Reason Structurally? Benchmarking via the lens of Data Structures

Yu He [* 1]   Yingxi Li [* 1]   Colin White [2]   Ellen Vitercik [1]

## Abstract

Large language models (LLMs) are deployed on increasingly complex tasks that require multi-step decision-making. Understanding their algorithmic reasoning abilities is therefore crucial. However, we lack a diagnostic benchmark for evaluating these capabilities. We propose to use data structures as a principled lens: as fundamental building blocks of algorithms, they naturally probe *structural reasoning*—the ability to understand and manipulate relationships such as order, hierarchy, and connectivity that underpin algorithmic reasoning. We introduce `DSR-Bench` (Data Structure Reasoning Benchmark), spanning 20 data structures, 35 operations, and 4,140 problem instances. `DSR-Bench` features hierarchical task organization, fully automated generation and evaluation, and fine-grained diagnostics. Evaluating 13 state-of-the-art LLMs reveals critical limitations: the top-performing model achieves only 0.46/1 on challenging instances. Three auxiliary probes targeting more realistic usages expose further weaknesses: models perform poorly on spatial data and context-rich scenarios, and they struggle to reason over their own code.

## 1. Introduction

As large language models (LLMs) tackle increasingly complex real-world tasks, it is essential to understand their *algorithmic reasoning* abilities (Eberle et al., 2025)—that is, their ability to understand, learn, and execute algorithmic operations. By studying the algorithmic primitives that LLMs learn and employ, we can analyze their reasoning processes and how they compose these primitives in multi-step problem-solving. Such insights may also inspire algorithm-centric model design as an alternative to the scaling paradigm (Bounsi et al., 2024; Eberle et al., 2025).

*Equal contribution. [1]Stanford University, Stanford, CA, USA [2]Abacus.AI, San Francisco, CA, USA. Correspondence to: Yu He <heyu@stanford.edu>, Yingxi Li <yingxi@stanford.edu>.

*Proceedings of the 43$^{rd}$ International Conference on Machine Learning*, Seoul, South Korea. PMLR 306, 2026. Copyright 2026 by the author(s).

In our analysis of algorithmic reasoning, we explicitly focus on LLMs' *inherent* reasoning abilities, isolated from external tools. This emphasis is motivated by recent initiatives like Gemini-Deep-Think's and OpenAI's participation in IMO competitions (Luong & Lockhart, 2025; Wei, 2025), which strictly prohibit the use of code and proof assistance, emphasizing end-to-end reasoning as a step toward artificial general intelligence.

**Limitations of existing *reasoning* and *coding* benchmarks.** Most reasoning benchmarks focus on high-level, domain-specific tasks such as mathematics (Liu et al., 2024c;a), data analysis (White et al., 2025), or STEM questions (Hendrycks et al., 2021). These tasks often require complex responses with intertwined reasoning skills, making it difficult to isolate algorithmic primitives. Coding benchmarks such as LiveCodeBench (Jain et al., 2025), SWEBench (Jimenez et al., 2024), and Aider (Aider-AI, 2025) assess code generation, which risks contamination due to the abundance of online resources. Furthermore, coding offloads essential reasoning processes to external interpreters, obscuring the inherent reasoning abilities we aim to study. We therefore need a fundamental approach to isolate algorithmic reasoning from domain-specific complexities and evaluate it without using external tools.

**Limitations of existing *algorithmic* benchmarks.** Despite growing research on algorithmic reasoning, existing benchmarks fall short in enabling systematic, granular assessment. CLRS-Text (Markeeva et al., 2024) evaluates the simulation of 30 classical algorithms (Cormen et al., 2009), but empirical differences across algorithms (e.g., success on bubble sort versus failure on insertion sort) provide limited insights into why these differences arise or which underlying operations are driving them. Additionally, its prompts are direct textual translations of the CLRS-30 Benchmark (Veličković et al., 2022), originally designed for non-textual models such as graph neural networks: they use code-like syntax and provide no natural language task description beyond the algorithm name. Existing algorithmic benchmarks with natural language prompts are primarily limited to graphs (Wang et al., 2023a; Fatemi et al., 2024). Therefore, we need a comprehensive natural language benchmark over diverse structures that enables fine-grained diagnosis, beyond end-to-end algorithmic simulation.

**Key challenge.** The central challenge of curating a diagnostic benchmark that addresses these limitations is defining algorithmic primitives (Eberle et al., 2025) that enable decomposition and failure localization. We need a framework that (i) provides a principled dictionary of fundamentals, (ii) supports the composition of basic skills for complex reasoning, and (iii) enables automated, deterministic, and unambiguous evaluation. This is difficult because these requirements are in tension: tasks should be *atomic* to prevent heuristic or shortcut solutions and enable precise error attribution to satisfy (i), yet *expressive* enough to cover a wide range of complex tasks for (ii). Moreover, satisfying (iii) is challenging because deterministic, unambiguous evaluation is hard in natural language, where errors can be confounded by unspecified tie-breaking or prompt ambiguity.

**Why data structures?** We propose data structure tasks as a well-suited testbed for algorithmic reasoning. Data structures are fundamental to algorithms, interpretable in their operations, and deterministic in their outputs. They also abstract the key relationships underlying algorithmic reasoning: arrays represent sequences, trees encode hierarchies, and graphs capture complex networks. This diversity enables us to holistically assess whether models can correctly construct, manipulate, and compose different relationships—the core capability we term *structural reasoning*, which we use as a operational, behavioral framework for probing algorithmic reasoning.

More importantly, structural reasoning is a fundamental requirement for many real-world applications. For example, trip planning requires interpreting maps as graphs and managing scheduling priorities with queues, while supply chain optimization depends on hierarchical resource allocation and temporal sequencing. Understanding how well LLMs reason structurally is thus essential for their safe deployment in critical domains such as robotics (Sado et al., 2023) and healthcare (Sadeghi et al., 2024).

**Our contribution.** We introduce `DSR-Bench` (Data Structure Reasoning Benchmark), the first benchmark for systematically evaluating structural reasoning in LLMs through data structure tasks. It comprises 4,140 problem instances spanning 20 data structures, 35 operations, three length levels (*short*, *medium*, and *long*), and five evaluation components (`main`, `challenge`, `spatial`, `realistic`, and `code`). Data structures are grouped into six categories (see Figure 1), and each task enables interpretable, automatically verifiable assessments of whether models can reason over a specific relationship. The highlights and main takeaways of `DSR-Bench` are:

- **Hierarchical task organization.** Tasks are organized in increasing difficulty for structural reasoning, where simpler tasks (e.g., queues) serve as prerequisites for complex

ones (e.g., breadth-first traversal on trees). This design enables fine-grained localization of failure modes.

- **Deterministic, automated evaluation.** All data is generated synthetically, enabling efficiency while reducing test-set contamination. Evaluation is fully automated with unambiguous ground truth, avoiding human or LLM-based judging for fair assessment.

- **Interpretable, diagnostic insights.** Data structures have well-defined semantics and easily computable ground truth. This allows for direct comparisons of model reasoning traces and interpretable analyses of how models reason and where they fail, mitigating leaderboard-driven reward hacking (Amodei et al., 2016).

- **Five evaluation components covering diverse reasoning settings, offering practical takeaways.**
  - `main`: Canonical data structure tasks. Empirical analysis reveals several gaps, e.g., instruction-tuned models struggle with multi-attribute and multi-hop reasoning, while reasoning models fail to adapt to user-defined constraints. Simpler prompts consistently help, though CoT requires careful design.
  - `challenge`: Complex structures (e.g., hybrid and compositional) with longer inputs. The best-performing model scores only 0.46 out of 1, revealing limitations in frontier models.
  - `spatial`: High-dimensional data structural tasks common in real-world applications. Results show that models degrade as dimensionality increases and struggle with non-uniform distributions.
  - `realistic`: Data structures embedded in realistic scenarios (e.g., clinic appointments). We find that models struggle to extract structure and navigate language ambiguity, exposing a significant gap between formal reasoning and real-world deployment.
  - `code`: A *supplementary* probe evaluating structural reasoning with and without code generation. Though our focus is on LLMs' inherent reasoning, we include this probe to assess whether code provides additional benefit. We observe that models rarely benefit from reasoning over self-generated code. External interpreters help with familiar tasks but fail on non-standard or realistic variants.

- **Comprehensive empirical analysis.** We evaluate 13 state-of-the-art LLMs spanning open- and closed-source, instruction-tuned and reasoning models. Our ablations examine prompting strategies, distribution shifts, and qualitative error patterns (such as testing implicit priors and instruction-following failures). Detailed empirical analysis is presented in Section 4. These findings offer actionable insights for designing architectures and training strategies to improve algorithmic reasoning in LLMs.

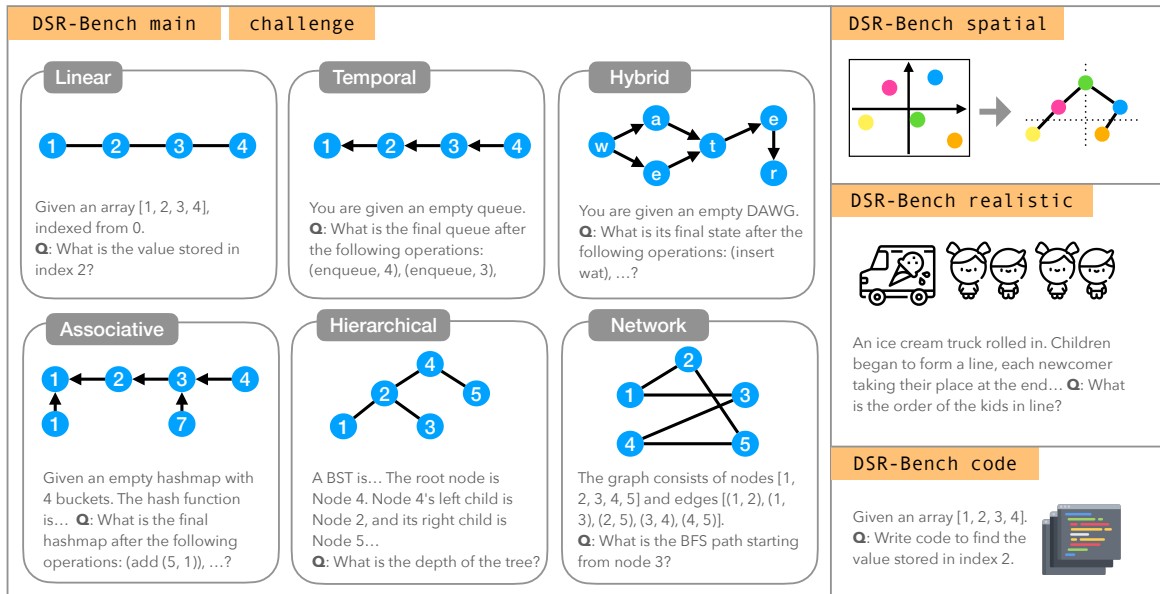

**Figure 1.** High-level overview of `DSR-Bench-main` with six data structure categories capturing distinct relationships, plus the `challenge` subset. Three specialized probes that holistically evaluate structural reasoning under different settings: `spatial` (multi-dimensional data), `realistic` (realistic context-rich scenarios), and `code` (code generation). See Section B for full list of tasks.

- **Open-source.** We release all code at `https://github.com/dransyhe/DSR-Bench` and datasets at `https://huggingface.co/collections/vitercik-lab/dsr-bench` for community engagement.

## 2. Additional related work

We review the limitations of related benchmarks in Section 1, with an extended discussion for LLM reasoning and coding benchmarks in Section A.

**Table 1.** Comparison with prior algorithmic benchmarks.

| Benchmark | Coverage | Hierarchical | Atomic | Expressive | Natural language | Auxiliary probes |
|---|---|---|---|---|---|---|
| CLRS-Text | 30 algorithms | ✗ | ✗ | ✓ | ✗ | ✗ |
| NLGraph | Graphs only | ✗ | ✗ | ✓ | ✓ | ✗ |
| GraphQA | Graphs only | ✗ | ✓ | ✗ | ✓ | ✗ |
| DSR-Bench | 20 DS, 6 categories | ✓ | ✓ | ✓ | ✓ | ✓ |

In Table 1, we summarize how `DSR-Bench` differs from prior algorithmic benchmarks, where "atomic" and "expressive" correspond to requirements (i) and (ii) discussed in Section 1's Key Challenge. We further elaborate on *task*-level distinctions. NLGraph (Wang et al., 2023a) and GraphQA (Fatemi et al., 2024) focus on graph tasks (e.g., connectivity, cycle detection). While `DSR-Bench` also has graph tasks (breadth-first search (BFS), depth-first search (DFS)), these tasks do not appear in those benchmarks and are integral parts of our hierarchical design, serving as preliminaries for harder tasks (e.g., directed acyclic word

graph). CLRS-Text (Markeeva et al., 2024) evaluates end-to-end simulation of 30 classical algorithms from Cormen et al. (2009) (e.g., sorting, greedy algorithms), with only BFS and DFS overlapping with ours. In contrast, `DSR-Bench` targets data structures as core building blocks underlying these algorithms, enabling finer-grained diagnosis and more interpretable, actionable insights.

## 3. `DSR-Bench`: the Data Structure Reasoning Benchmark

We detail the design of `DSR-Bench`. In Section 3.1, we describe the task composition, which spans six relationship categories. We then present the task organization and five evaluation components (Section 3.2), prompt design (Section 3.3), and data generation and evaluation (Section 3.4).

### 3.1. Tasks

We propose six relationship categories fundamental to algorithm reasoning: Linear, Temporal, Associative, Hierarchical, Network, and Hybrid. We design data structure tasks to cover each category; see Section B for their details.

- **Linear (Sequential):** This category includes ARRAY and its operations (access, insert, delete, reverse, search). It captures basic ordering of elements, which forms the basis of more complex structures.
- **Temporal (Time-based ordering):** Temporal includes STACK, QUEUE, LRU CACHE, and PRIORITY QUEUE,

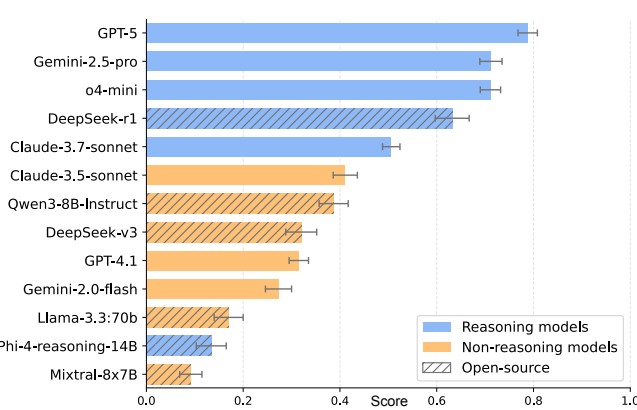 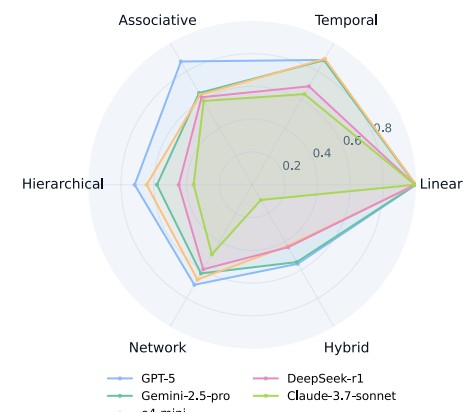

*Figure 2.* Left: Scores of thirteen models on `DSR-Bench-main`, averaged across three runs. Right: Radar chart showing scores of top-performing models across six relationship categories. Note that the best model on `DSR-Bench-challenge` scores only 0.46.

imposing additional constraints on top of linear structures. Our tasks include compound sequences of insertions and deletions. Temporal structure is often used in time-sensitive applications, including scheduling.

- **Associative (Key-value mapping):** This category includes HASHMAP, TRIE, SUFFIX TREE, and SKIP LIST. We evaluate the ability to reason about associative relationships through construction and compound operations. Associative structures are essential for efficient lookup and access in systems such as databases.

- **Hierarchical (Tree-like):** This group includes BINARY SEARCH TREE (BST), HEAP, RED-BLACK (RB) TREE, and B+ TREE, testing traversal and compound operations. These structures require data maintenance over multiple level sets and are common in file systems and databases.

- **Network (Connectivity and group membership):** This category includes GRAPH and DISJOINT SET UNION (DSU) tasks. Graph tasks include traversals, testing reasoning over many-to-many relationships, as in social networks. DSU tasks test union–find operations used in connectivity and clustering algorithms.

- **Hybrid (Combined relationships):** Real-world systems often require a combination of different structural reasoning skills. This category includes BLOOM FILTER (probabilistic set membership) and DIRECTED ACYCLIC WORD GRAPH (DAWG) (trie-like hierarchical graph). These tasks test whether models can compose and generalize beyond individual structures.

### 3.2. Benchmark design

**Hierarchical task organization** A key strength of `DSR-Bench` is that it organizes tasks by increasing complexity, where simpler tasks are prerequisites for more complex ones. For example, performing a DFS (Depth-First Search) on a graph requires STACK operations to maintain

the frontier, enabling fine-grained diagnoses that isolate where reasoning begins to fail.

**Operation types** For each data structure, `DSR-Bench` provides a diverse set of operation tasks (see the full list in Section B), spanning three categories: construction, inspection (e.g., access, traversal), and manipulation (e.g., insert, delete). Beyond these atomic operations, it also includes compound operations, which are sequences of operations (e.g., [insert, insert, delete, ...]) designed to test whether models can perform multi-step structural reasoning.

**Length levels** Tasks are assigned to three length levels based on input length: *short* (5–10), *medium* (11–20), and *long* (21–30), to assess length generalization. For atomic operations, length refers to the number of input elements (e.g., tree nodes). For compound operations, length is defined by the number of sequential operations.

**Evaluation components** Beyond the `main` suite, we curate a `challenge` subset with particularly complex structures to stress-test advanced reasoning. We further introduce three targeted probes to evaluate structural reasoning in scenarios applicable to real-world deployment: `spatial`, evaluating on high-dimensional data; `realistic`, embedding data structure tasks in realistic scenarios with context-rich language; and `code`, assessing whether models can leverage code generation to aid structural reasoning.

### 3.3. Prompt design

For each task, we design a prompt template and populate it with synthetic data to generate problem instances. Each prompt has the following format: (i) a concise description of the data structure; (ii) a detailed explanation of the operations to be performed, written to avoid ambiguity; (iii) the initial state of the data structure (e.g., an existing tree or

---

Example prompt for QUEUE compound.

(i) A queue maintains a first-in, first-out (FIFO) order, where items are added at one end and removed from the other.

(ii) There are two operations: `(enqueue, k)` adds $k$ to the back; `(dequeue)` removes the front.

(iii) You are given an empty queue initially.

(iv) **Q:** What is the final queue after: `(enqueue, 49)`, `(enqueue, 85)`, `(dequeue)`,...

---

list) and any additional inputs required to execute the task (e.g., new elements to insert or delete); and (iv) a specific question requesting the final outcome. Explicit task descriptions are included in the prompt to ensure fair evaluation and reduce bias from differences in models' prior knowledge, minimizing the role of knowledge retrieval. Following recommended practices in prompt engineering, we also append the instruction *"Answer the question in <number> tokens"* to encourage concise outputs within a specified token budget. We also implement five different prompting strategies, as we detail in the next section.

### 3.4. Data generation and evaluation

**Data generation** As a key strength of `DSR-Bench`, all data is synthetically generated to ensure scalability and minimal risk of data contamination (Zhang et al., 2024a; Xu et al., 2024a). Numerical inputs are sampled uniformly from the range of 0 to 100, and string inputs are composed of uniformly sampled lowercase English letters. Each data structure and its operations are programmatically implemented to produce ground-truth outputs, enabling accurate labeling and easy integration of new data structures.

**Automated evaluation** The automated evaluation is fully deterministic and reproducible, constituting another notable advantage of `DSR-Bench`. This feature avoids subjective human or LLM-based judging, as commonly required in proof-writing or math benchmarks (Chiang et al., 2024; Feuer et al., 2025; Ye et al., 2025). We use Structured Output (either built-in for most models or supported via Ollama/Instructor) to enforce outputs that conform to a predefined JSON Schema. For example, a BST pre-order traversal must return a `list[int]`. Across 1,620 trials with nine models and six structures, we observe zero schema violations (Table 34 in the Appendix), confirming its robustness.

**Scoring system** We use binary (0/1) scoring on the final answer, comparing against a single well-defined solution. We do not quantitatively score intermediate steps, allowing models to reason freely and adopt different strategies. Any potential ambiguity (hash functions, tie-breaking rules) is explicitly specified in the prompts. As an alternative metric, we include an ablation using the Levenshtein distance

as a partial-credit metric in Section I.1, which shows that binary scoring provides a clearer distinction and a fairer comparison. We further supplement with an ablation using edit distances for tree- and graph-based data structures in Section I.2, which reveals additional information, but such metrics are only applicable to a small subset of tasks.

## 4. Empirical analysis

**Models** We evaluate 13 state-of-the-art LLMs, including instruction-tuned and reasoning models, as well as open- and closed-source models. We select the flagship models from each major provider, with each problem evaluated three times (161,460 total evaluations). **Instruction-tuned models** are widely deployed due to their efficiency and scalability, making their structural reasoning capabilities practically important: we cover GPT-4.1-2025-04-14 (OpenAI, 2025a), Gemini-2.0-Flash-001 (Google, 2025), DeepSeek-V3 (DeepSeek et al., 2025a), Qwen3-8B (Yang et al., 2025), Claude-3-5-Sonnet-20241022 (Anthropic, 2024), Llama3.3-70B (Grattafiori et al., 2024), and Mixtral-8x7B-Instruct-v0.1 (Jiang et al., 2024). **Reasoning models** are explicitly trained for complex, multi-step reasoning: we evaluate GPT-5-2025-08-07 with medium thinking effort (OpenAI, 2025), o4-mini-2025-04-16 (OpenAI, 2025b), Gemini-2.5-Pro (stable) (Comanici et al., 2025), Claude-3-7-Sonnet-20250219 (Anthropic, 2025), DeepSeek-R1 (DeepSeek et al., 2025b), Phi-4-reasoning-14B (Abdin et al., 2025).

**Prompting strategies** We also study the impact of prompting strategies on data structure tasks. Unlike reasoning models with internal multi-step inference (e.g., reasoning tokens), instruction-tuned models are particularly sensitive to prompt formulation. We evaluate five prompting strategies: (i) **Stepwise**, which adds a "steps" field to the output JSON schema; (ii) **0-CoT**, which appends *"Let's think step by step"* without examples; (iii) **CoT**, which provides a single example with intermediate reasoning steps; (iv) **3-shot**, which includes three input-output examples; and (v) **None**, the default prompting setting with no added text. See Section C for examples.

### 4.1. Can LLMs understand and manipulate data structures?

We present empirical results on `DSR-Bench-main` and its challenging subset `challenge`. We discuss insights from seven instruction-tuned models in Section 4.1.1 and from six reasoning models in Section 4.1.2.

#### 4.1.1. INSTRUCTION-TUNED MODELS

**Instruction-tuned models struggle with multi-attribute reasoning.** As shown in Table 2, these models show sharp degradation in tasks involving elements with multiple at-

*Table 2.* Scores on `DSR-Bench-main` across 13 models. The table aggregates results by data structure and relationship type across three runs, including category scores and an overall score.

| Relationship | Data Structure | GPT-5 (med) | Gemini-2.5-Pro | o4-mini | DeepSeek-R1 | Claude-3.7-Sonnet | Claude-3.5-Sonnet | Qwen3-8B | DeepSeek-V3 | GPT-4.1 | Gemini-2.0-Flash | Llama 3.3-70B | Phi-4-R-14B | Mixtral-8x7B |
|---|---|---|---|---|---|---|---|---|---|---|---|---|---|---|
| Linear | Array | 1.00 | 1.00 | 1.00 | 0.99 | 0.99 | 0.96 | 0.99 | 0.98 | 0.94 | 0.90 | 0.69 | 0.63 | 0.49 |
| | *Category avg.* | *1.00* | *1.00* | *1.00* | *0.99* | *0.99* | *0.96* | *0.99* | *0.98* | *0.94* | *0.90* | *0.69* | *0.63* | *0.49* |
| Temporal | Stack | 1.00 | 1.00 | 1.00 | 0.99 | 0.97 | 0.99 | 0.99 | 0.41 | 0.55 | 0.36 | 0.04 | 0.12 | 0.09 |
| | Queue | 1.00 | 1.00 | 1.00 | 0.98 | 1.00 | 0.99 | 0.97 | 0.43 | 0.55 | 0.36 | 0.25 | 0.05 | 0.03 |
| | LRU | 1.00 | 1.00 | 1.00 | 0.16 | 0.30 | 0.82 | 0.49 | 0.01 | 0.85 | 0.78 | 0.50 | 0.00 | 0.00 |
| | Priority Queue | 0.52 | 0.51 | 0.55 | 0.65 | 0.28 | 0.33 | 0.35 | 0.20 | 0.25 | 0.16 | 0.08 | 0.01 | 0.01 |
| | *Category avg.* | *0.88* | *0.84* | *0.89* | *0.69* | *0.64* | *0.79* | *0.70* | *0.26* | *0.55* | *0.42* | *0.22* | *0.04* | *0.03* |
| Associative | Hashmap | 0.87 | 0.28 | 0.51 | 0.33 | 0.63 | 0.16 | 0.01 | 0.00 | 0.06 | 0.10 | 0.00 | 0.00 | 0.02 |
| | Trie | 0.94 | 0.62 | 0.68 | 0.49 | 0.08 | 0.49 | 0.00 | 0.02 | 0.18 | 0.17 | 0.00 | 0.00 | 0.00 |
| | Suffix Tree | 0.98 | 0.90 | 0.73 | 0.96 | 0.91 | 0.08 | 0.36 | 0.67 | 0.00 | 0.01 | 0.00 | 0.11 | 0.00 |
| | Skip List | 0.68 | 0.78 | 0.62 | 0.69 | 0.75 | 0.42 | 0.03 | 0.02 | 0.07 | 0.06 | 0.01 | 0.01 | 0.00 |
| | *Category avg.* | *0.87* | *0.65* | *0.63* | *0.62* | *0.59* | *0.29* | *0.10* | *0.18* | *0.08* | *0.09* | *0.00* | *0.03* | *0.00* |
| Hierarchical | BST | 1.00 | 0.97 | 0.86 | 0.73 | 0.64 | 0.71 | 0.52 | 0.58 | 0.59 | 0.43 | 0.34 | 0.18 | 0.09 |
| | Heap | 0.61 | 0.38 | 0.68 | 0.48 | 0.40 | 0.27 | 0.29 | 0.15 | 0.20 | 0.10 | 0.07 | 0.03 | 0.01 |
| | RB tree | 0.76 | 0.49 | 0.65 | 0.62 | 0.30 | 0.46 | 0.09 | 0.09 | 0.12 | 0.12 | 0.05 | 0.02 | 0.01 |
| | B+ tree | 0.98 | 0.97 | 0.97 | 0.31 | 0.38 | 0.23 | 0.14 | 0.08 | 0.23 | 0.12 | 0.01 | 0.03 | 0.00 |
| | K-D Tree | 0.85 | 0.59 | 0.47 | 0.45 | 0.00 | 0.03 | 0.00 | 0.00 | 0.00 | 0.01 | 0.00 | 0.00 | 0.00 |
| | K-D Heap | 0.10 | 0.10 | 0.10 | 0.11 | 0.05 | 0.05 | 0.05 | 0.03 | 0.04 | 0.04 | 0.01 | 0.01 | 0.01 |
| | *Category avg.* | *0.72* | *0.58* | *0.64* | *0.45* | *0.33* | *0.34* | *0.18* | *0.16* | *0.20* | *0.14* | *0.08* | *0.05* | *0.02* |
| Network | Graph | 0.96 | 0.78 | 0.87 | 0.67 | 0.11 | 0.06 | 0.13 | 0.06 | 0.15 | 0.05 | 0.02 | 0.01 | 0.01 |
| | DSU | 1.00 | 0.97 | 0.98 | 1.00 | 0.99 | 0.02 | 0.73 | 0.82 | 0.02 | 0.00 | 0.00 | 0.16 | 0.02 |
| | Geom Graph | 0.16 | 0.13 | 0.16 | 0.13 | 0.02 | 0.02 | 0.00 | 0.01 | 0.07 | 0.02 | 0.02 | 0.00 | 0.00 |
| | *Category avg.* | *0.71* | *0.63* | *0.67* | *0.60* | *0.38* | *0.03* | *0.29* | *0.30* | *0.08* | *0.03* | *0.01* | *0.06* | *0.01* |
| Hybrid | Bloom Filter | 0.77 | 0.86 | 0.61 | 0.74 | 0.16 | 0.04 | 0.12 | 0.04 | 0.04 | 0.04 | 0.02 | 0.01 | 0.00 |
| | DAWG | 0.34 | 0.23 | 0.25 | 0.14 | 0.06 | 0.07 | 0.00 | 0.06 | 0.05 | 0.06 | 0.01 | 0.00 | 0.00 |
| | *Category avg.* | *0.56* | *0.55* | *0.43* | *0.44* | *0.11* | *0.06* | *0.06* | *0.05* | *0.05* | *0.05* | *0.02* | *0.00* | *0.00* |
| **Score** | *Overall avg.* | **0.79** | **0.71** | **0.71** | **0.63** | **0.51** | **0.41** | **0.39** | **0.32** | **0.31** | **0.27** | **0.17** | **0.13** | **0.09** |

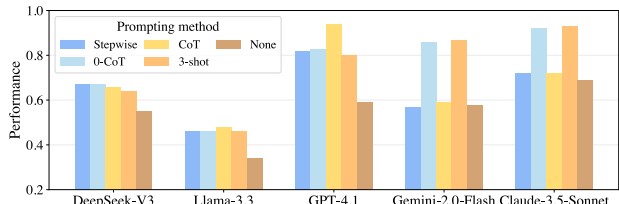

*Figure 3.* Average scores across all tasks for five instruction-tuned models under different prompting strategies.

tributes. For instance, while they perform well on QUEUE, their accuracy drops 30-50% on PRIORITY QUEUE, where each element includes a priority. Similarly, in the HASHMAP task, manual inspection of errors shows that models confuse keys and values, delete the wrong items, or hallucinate entries. These results reveal a key limitation for real-world deployment, where managing entities with multiple properties, such as deadlines and key-value records, is common.

**Multi-hop reasoning in hierarchical or network structures remains a key challenge.** We see from Table 2 that, while models perform reliably on BST, their accuracy drops by over 30% on RED-BLACK TREES, reflecting the difficulty of handling multi-hop properties such as maintaining balance across ancestral levels. Performance declines further on B+ TREES, which requires reasoning over wider spans of child pointers, and on GRAPH traversal tasks with many-to-

many relationships. Manual inspection of reasoning traces reveals failures to retain earlier information: in GEOMETRIC GRAPH-*long*, for example, all GPT-4.1 errors stemmed from dropping nodes during intermediate steps. We provide an additional analysis of zero-score cases in Appendix E.4. We compare performances in a single task across models and multiple tasks for the same model. Results show that near-zero performance can arise from qualitatively different failure modes. For KD-TREE, models often fail due to early median-selection errors that propagate through recursive construction, while Qwen3-8B shows broader issues such as unparsable outputs, hallucinated or dropped elements, and reliance on superficial output patterns. These findings suggest that task failures stem not only from isolated mistakes, but also from limitations in sustained multi-step construction and instruction following.

**Prompting can help, but only when carefully designed.** As shown in Figure 3, the **None** prompt performs worst, suggesting that prompts encouraging stepwise reasoning is generally beneficial. Our findings indicate two practical strategies: (i) Lightweight prompts such as **Stepwise** and **0-CoT** are easily implemented and consistently improve performance (Section E); (ii) Crafted prompts like **CoT** and **3-shot** are most effective for uncommon data structures, but require careful design. A representative case is SUFFIX-TREE: across all models, zero-shot accuracy is below 0.40,

but a well-designed CoT prompt doubles accuracy for three models (Section E). We include additional CoT analysis with practical takeaways in Section E.1.

### 4.1.2. REASONING MODELS

*Table 3.* Scores of the `challenge` subset for reasoning models.

| Task | GPT-5 | o4-mini | Gemini-2.5-Pro | DeepSeek-R1 | Claude-3.7-Sonnet |
|---|---|---|---|---|---|
| Challenge Score | 0.46 | 0.32 | 0.30 | 0.21 | 0.10 |
| Priority Queue | 0.28 | 0.30 | 0.23 | 0.48 | 0.04 |
| Skip List | 0.51 | 0.41 | 0.53 | 0.54 | 0.61 |
| Heap | 0.71 | 0.71 | 0.53 | 0.27 | 0.13 |
| Red-black tree | 0.59 | 0.37 | 0.08 | 0.37 | 0.12 |
| B+ Tree | 0.98 | 0.94 | 0.94 | 0.21 | 0.13 |
| K-D Tree | 0.67 | 0.38 | 0.16 | 0.01 | 0.00 |
| K-D Heap | 0.00 | 0.00 | 0.00 | 0.01 | 0.00 |
| Geom Graph | 0.19 | 0.00 | 0.02 | 0.01 | 0.00 |
| Bloom Filter | 0.47 | 0.07 | 0.69 | 0.31 | 0.00 |
| DAWG | 0.00 | 0.06 | 0.01 | 0.00 | 0.00 |
| Hashmap | 0.71 | 0.26 | 0.11 | 0.12 | 0.04 |

**Reasoning models remain brittle on complex and spatial data structures.** From Table 2, we see reasoning models outperform instruction-tuned models in general, especially on hierarchical and networked structures. However, the overall score remains below 0.5 on `challenge` in Table 3, in particular for *long* tasks and complex data structures. For example, the highest score on SKIP LIST is only 0.61 for the `challenge` subset, despite its prevalence in introductory-level textbooks and its wide use in dictionaries and maps. Notably, accuracy on K-D TREE, K-D HEAP, and GEO-METRIC GRAPHS is low even for *short* tasks, suggesting that high-dimensional spatial reasoning remains a significant challenge (Section D). To further probe these limitations, we introduce `spatial`, described in Section 4.2.

**Implicit priors may hinder instruction following.** Ablation on K-D HEAP (Figure 5a) shows that switching the tie-breaking rule from lexicographic order to Euclidean norm leads to a drop of over 0.40 on o4-mini. We observe that the model continues to apply lexicographic order, even when instructed otherwise. When queried directly, o4-mini confirms that it assumes lexicographic order for K-D heaps by default. These results suggest that o4-mini struggles to adapt to user-defined constraints, likely due to its reliance on learned priors from training.

### 4.2. Can LLMs reason structurally on spatial data?

Real-world data is often represented in high-dimensional feature spaces. To assess whether LLMs can reason over such spatial data, we extend the benchmark with the `spatial` probe, which includes three multi-dimensional variants: K-D HEAP, K-D TREE, and GEOMETRIC GRAPH embedded in Euclidean space. These structures are common in practice; for instance, K-D trees are key data structures

in computer vision and graphics. Given the complexity of these tasks, we use GPT-4.1 with the **Stepwise** prompt to encourage intermediate reasoning steps.

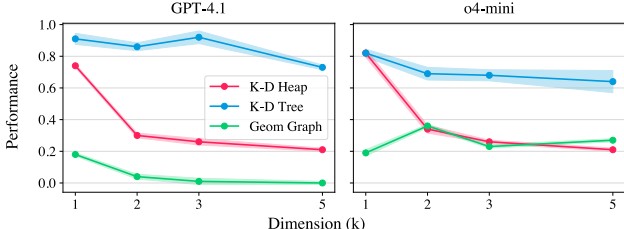

*Figure 4.* Scores for the three spatial data structures with input data of varying dimensionality ($k = 1, 2, 3, 5$).

**Performance degrades as dimensionality increases.** In Figure 4, accuracy declines for both models as dimensionality increases. Higher-dimensional data challenges models with complex computation over distance metrics and partitions, limiting their effectiveness in spatial tasks. For instance, K-D trees are widely used to expedite nearest neighbor queries over 128-dimensional SIFT descriptors in computer vision (Silpa-Anan & Hartley, 2008). Interestingly, 2D outperforms 1D in GEOMETRIC GRAPH, likely because it is more common in training data from textbooks.

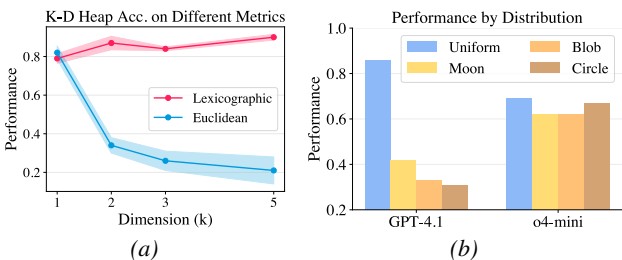

*Figure 5.* (a) Performance on the two metrics for K-D HEAP. (b) K-D Tree with varying input data distributions.

**Limited robustness to non-uniform data distributions.** We assess LLM robustness to distribution shifts by comparing performance on uniformly sampled versus skewed or clustered data. We test K-D tree construction tasks using three non-uniform distributions from scikit-learn (Pedregosa et al., 2011): circles, moons, and blobs (illustration in Figure 8, Section F). As shown in Figure 5b, GPT-4.1's performance drops sharply on non-uniform inputs, possibly due to a higher likelihood of uniformly distributed examples in the training data. Since task difficulty is held constant, this gap suggests a reliance on pattern memorization rather than true reasoning. In contrast, o4-mini shows a smaller drop, indicating that reasoning models may generalize better to distribution shifts. A more in-depth inspection of errors and discussion on root causes can be found in Section F.

### 4.3. Can LLMs reason structurally on realistic tasks?

While the previous sections evaluated LLMs on canonical data structures, real-world use cases are often described in messy, context-rich language. To bridge this gap, we extend the benchmark with the `realistic` probe, which embeds data structure tasks in narrative contexts, evaluating whether LLMs can generalize structural reasoning beyond formal descriptions. This probe serves as an auxiliary transfer probe that tests whether a model's ability to execute explicit structural computations transfers from formal prompts to context-rich language. It introduces an additional step not present in the formal setting: when structural reasoning problems are presented in natural language, models must first (i) recover the relevant structure from context-rich language before they can (ii) apply the correct structural execution.

---

**Example prompt for QUEUE in the `realistic` probe.**

On a sunny afternoon in the park, an ice cream truck rolled in... Children began to form a line, each newcomer taking their place at the end while the vendor served from the front...

- Isabella Miller ran over and joined the ice cream line.
- The next kid in line was served promptly.
- ...

**Q**: What is the order of the remaining kids in line?

---

We design three real-world scenarios that implicitly require data structures: QUEUE (children buying ice cream), BINARY SEARCH TREE (clinic appointments), and GRAPH (galaxy traveling game). Synthetic data follows the same distributions as Section 3.4, with realistic substitutions (e.g., names for integers). Each scenario was written by humans and paraphrased by GPT-4o. All prompts were reviewed by three annotators for clarity and unambiguity. Details are provided in Section G.

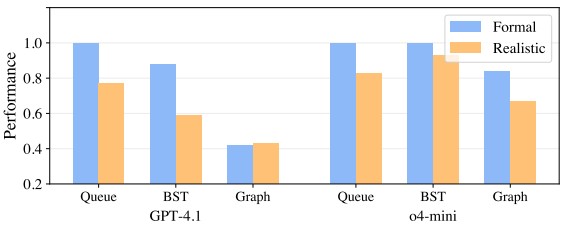

*Figure 6.* Model performance on formal and realistic usages.

**LLMs struggle when shifting from formal to realistic usage.** Performance generally drops when tasks are using realistic, context-rich language compared to formal descriptors, despite identical problem distribution (Figure 6). Errors in `realistic` mainly have two types. One type is the same underlying reasoning failures already seen in canonical data structures (e.g., multi-hop, backtracking failures).

The other type arises from the additional difficulty of recovering and maintaining the relevant structures from narrative language. For example, not recognizing that "Another child was served and happily walked away" implies a dequeue operation. Models also struggle with complex tie-breaking rules expressed in natural language, such as lexicographical ordering ("Kelvin" before "Krypton"), which can be harder than comparing simple numeric values. In addition, hallucination errors are common, with models generating names or timestamps that do not appear in the input. The higher accuracy on formal descriptors may also stem from training on textbook-style patterns, where integers and explicit syntax are common. This observation suggests that even reasoning models struggle to apply reasoning in realistic contexts. Bridging this gap is crucial for reliable deployment and presents a key direction for future research.

### 4.4. Can LLMs reason structurally with code?

As motivated in Sections 1 and 2, our benchmark targets LLMs' *inherent* reasoning independent of code execution or tool use. Nonetheless, to assess whether code generation provides any benefit, we run ablations on six models using the `code` probe across three modes: (i) *CodeOnly*, where models generate Python code executed by an external interpreter; (ii) *CodeEnforce*, where models must write code and reason through its execution internally without relying on an interpreter; and (iii) *CodeMaybe*, similar to *CodeEnforce* but makes code generation optional. Full details on the experimental setup and results are provided in Section H.

*Table 4.* Average performance on seven data structures across three code generation modes and the default setting.

| Mode | GPT-4.1 | o4-mini | Gemini-2.0-Flash | Gemini-2.5-Pro | Claude-3.5-Sonnet | Claude-3.7-Sonnet |
|---|---|---|---|---|---|---|
| Default | 0.40 | 0.73 | 0.38 | 0.55 | 0.44 | 0.55 |
| CodeMaybe | 0.38 | 0.76 | 0.41 | 0.55 | 0.43 | 0.57 |
| CodeEnforce | 0.38 | 0.75 | 0.41 | 0.53 | 0.42 | 0.55 |
| CodeOnly | 0.95 | 0.82 | 0.44 | 0.57 | 0.74 | 0.87 |

**Models cannot reason over generated code.** In Table 4, performance in *CodeMaybe* and *CodeEnforce* matches the default setup, despite writing high-quality code as shown in *CodeOnly*. This observation suggests that writing code offers little benefit over natural language reasoning when models must internally simulate execution. This reinforces our claim: LLMs remain limited in their ability to perform structural reasoning, even when guided by their own code.

**Code helps only with standard tasks and fails on natural language ones.** As shown in Figure 7, with an external interpreter in *CodeOnly*, models perform well on GEOM GRAPH, a standard structure in computer graphics with widely available implementations online.

In contrast, o4-mini struggles on the less familiar DAWG, where custom constraints enforce unambiguous outputs, suggesting reliance on memorized solutions rather than genuine reasoning. Perfor-

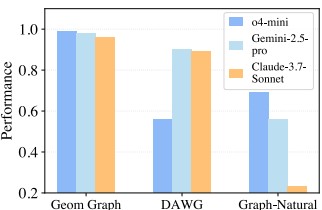

*Figure 7.* Scores for *CodeOnly*.

mance drops further on GRAPH-NATURAL, where models default to brittle pattern matching (e.g., rigidly mapping "A tunnel links A and B" to "G.add_edge(A, B)") but failing to understand paraphrases like "Couriers frequently travel the tunnel connecting A to B." These results highlight the fragility of structural reasoning in the presence of context-rich language, even with code generation and external execution.

## 5. Conclusion

Can LLMs reason structurally? Through DSR-Bench, we provide a systematic answer: not yet. Instruction-tuned models struggle with multi-attribute reasoning (e.g., database indexing) and multi-hop reasoning (e.g., trip planning), while reasoning models achieve only 0.46 accuracy on complex structures and can ignore user-defined constraints. These limitations highlight the need for architectures that support precise function computation, memory mechanisms, and the flexibility to adapt to user-defined constraints. Evaluations on high-dimensional data (spatial) and context-rich scenarios (realistic) further reveal gaps between current reasoning capabilities and real-world deployment. Code generation modes (code) show that models cannot reliably reason about their own code, often reverting to memorized patterns or brittle mappings even with external execution.

**Limitations and future directions**  We focus on whether LLMs can perform the required structural reasoning for specified tasks by providing detailed task descriptions to ensure fairness and reduce bias from differences in prior knowledge. A valuable next step is evaluating whether models can identify the appropriate data structures and algorithms given only the task requirement. DSR-Bench is extensible to such settings, and Section L presents a preliminary study of this identification challenge. Another promising direction is incorporating carefully designed LeetCode-style tasks, but with an emphasis on inherent reasoning over coding proficiency. Such an extension would complement the fundamental abilities studied here by evaluating higher-level skills, including problem decomposition, advanced algorithm formulation, and computational complexity analysis. In addition, while we focus on final-answer evaluation to allow different models to adopt flexible reasoning strategies,

intermediate-step scoring could provide additional insights into error types and model bottlenecks. Future extensions of DSR-Bench may therefore incorporate more tailored intermediate scoring systems that account for dynamic reasoning and error-correction strategies. Similarly, while we treat hallucinations as reasoning failures, alternative error classification schemes may treat them as a separate category of generation failures. Furthermore, DSR-Bench minimizes test-set contamination by generating new instances from hand-crafted prompt templates and randomized generators, consistent with prior definitions of contamination (White et al., 2025; Golchin & Surdeanu, 2025; Oren et al., 2024). However, given the widespread availability of data structure knowledge, DSR-Bench does not aim to eliminate task-level contamination. Future extensions can instead explicitly target task-level novelty by designing entirely new families of structural reasoning tasks rather than relying on canonical textbook problems.

Notably, we use "structural reasoning" in an operational, behavioral sense: whether a model can correctly maintain intermediate states and follow task-specific rules. Therefore, DSR-Bench doesn't assume that success or failure can be cleanly partitioned into "true reasoning" or "pattern matching", as this distinction is not well-defined and remains actively debated. In our context, *memorization* corresponds to success driven by prior exposure to the same (or nearly identical) instances, which is why minimizing test-set contamination is important. Meanwhile, *generalization* refers to the ability to apply learned computational operations to unseen instances. In this sense, DSR-Bench is best interpreted as evaluating *structural generalization* capabilities: whether models can correctly perform specific structural computations under controlled changes in lengths, constraints, distributions, and linguistic forms. We do not intend to imply that DSR-Bench measures "reasoning" in the broadest sense. Rather, it evaluates whether models can solve tasks by correctly performing explicit structural computations. We view this ability as a necessary but not sufficient prerequisite for broader *algorithmic reasoning*.

**Outlook**  DSR-Bench provides a systematic framework for evaluating algorithmic reasoning through the lens of data structures. It provides the community with a powerful diagnostic tool: researchers can pinpoint failure modes, test targeted improvements, and measure progress on specific relationship types, paving the way for algorithm-centric model design (Eberle et al., 2025). It also raises new questions: Can LLMs dynamically choose reasoning strategies? Can multimodal LLMs reason over DSR-Bench tasks using visual inputs? Furthermore, it is well-suited as a testbed for mechanistic interpretability and post-training due to its synthetic and verifiable nature. We invite the community to use DSR-Bench to explore these future directions.

## Acknowledgements

YH is supported by a Cubist PhD Fellowship. YL is supported by an Amazon AI Fellowship. This work was also supported in part by NSF grant CCF-2338226. We thank Connor Lawless, Han Xuanyuan, and members of the Vitercik Lab for their meaningful discussions and valuable feedback on the manuscript.

## Impact Statement

This paper presents work whose goal is to advance the field of Machine Learning. There are many potential societal consequences of our work, none of which we feel must be specifically highlighted here.

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

## A. Additional related works

**LLM Benchmarking** LLMs have demonstrated remarkable performance across a wide range of applications, prompting growing interest in understanding their capabilities and limitations. Recent efforts have focused on systematically benchmarking and evaluating LLMs on core natural language processing tasks, including language understanding (Wang et al., 2024b), text generation (Maynez et al., 2023; Singh et al., 2024), reasoning (Valmeekam et al., 2023; Wang et al., 2024a), and machine translation (Yao et al., 2024). Additionally, some benchmarks evaluate alignment and safety, such as robustness, bias, and trustworthiness (Siska et al., 2024; Luo et al., 2025; Koo et al., 2024; Li et al., 2024b; Zhang et al., 2024b; Liu et al., 2024d). Specialized benchmarks have also emerged in scientific and technical domains, including mathematics (White et al., 2025; Liu et al., 2024a), programming (White et al., 2025; Jain et al., 2025; Chen et al., 2021; Zheng et al., 2023), data analysis (Sui et al., 2024; Liu et al., 2023b; White et al., 2025; Tang et al., 2024; Jiang et al., 2023), and medicine (Li et al., 2024a; Liu et al., 2023a; Xu et al., 2024b; Kanithi et al., 2026). While several studies examine how LLMs convert unstructured inputs into tabular or relational formats (Tang et al., 2024; Jiang et al., 2023), our work explores a distinct question: How well can LLMs construct, manipulate, and reason about classic *data structures* such as stacks, trees, and graphs? To our knowledge, this is the first comprehensive benchmark targeting this capability, offering a conceptually different evaluation from prior work on structured data.

**Reasoning benchmarks** Existing LLM reasoning benchmarks are predominantly high-level and domain-specific, targeting math (Cobbe et al., 2021; Liu et al., 2024a;c), STEM (Hendrycks et al., 2021), and logic puzzles (White et al., 2025; Giadikiaroglou et al., 2024). These often require complex responses with intertwined reasoning steps, relying on subjective human or LLM-based evaluation (Chiang et al., 2024; Feuer et al., 2025; Ye et al., 2025). We focus on structural reasoning, an implicit requirement underlying problem-solving across many domains, and use it as a framework to evaluate algorithmic reasoning. By using data structures as clear abstractions of different data relationships, we isolate algorithmic reasoning from domain-specific complexities.

**Coding benchmarks** Coding benchmarks evaluate how well LLMs write syntactically correct code or function as coding agents (Chen et al., 2021; Zheng et al., 2023; Jimenez et al., 2024; Jain et al., 2025; White et al., 2025; Aider-AI, 2025), typically requiring external interpreters for verification. While useful, these benchmarks conflate reasoning with tool execution and are limited to domains where coding applies. Coding tasks for algorithmic reasoning are also susceptible to data contamination due to the abundance of online coding materials for algorithmic tasks. In contrast, we target LLMs' inherent reasoning skills independent of external tools, reflecting the broader goal of assessing progress toward general intelligence. Prior works (Malfa et al., 2024; 2025; Liu et al., 2025) study code simulation as a lens to probe general reasoning. Instead, we specifically focus on structural reasoning via data structure tasks and include `code` to probe whether code generation aids this process.

## B. Details of data structures and operations

In this section, we list the data structures and the corresponding operations tested in Table 5. We then provide detailed descriptions of each data structure and explain how we specify their implementations to eliminate ambiguity.

**Array** An array contains a list of elements stored in contiguous memory. We test its access, insertion, deletion, reversal, and search operations. To remove ambiguity, we specify that the array is 0-indexed. For operations like deletion, if duplicates exist, we delete the first occurrence. The final state is a list of elements in the array.

**Stack** A stack is a linear data structure that follows a Last-In-First-Out (LIFO) order. We test compound operations consisting of random sequences of push and pop from the top of the stack. The final state is a list of remaining elements in the stack.

**Queue** A queue is a linear data structure that follows a First-In-First-Out (FIFO) order. We evaluate compound operations of enqueue to the back and dequeue from the front. The final state is a list of remaining elements in the queue.

**LRU Cache** An LRU (Least Recently Used) cache stores a fixed number of items and evicts the least recently accessed one when full. We evaluate this caching operation with a sequence of requests as input. The final state is a set of elements in the LRU cache.

*Table 5.* Summary of data structures and associated operations in `DSR-Bench`. Data structures marked with * are included in the `challenge` subset. All compound operations without explicit specification consist of (insert, delete).

| Category | Data Structure | Description | Operation | Application |
|---|---|---|---|---|
| Linear | Array | Contiguous memory | Access, Delete, Insert, Reverse, Search | Data storage |
| Temporal | Stack | LIFO (Last-In, First-Out) | Compound (Push, Pop) | Syntax parsing |
| | Queue | FIFO (First-In, First-Out) | Compound | OS management |
| | LRU Cache | Least-recently-used | Cache (Evict, Add) | Web browsers |
| | Priority Queue* | Priority ordering | Compound | Job scheduling |
| Associative | Hashmap* | Key-value storage | Compound | Large-scale storage |
| | Trie | Hierarchical mapping of strings | Compound | Autocomplete |
| | Suffix Tree | Text indexing via suffixes | Compound | DNA pattern matching |
| | Skip List* | Probabilistic layers for fast search | Compound | Concurrent databases |
| Hierarchical | Binary Search Tree | Hierarchical storage | Pre/In/Post-Order Traversal, Insert, Remove, Compound | Computer networks |
| | Heap* | Complete binary tree with priority ordering | Heapify, Compound | Memory management |
| | Red-Black Tree* | Self-balanced tree | Construct, Compound | Database indexes |
| | B+ Tree* | Multi-way balanced tree | Compound | File systems |
| | K-D Tree* | Hierarchical, spatial partition | Construct, Compound | 3D graphics |
| | K-D Heap* | Hierarchical, complete binary tree, high-dimensional priority | Compound | GPU job scheduling |
| Network | Graph | Many-to-many relationships | Breadth-First Traversal, Depth-First Traversal | Social networks |
| | Disjoint Set Union* | Sets partition & union | Compound (Union, Find) | Physics simulation |
| | Geometric Graph* | Graph modeling spatial data | Construct | Public transportation |
| Hybrid | Bloom Filter* | Probabilistic set and hashmap | Compound | Spam detection |
| | Directed Acyclic Word Graph* | Graph and trie tree | Compound | Compilers |

**Priority Queue**  A priority queue stores elements with integer priorities, allowing access to the highest-priority element. We test compound operations including insert, remove, raise key, and decrease key, using a Fibonacci heap. Ties are broken by insertion order. The final state is a level-order traversal of the Fibonacci heap forest, outputting (value,priority) pairs, with nodes at each level sorted by descending priority and ties broken by larger value first.

**Hashmap**  A hashmap is a key-value structure supporting fast access, insertion, and deletion via hashed keys. We test compound insert and delete operations, specifying the hash function and using chaining for collision resolution. The final state is a list of key-value pairs per bucket, preserving insertion order within each chain.

**Trie**  A trie is a tree-based data structure for storing strings, where each node represents a character and paths from the root to leaves represent complete words, where common prefixes are shared. When generating strings, we increase the likelihood of shared prefixes to ensure the resulting trie has meaningful structure. The final state is a pre-order traversal of the trie, where each node's children are visited in lexicographical order to ensure an unambiguous representation.

**Suffix Tree**  A suffix tree is a compressed trie built from all suffixes of a string, where each edge can represent multiple characters and each path from the root corresponds to a substring. We test the construction of a suffix tree from a given word, appending a terminal character "$" to ensure a unique structure. The final state is a pre-order traversal collecting edge labels, with child edges visited in lexicographical order and "$" taking priority.

**Skip List**     A skip list is a probabilistic data structure composed of multiple layers of linked lists, where higher layers allow "skipping" over elements for faster access. We test compound operations of insert and delete. Insertion begins at the bottom layer, with the element randomly promoted to higher levels; pointers are updated at each level to preserve the structure. To remove ambiguity, promotion probabilities are explicitly specified in the prompts. The final state is represented as a list of lists, each corresponding to a layer of the skip list.

**Binary Search Tree (BST)**     A binary search tree is a hierarchical structure where each node has at most two children: the left holds smaller values, and the right holds larger ones. We test insert, remove, tree traversals (pre-order, in-order, post-order), depth computation, and compound insert-remove operations. Inputs are guaranteed to contain no duplicates to ensure unique outputs. The final state for traversal tasks is a list of elements in the specified order, while for insert, remove, and compound tasks, it includes both pre-order and post-order traversals.

**Heap**     A heap is a complete binary tree that satisfies the min-heap property, where each parent node is less than or equal to its children. We test both heapify and compound insert-delete operations using an array-based heap. Comparisons follow min-heap ordering, with ties broken by preferring the left child. The final state is the array representation of the heap.

**Red-Black (RB) Tree**     A red-black tree is a self-balancing binary search tree where each node is colored red or black and must satisfy specific balance rules: no two consecutive red nodes are allowed, and all root-to-leaf paths must have the same number of black nodes. We test both construction and compound (insert, delete) operations. The final state is a pre-order traversal of the nodes, represented as tuples (value, color).

**B+ Tree**     A B+ tree is a multi-way search tree used in databases and filesystems, where values are stored in leaf nodes and internal nodes serve as routing indexes. Leaf nodes are linked for efficient range queries. We specify splitting and merging rules to ensure unambiguous, balanced updates during compound insert and delete operations. The final state is a pre-order traversal of nodes, with keys in each node sorted in ascending order.

**K-D Tree**     A K-D (k-dimensional) tree recursively partitions space by alternating the splitting axis at each level. Each node represents a point and divides the space into two halves based on a chosen coordinate. It is commonly used for spatial indexing, range queries, and nearest neighbor search. We test the construction of K-D trees across different dimensionalities, specifying the axis splitting sequence and tie-breaking rules (e.g., median selection for even-sized splits) to ensure consistency. The final state is a pre-order traversal of the tree.

**K-D Heap**     A K-D heap maintains heap order based on a $k$-dimensional priority with a comparison metric, enabling efficient access to extremal points in multidimensional datasets. We test compound operations of insert and delete across different dimensionalities. We specify an array-based heap implementation, with comparisons based on Euclidean distance and tie-breaking rules that prefer the left child in case of a tie. The final state is a list of vectors representing the contents of the min-heap.

**Graph**     A graph is a collection of nodes connected by edges, which can be directed or undirected, and is used to model networks, dependencies, and paths. We define graphs using edge list statements and test both breadth-first and depth-first traversals from a given source node, visiting neighbors in ascending order. Node values are unique to ensure consistent outputs. The final state is the list of nodes visited during the traversal.

**Disjoint Set Union (DSU)**     A disjoint set union maintains a partition of elements into disjoint subsets, supporting efficient merges and membership queries. Internally, it forms a forest where each node points to a representative root. We test two operations: a sequence of unions between subsets, followed by queries for each element's representative. To ensure consistency, we specify that lower-rank roots are always attached to higher-rank ones. The final state lists the representative root of each input element in its original order.

**Geometric (Geom) Graph**     Geometric graphs are graphs with nodes embedded in geometric space, typically Euclidean, where edges are formed based on spatial relationships such as proximity. They are widely used in robotics, computer graphics, and sensor networks where spatial structure is essential. We compute the Euclidean distance between each pair of points and add an edge if the distance is below a given threshold, assigning the edge a weight equal to that distance. The

final state is a breadth-first traversal from a specified source node, exploring all neighbors at each level before proceeding. We specify the order of search based on the edge weights.

**Bloom Filter**   A (counting) Bloom filter is a compact, probabilistic data structure for set membership testing, guaranteeing no false negatives and allowing a tunable false positive rate. It uses multiple hash functions to map each element to several positions in a counter array, incrementing or decrementing counts. We test on compound operations of insert and delete. We specify the hash functions used in the prompt to avoid ambiguity. The final state is the array of counters representing the Bloom filter.

**Directed Acyclic Word Graph (DAWG)**   A Directed Acyclic Word Graph (DAWG) is a compressed data structure for storing a set of words, sharing both prefixes and suffixes. Nodes indicate whether they mark the end of a word, and edges are labeled with characters. Unlike a trie, a DAWG merges equivalent subtrees to reduce redundancy, making it well-suited for large static dictionaries and lexicon lookups. We test compound operations of insert and delete, specifying that merging should occur at the final step along with the merging rules. To ensure a meaningful structure, we increase the likelihood of generating words with shared prefixes. The final state is a breadth-first traversal from the root (an empty string), where each node is recorded by the prefix it represents and whether it marks the end of a word.

## C. Examples of prompting strategies

In this section, we illustrate prompting methods using compound operations of QUEUE as an example.

**Stepwise**   This method explicitly adds a `steps` attribute in the JSON schema of Structured Output, guiding the model to produce operations in a sequential and interpretable manner. Below is an example schema for an array task:

```
class Step(BaseModel):
    explanation: str
    output: str
class ArraySchema(BaseModel):
    steps: list[Step]
    final_answer: int
```

> ### *Stepwise* prompting on compound operations of QUEUE.
>
> A queue is a data structure in which items are added at one end and removed from the other, maintaining a first-in, first-out (FIFO) order. You should create a queue. There are two types of operations: 1. (enqueue, k) means an element k is appended to the queue as the last element. 2. (dequeue) means the first element of the queue is deleted. You are given an empty queue initially.
> **Q**: What is the final queue, when performing the following operations:
> - (enqueue, 49)
> - (dequeue)
> - (enqueue, 86)
> - (enqueue, 52)
> Answer the question in 8000 tokens.

**0-CoT**   This method appends the phrase "Let's think step by step" to the prompt to encourage reasoning without providing exemplars.

---

*0-CoT* **prompting on compound operations of** QUEUE.

A queue is a data structure in which items are added at one end and removed from the other, maintaining a first-in, first-out (FIFO) order. You should create a queue. There are two types of operations: 1. (enqueue, k) means an element k is appended to the queue as the last element. 2. (dequeue) means the first element of the queue is deleted.

You are given an empty queue initially.

**Q**: What is the final queue, when performing the following operations:

- (enqueue, 49)

- (dequeue)

- (enqueue, 86)

- (enqueue, 52)

Let's think step by step.
Answer the question in 8000 tokens.

---

**CoT**   This strategy provides a single example that includes both intermediate reasoning steps and the final answer.

---

*CoT* **prompting on compound operations of** QUEUE.

A queue is a data structure in which items are added at one end and removed from the other, maintaining a first-in, first-out (FIFO) order. You should create a queue. There are two types of operations: 1. (enqueue, k) means an element k is appended to the queue as the last element. 2. (dequeue) means the first element of the queue is deleted. You are given an empty queue initially.

Q: What is the final queue, when performing the following operations:
- (enqueue, 21)
- (enqueue, 3)
- (dequeue)
- (dequeue)
- (enqueue, 48)

A: Initially, the queue is []. After (enqueue, 21), it becomes [21]. After (enqueue, 3), it becomes [21, 3]. After (dequeue), it becomes [3]. After (dequeue), it becomes []. After (enqueue, 48), it becomes [48]. The final queue is [48].

**Q**: What is the final queue, when performing the following operations:
- (enqueue, 49)
- (dequeue)
- (enqueue, 86)
- (enqueue, 52)
Answer the question in 8000 tokens.

---

**3-shot**   This strategy provides three input-output examples to guide the model through pattern matching and demonstration.

---

*3-shot* **prompting on compound operations of** QUEUE.

---

A queue is a data structure in which items are added at one end and removed from the other, maintaining a first-in, first-out (FIFO) order. You should create a queue. There are two types of operations: 1. (enqueue, k) means an element k is appended to the queue as the last element. 2. (dequeue) means the first element of the queue is deleted. You are given an empty queue initially.

Q: What is the final queue, when performing the following operations:

- (enqueue, 21)
- (enqueue, 3)
- (dequeue)
- (dequeue)
- (enqueue, 48)

A: The final queue is [48]. (... Example 2...) (... Example 3...)

**Q**: What is the final queue, when performing the following operations:

- (enqueue, 49)
- (dequeue)
- (enqueue, 86)
- (enqueue, 52)

Answer the question in 8000 tokens.

---

**None**   This method adds the instruction "No additional text needed" to prompt concise, direct answers that fit within the token limit and conform to the structured output format.

---

**None prompting on compound operations of** QUEUE.

---

A queue is a data structure in which items are added at one end and removed from the other, maintaining a first-in, first-out (FIFO) order. You should create a queue. There are two types of operations: 1. (enqueue, k) means an element k is appended to the queue as the last element. 2. (dequeue) means the first element of the queue is deleted. You are given an empty queue initially.

**Q**: What is the final queue, when performing the following operations:

- (enqueue, 49)
- (dequeue)
- (enqueue, 86)
- (enqueue, 52)

No additional text needed.
Answer the question in 8000 tokens.

---

## D. Accuracy by task and length level across all models

In this section, we provide supplementary accuracy tables for all models in DSR-Bench, broken down by task and length level. Table 6 summarizes the accuracy of instruction-tuned models on a subset of basic data structures across different length levels. For detailed per-model results across all tasks and length levels, see  Table 7 (GPT-5),  Table 8 (o4-mini), Table 9 (Gemini-2.5-Pro),  Table 10 (Claude-3.7-Sonnet),  Table 11 (DeepSeek-R1),  Table 12 (GPT-4.1),  Table 13 (Gemini-2.0-Flash),  Table 14 (Claude-3.5-Sonnet),  Table 15 (DeepSeek-V3), and  Table 16 (Llama-3.3).

*Table 6.* Average accuracy on basic data structure tasks for instruction-tuned models (3 runs, scaled to [0, 1], rounded to two decimals).

| Category | DS | Length | GPT-4.1 | Gemini-2.0-Flash | Claude-3.5-Sonnet | DeepSeek-V3 | Llama-3.3 |
|---|---|---|---|---|---|---|---|
| Linear | Array | Short | 0.98 | 0.98 | 1.00 | 1.00 | 0.89 |
| | | Medium | 0.95 | 0.92 | 0.96 | 0.97 | 0.70 |
| | | Long | 0.88 | 0.88 | 0.91 | 0.96 | 0.48 |
| Temporal | Queue | Short | 0.82 | 0.87 | 0.87 | 0.84 | 0.58 |
| | | Medium | 0.59 | 0.33 | 0.67 | 0.38 | 0.12 |
| | | Long | 0.19 | 0.10 | 0.79 | 0.07 | 0.06 |
| | Stack | Short | 0.97 | 0.67 | 1.00 | 0.70 | 0.09 |
| | | Medium | 0.49 | 0.37 | 1.00 | 0.49 | 0.04 |
| | | Long | 0.18 | 0.03 | 0.98 | 0.04 | 0.00 |
| Associative | Hashmap | Short | 0.19 | 0.28 | 0.37 | 0.00 | 0.00 |
| | | Medium | 0.00 | 0.01 | 0.10 | 0.00 | 0.00 |
| | | Long | 0.00 | 0.00 | 0.00 | 0.00 | 0.00 |
| Hierarchical | BST | Short | 0.82 | 0.63 | 0.89 | 0.76 | 0.46 |
| | | Medium | 0.56 | 0.37 | 0.66 | 0.55 | 0.31 |
| | | Long | 0.39 | 0.29 | 0.57 | 0.43 | 0.26 |
| Network | Graph | Short | 0.41 | 0.15 | 0.15 | 0.16 | 0.06 |
| | | Medium | 0.05 | 0.02 | 0.02 | 0.02 | 0.01 |
| | | Long | 0.00 | 0.00 | 0.00 | 0.00 | 0.00 |

## D.1. Performance of GPT-5

*Table 7.* Mean (± std) accuracy of **GPT-5** on all `DSR-Bench` tasks over three runs. Data structures marked with * are included in the `DSR-Bench-challenge` subset.

| Category | Data Structure | Operation | Short | Medium | Long |
|---|---|---|---|---|---|
| Linear | Array | Access | 1.00 (0.00) | 1.00 (0.00) | 1.00 (0.00) |
| | | Delete | 1.00 (0.00) | 1.00 (0.00) | 1.00 (0.00) |
| | | Insert | 1.00 (0.00) | 1.00 (0.00) | 1.00 (0.00) |
| | | Reverse | 1.00 (0.00) | 1.00 (0.00) | 1.00 (0.00) |
| | | Search | 1.00 (0.00) | 1.00 (0.00) | 1.00 (0.00) |
| Temporal | Stack | Compound | 1.00 (0.00) | 1.00 (0.00) | 1.00 (0.00) |
| | Queue | Compound | 1.00 (0.00) | 1.00 (0.00) | 1.00 (0.00) |
| | LRU Cache | Cache | 1.00 (0.00) | 1.00 (0.00) | 1.00 (0.00) |
| | Priority Queue* | Compound | 0.84 (0.02) | 0.44 (0.02) | 0.28 (0.07) |
| Associative | Hashmap* | Compound | 1.00 (0.00) | 0.89 (0.02) | 0.71 (0.10) |
| | Trie | Compound | 0.97 (0.03) | 0.94 (0.05) | 0.92 (0.07) |
| | Suffix Tree | Construct | 1.00 (0.00) | 1.00 (0.00) | 0.95 (0.02) |
| | Skip List* | Compound | 0.88 (0.05) | 0.66 (0.08) | 0.51 (0.05) |
| Hierarchical | BST | Insert | 1.00 (0.00) | 1.00 (0.00) | 1.00 (0.00) |
| | | Remove | 0.98 (0.02) | 1.00 (0.00) | 0.98 (0.02) |
| | | In-Order Traversal | 1.00 (0.00) | 1.00 (0.00) | 1.00 (0.00) |
| | | Pre-Order Traversal | 0.98 (0.04) | 1.00 (0.00) | 1.00 (0.00) |
| | | Post-Order Traversal | 1.00 (0.00) | 1.00 (0.00) | 1.00 (0.00) |
| | | Depth | 1.00 (0.00) | 1.00 (0.00) | 1.00 (0.00) |
| | | Compound | 1.00 (0.00) | 1.00 (0.00) | 0.98 (0.02) |
| | Heap* | Compound | 0.51 (0.02) | 0.81 (0.08) | 0.66 (0.12) |
| | | Heapify | 0.33 (0.00) | 0.61 (0.05) | 0.76 (0.10) |
| | RB Tree* | Construct | 0.93 (0.00) | 0.90 (0.03) | 0.68 (0.13) |
| | | Compound | 0.91 (0.04) | 0.67 (0.03) | 0.49 (0.04) |
| | B$^+$ Tree* | Compound | 0.97 (0.00) | 0.99 (0.02) | 0.98 (0.02) |
| | K-D Tree* | Construct | 0.97 (0.03) | 0.90 (0.00) | 0.67 (0.06) |
| | K-D Heap* | Compound | 0.23 (0.00) | 0.08 (0.02) | 0.00 (0.00) |
| Network | Graph | Breadth-First Traversal | 0.92 (0.02) | 0.98 (0.02) | 0.94 (0.02) |
| | | Depth-First Traversal | 0.93 (0.03) | 1.00 (0.00) | 0.96 (0.02) |
| | DSU | Compound | 1.00 (0.00) | 1.00 (0.00) | 1.00 (0.00) |
| | Geom Graph* | Construct | 0.21 (0.10) | 0.08 (0.04) | 0.19 (0.13) |
| Hybrid | Bloom Filter* | Compound | 1.00 (0.00) | 0.84 (0.07) | 0.47 (0.00) |
| | DAWG* | Compound | 1.00 (0.00) | 0.03 (0.03) | 0.00 (0.00) |

## D.2. Performance of o4-mini

*Table 8.* Mean (± std) accuracy of **o4-mini** on all `DSR-Bench` tasks over three runs. Data structures marked with * are included in the `DSR-Bench-challenge` subset.

| Category | Data Structure | Operation | Short | Medium | Long |
|---|---|---|---|---|---|
| Linear | Array | Access | 1.00 (0.00) | 1.00 (0.00) | 1.00 (0.00) |
| | | Delete | 1.00 (0.00) | 1.00 (0.00) | 1.00 (0.00) |
| | | Insert | 1.00 (0.00) | 1.00 (0.00) | 1.00 (0.00) |
| | | Reverse | 1.00 (0.00) | 1.00 (0.00) | 1.00 (0.00) |
| | | Search | 1.00 (0.00) | 1.00 (0.00) | 1.00 (0.00) |
| Temporal | Stack | Compound | 1.00 (0.00) | 1.00 (0.00) | 1.00 (0.00) |
| | Queue | Compound | 1.00 (0.00) | 1.00 (0.00) | 1.00 (0.00) |
| | LRU Cache | Cache | 1.00 (0.00) | 1.00 (0.00) | 1.00 (0.00) |
| | Priority Queue* | Compound | 0.89 (0.02) | 0.47 (0.06) | 0.30 (0.03) |
| Associative | Hashmap* | Compound | 0.89 (0.04) | 0.37 (0.00) | 0.26 (0.08) |
| | Trie | Compound | 0.99 (0.02) | 0.73 (0.06) | 0.32 (0.05) |
| | Suffix Tree | Construct | 0.96 (0.02) | 0.87 (0.03) | 0.37 (0.07) |
| | Skip List* | Compound | 0.84 (0.02) | 0.60 (0.03) | 0.41 (0.02) |
| Hierarchical | BST | Insert | 1.00 (0.00) | 1.00 (0.00) | 0.99 (0.02) |
| | | Remove | 1.00 (0.00) | 1.00 (0.00) | 0.97 (0.03) |
| | | In-Order Traversal | 1.00 (0.00) | 1.00 (0.00) | 1.00 (0.00) |
| | | Pre-Order Traversal | 1.00 (0.00) | 1.00 (0.00) | 1.00 (0.00) |
| | | Post-Order Traversal | 1.00 (0.00) | 1.00 (0.00) | 1.00 (0.00) |
| | | Depth | 1.00 (0.00) | 0.99 (0.02) | 1.00 (0.00) |
| | | Compound | 1.00 (0.00) | 1.00 (0.00) | 0.98 (0.02) |
| | Heap* | Compound | 0.77 (0.06) | 0.86 (0.07) | 0.74 (0.05) |
| | | Heapify | 0.44 (0.05) | 0.58 (0.04) | 0.67 (0.03) |
| | RB Tree* | Construct | 0.90 (0.03) | 0.32 (0.13) | 0.05 (0.02) |
| | | Compound | 0.97 (0.00) | 0.64 (0.04) | 0.26 (0.02) |
| | $B^+$ Tree* | Compound | 0.99 (0.02) | 0.98 (0.04) | 0.94 (0.02) |
| | K-D Tree* | Construct | 0.59 (0.07) | 0.43 (0.06) | 0.38 (0.10) |
| | K-D Heap* | Compound | 0.22 (0.02) | 0.09 (0.02) | 0.00 (0.00) |
| Network | Graph | Breadth-First Traversal | 0.99 (0.02) | 0.97 (0.03) | 0.72 (0.14) |
| | | Depth-First Traversal | 1.00 (0.00) | 0.88 (0.02) | 0.64 (0.10) |
| | DSU | Compound | 1.00 (0.00) | 1.00 (0.00) | 0.94 (0.04) |
| | Geom Graph* | Construct | 0.83 (0.07) | 0.13 (0.00) | 0.01 (0.02) |
| Hybrid | Bloom Filter* | Compound | 1.00 (0.00) | 0.77 (0.09) | 0.07 (0.00) |
| | DAWG* | Compound | 0.49 (0.10) | 0.20 (0.09) | 0.06 (0.05) |

## D.3. Performance of Gemini-2.5-Pro

*Table 9.* Mean (± std) accuracy of **Gemini-2.5-Pro** on all `DSR-Bench` tasks over three runs. Data structures marked with * are included in the `DSR-Bench-challenge` subset.

| Category | Data Structure | Operation | Short | Medium | Long |
|---|---|---|---|---|---|
| Linear | Array | Access | 1.00 (0.00) | 1.00 (0.00) | 1.00 (0.00) |
| | | Delete | 1.00 (0.00) | 1.00 (0.00) | 1.00 (0.00) |
| | | Insert | 1.00 (0.00) | 1.00 (0.00) | 1.00 (0.00) |
| | | Reverse | 1.00 (0.00) | 0.98 (0.02) | 0.99 (0.02) |
| | | Search | 1.00 (0.00) | 1.00 (0.00) | 1.00 (0.00) |
| Temporal | Stack | Compound | 1.00 (0.00) | 1.00 (0.00) | 1.00 (0.00) |
| | Queue | Compound | 1.00 (0.00) | 1.00 (0.00) | 1.00 (0.00) |
| | LRU Cache | Cache | 1.00 (0.00) | 1.00 (0.00) | 1.00 (0.00) |
| | Priority Queue* | Compound | 0.89 (0.02) | 0.41 (0.02) | 0.23 (0.05) |
| Associative | Hashmap* | Compound | 0.58 (0.02) | 0.16 (0.07) | 0.11 (0.04) |
| | Trie | Compound | 0.93 (0.03) | 0.56 (0.02) | 0.37 (0.03) |
| | Suffix Tree | Construct | 0.96 (0.04) | 0.89 (0.02) | 0.86 (0.08) |
| | Skip List* | Compound | 0.94 (0.02) | 0.87 (0.06) | 0.53 (0.03) |
| Hierarchical | BST | Insert | 1.00 (0.00) | 0.94 (0.02) | 0.94 (0.05) |
| | | Remove | 0.87 (0.06) | 0.86 (0.07) | 0.89 (0.02) |
| | | In-Order Traversal | 1.00 (0.00) | 1.00 (0.00) | 1.00 (0.00) |
| | | Pre-Order Traversal | 1.00 (0.00) | 1.00 (0.00) | 1.00 (0.00) |
| | | Post-Order Traversal | 1.00 (0.00) | 0.99 (0.02) | 0.94 (0.04) |
| | | Depth | 1.00 (0.00) | 1.00 (0.00) | 1.00 (0.00) |
| | | Compound | 0.99 (0.02) | 0.99 (0.02) | 0.98 (0.02) |
| | Heap* | Compound | 0.78 (0.04) | 0.49 (0.10) | 0.53 (0.12) |
| | | Heapify | 0.36 (0.02) | 0.06 (0.05) | 0.04 (0.04) |
| | RB Tree* | Construct | 0.91 (0.07) | 0.53 (0.09) | 0.03 (0.03) |
| | | Compound | 0.91 (0.02) | 0.41 (0.07) | 0.13 (0.03) |
| | $B^+$ Tree* | Compound | 1.00 (0.00) | 0.97 (0.03) | 0.94 (0.06) |
| | K-D Tree* | Construct | 0.96 (0.02) | 0.64 (0.13) | 0.16 (0.02) |
| | K-D Heap* | Compound | 0.23 (0.00) | 0.06 (0.02) | 0.00 (0.00) |
| Network | Graph | Breadth-First Traversal | 1.00 (0.00) | 1.00 (0.00) | 0.74 (0.02) |
| | | Depth-First Traversal | 1.00 (0.00) | 0.81 (0.05) | 0.14 (0.02) |
| | DSU | Compound | 1.00 (0.00) | 0.99 (0.02) | 0.90 (0.05) |
| | Geom Graph* | Construct | 0.19 (0.02) | 0.17 (0.06) | 0.02 (0.02) |
| Hybrid | Bloom Filter* | Compound | 0.94 (0.02) | 0.97 (0.00) | 0.66 (0.02) |
| | DAWG* | Compound | 0.61 (0.05) | 0.08 (0.07) | 0.01 (0.02) |

## D.4. Performance of Claude-3.7-Sonnet

*Table 10.* Mean (± std) accuracy of **Claude-3.7-Sonnet** on all `DSR-Bench` tasks over three runs. Data structures marked with * are included in the `DSR-Bench-challenge` subset.

| Category | Data Structure | Operation | Short | Medium | Long |
|---|---|---|---|---|---|
| Linear | Array | Access | 1.00 (0.00) | 1.00 (0.00) | 1.00 (0.00) |
| | | Delete | 1.00 (0.00) | 1.00 (0.00) | 1.00 (0.00) |
| | | Insert | 1.00 (0.00) | 1.00 (0.00) | 0.96 (0.02) |
| | | Reverse | 1.00 (0.00) | 0.98 (0.02) | 0.97 (0.00) |
| | | Search | 1.00 (0.00) | 1.00 (0.00) | 1.00 (0.00) |
| Temporal | Stack | Compound | 1.00 (0.00) | 1.00 (0.00) | 1.00 (0.00) |
| | Queue | Compound | 1.00 (0.00) | 0.93 (0.00) | 0.98 (0.02) |
| | LRU Cache | Compound | 1.00 (0.00) | 1.00 (0.00) | 0.98 (0.02) |
| | Priority Queue* | Compound | 0.70 (0.06) | 0.11 (0.02) | 0.04 (0.02) |
| Associative | Hashmap* | Compound | 0.71 (0.02) | 0.16 (0.05) | 0.04 (0.05) |
| | Trie | Compound | 0.94 (0.02) | 0.64 (0.07) | 0.31 (0.10) |
| | Suffix Tree | Construct | 0.23 (0.00) | 0.00 (0.00) | 0.00 (0.00) |
| | Skip List* | Compound | 0.87 (0.06) | 0.76 (0.05) | 0.61 (0.11) |
| Hierarchical | BST | Insert | 0.96 (0.04) | 0.98 (0.04) | 0.79 (0.02) |
| | | Remove | 0.94 (0.02) | 0.91 (0.02) | 0.92 (0.02) |
| | | In-Order Traversal | 1.00 (0.00) | 1.00 (0.00) | 1.00 (0.00) |
| | | Pre-Order Traversal | 1.00 (0.00) | 0.99 (0.02) | 1.00 (0.00) |
| | | Post-Order Traversal | 1.00 (0.00) | 0.74 (0.02) | 0.93 (0.03) |
| | | Depth | 1.00 (0.00) | 1.00 (0.00) | 0.98 (0.02) |
| | | Compound | 0.90 (0.03) | 0.21 (0.02) | 0.24 (0.05) |
| | Heap* | Compound | 0.70 (0.03) | 0.32 (0.05) | 0.13 (0.03) |
| | | Heapify | 0.89 (0.02) | 0.62 (0.08) | 0.26 (0.02) |
| | RB Tree* | Construct | 0.19 (0.05) | 0.00 (0.00) | 0.00 (0.00) |
| | | Compound | 0.57 (0.03) | 0.03 (0.00) | 0.00 (0.00) |
| | $B^+$ Tree* | Compound | 0.80 (0.00) | 0.18 (0.02) | 0.23 (0.03) |
| | K-D Tree* | Construct | 0.00 (0.00) | 0.00 (0.00) | 0.00 (0.00) |
| | K-D Heap* | Compound | 0.11 (0.02) | 0.03 (0.00) | 0.00 (0.00) |
| Network | Graph | Breadth-First Traversal | 0.40 (0.03) | 0.08 (0.02) | 0.01 (0.02) |
| | | Depth-First Traversal | 0.50 (0.03) | 0.11 (0.02) | 0.00 (0.00) |
| | DSU | Compound | 0.04 (0.02) | 0.12 (0.04) | 0.00 (0.00) |
| | Geom Graph* | Construct | 0.04 (0.05) | 0.00 (0.00) | 0.00 (0.00) |
| Hybrid | Bloom Filter* | Compound | 0.44 (0.04) | 0.03 (0.00) | 0.00 (0.00) |
| | DAWG* | Compound | 0.17 (0.00) | 0.00 (0.00) | 0.00 (0.00) |

## D.5. Performance of DeepSeek-R1

*Table 11.* Mean (± std) accuracy of **DeepSeek-R1** on all `DSR-Bench` tasks over three runs. Data structures marked with * are included in the `DSR-Bench-challenge` subset.

| Category | Data Structure | Operation | Short | Medium | Long |
|---|---|---|---|---|---|
| Linear | Array | Access | 1.00 (0.00) | 0.98 (0.02) | 1.00 (0.00) |
| | | Delete | 0.99 (0.02) | 1.00 (0.00) | 0.98 (0.02) |
| | | Insert | 0.98 (0.02) | 0.99 (0.02) | 0.99 (0.02) |
| | | Reverse | 1.00 (0.00) | 1.00 (0.00) | 1.00 (0.00) |
| | | Search | 1.00 (0.00) | 1.00 (0.00) | 1.00 (0.00) |
| Temporal | Stack | Compound | 1.00 (0.00) | 1.00 (0.00) | 0.94 (0.07) |
| | Queue | Compound | 1.00 (0.00) | 1.00 (0.00) | 0.97 (0.00) |
| | LRU Cache | Compound | 1.00 (0.00) | 1.00 (0.00) | 0.99 (0.01) |
| | Priority Queue* | Compound | 0.92 (0.02) | 0.54 (0.07) | 0.48 (0.05) |
| Associative | Hashmap* | Compound | 0.44 (0.05) | 0.01 (0.02) | 0.03 (0.03) |
| | Trie | Compound | 0.54 (0.24) | 0.32 (0.04) | 0.12 (0.06) |
| | Suffix Tree | Construct | 0.93 (0.07) | 0.50 (0.07) | 0.05 (0.05) |
| | Skip List* | Compound | 0.89 (0.04) | 0.63 (0.03) | 0.54 (0.02) |
| Hierarchical | BST | Insert | 1.00 (0.00) | 0.98 (0.02) | 0.90 (0.03) |
| | | Remove | 0.98 (0.02) | 0.93 (0.03) | 0.88 (0.05) |
| | | In-Order Traversal | 1.00 (0.00) | 0.99 (0.02) | 1.00 (0.00) |
| | | Pre-Order Traversal | 1.00 (0.00) | 1.00 (0.00) | 1.00 (0.00) |
| | | Post-Order Traversal | 1.00 (0.00) | 1.00 (0.00) | 0.97 (0.00) |
| | | Depth | 1.00 (0.00) | 1.00 (0.00) | 1.00 (0.00) |
| | | Compound | 0.97 (0.03) | 0.84 (0.08) | 0.65 (0.07) |
| | Heap* | Compound | 0.49 (0.04) | 0.23 (0.07) | 0.21 (0.05) |
| | | Heapify | 0.34 (0.02) | 0.16 (0.08) | 0.08 (0.06) |
| | RB Tree* | Construct | 0.88 (0.02) | 0.10 (0.06) | 0.00 (0.00) |
| | | Compound | 0.91 (0.04) | 0.37 (0.10) | 0.03 (0.03) |
| | $B^+$ Tree* | Compound | 0.81 (0.02) | 0.88 (0.04) | 0.70 (0.06) |
| | K-D Tree* | Construct | 1.00 (0.00) | 0.34 (0.05) | 0.01 (0.02) |
| | K-D Heap* | Compound | 0.23 (0.00) | 0.08 (0.02) | 0.01 (0.02) |
| Network | Graph | Breadth-First Traversal | 0.92 (0.02) | 0.90 (0.06) | 0.46 (0.05) |
| | | Depth-First Traversal | 0.80 (0.09) | 0.58 (0.04) | 0.22 (0.02) |
| | DSU | Compound | 0.64 (0.56) | 0.92 (0.04) | 0.83 (0.07) |
| | Geom Graph* | Construct | 0.99 (0.02) | 0.00 (0.00) | 0.00 (0.00) |
| Hybrid | Bloom Filter* | Compound | 0.99 (0.02) | 0.92 (0.02) | 0.31 (0.02) |
| | DAWG* | Compound | 0.40 (0.12) | 0.02 (0.02) | 0.00 (0.00) |

## D.6. Performance of GPT-4.1

*Table 12.* Mean (± std) accuracy of **GPT-4.1** on all `DSR-Bench` tasks over three runs. Data structures marked with * are included in the `DSR-Bench-challenge` subset.

| Category | Data Structure | Operation | Short | Medium | Long |
|---|---|---|---|---|---|
| Linear | Array | Access | 1.00 (0.00) | 1.00 (0.00) | 1.00 (0.00) |
| | | Delete | 1.00 (0.00) | 1.00 (0.00) | 1.00 (0.00) |
| | | Insert | 0.91 (0.08) | 0.79 (0.02) | 0.54 (0.02) |
| | | Reverse | 0.98 (0.02) | 0.97 (0.00) | 0.86 (0.02) |
| | | Search | 1.00 (0.00) | 1.00 (0.00) | 1.00 (0.00) |
| Temporal | Stack | Compound | 0.97 (0.00) | 0.49 (0.02) | 0.18 (0.04) |
| | Queue | Compound | 0.82 (0.04) | 0.59 (0.04) | 0.19 (0.07) |
| | LRU Cache | Cache | 0.94 (0.02) | 0.80 (0.00) | 0.81 (0.02) |
| | Priority Queue* | Compound | 0.63 (0.03) | 0.10 (0.00) | 0.03 (0.00) |
| Associative | Hashmap* | Compound | 0.19 (0.07) | 0.00 (0.00) | 0.00 (0.00) |
| | Trie | Compound | 0.39 (0.07) | 0.13 (0.03) | 0.01 (0.02) |
| | Suffix Tree | Construct | 0.00 (0.00) | 0.00 (0.00) | 0.00 (0.00) |
| | Skip List* | Compound | 0.21 (0.02) | 0.00 (0.00) | 0.00 (0.00) |
| Hierarchical | BST | Insert | 0.79 (0.04) | 0.50 (0.03) | 0.14 (0.02) |
| | | Remove | 0.78 (0.04) | 0.58 (0.02) | 0.36 (0.04) |
| | | In-Order Traversal | 1.00 (0.00) | 1.00 (0.00) | 0.94 (0.02) |
| | | Pre-Order Traversal | 1.00 (0.00) | 0.97 (0.00) | 0.98 (0.02) |
| | | Post-Order Traversal | 0.82 (0.02) | 0.51 (0.04) | 0.23 (0.06) |
| | | Depth | 0.30 (0.04) | 0.07 (0.06) | 0.03 (0.03) |
| | | Compound | 0.69 (0.02) | 0.26 (0.02) | 0.03 (0.00) |
| | Heap* | Compound | 0.58 (0.02) | 0.01 (0.02) | 0.00 (0.00) |
| | | Heapify | 0.57 (0.03) | 0.04 (0.02) | 0.00 (0.00) |
| | RB Tree* | Construct | 0.12 (0.02) | 0.00 (0.00) | 0.00 (0.00) |
| | | Compound | 0.31 (0.04) | 0.02 (0.02) | 0.00 (0.00) |
| | $B^+$ Tree* | Compound | 0.27 (0.00) | 0.30 (0.00) | 0.13 (0.00) |
| | K-D Tree* | Construct | 0.00 (0.00) | 0.00 (0.00) | 0.00 (0.00) |
| | K-D Heap* | Compound | 0.10 (0.00) | 0.03 (0.00) | 0.00 (0.00) |
| Network | Graph | Breadth-First Traversal | 0.31 (0.05) | 0.09 (0.02) | 0.00 (0.00) |
| | | Depth-First Traversal | 0.50 (0.03) | 0.00 (0.00) | 0.00 (0.00) |
| | DSU | Compound | 0.06 (0.02) | 0.00 (0.00) | 0.00 (0.00) |
| | Geom Graph* | Construct | 0.03 (0.00) | 0.00 (0.00) | 0.00 (0.00) |
| Hybrid | Bloom Filter* | Compound | 0.10 (0.00) | 0.03 (0.00) | 0.00 (0.00) |
| | DAWG* | Compound | 0.16 (0.02) | 0.00 (0.00) | 0.00 (0.00) |

## D.7. Performance of Gemini-2.0-Flash

*Table 13.* Mean (± std) accuracy of **Gemini-2.0-Flash** on all `DSR-Bench` tasks over three runs. Data structures marked with * are included in the `DSR-Bench-challenge` subset.

| Category | Data Structure | Operation | Short | Medium | Long |
|---|---|---|---|---|---|
| Linear | Array | Access | 1.00 (0.00) | 1.00 (0.00) | 1.00 (0.00) |
| | | Delete | 0.96 (0.02) | 0.87 (0.03) | 0.77 (0.03) |
| | | Insert | 0.99 (0.02) | 0.96 (0.02) | 1.00 (0.00) |
| | | Reverse | 0.96 (0.02) | 0.78 (0.02) | 0.64 (0.04) |
| | | Search | 1.00 (0.00) | 1.00 (0.00) | 0.97 (0.00) |
| Temporal | Stack | Compound | 0.67 (0.00) | 0.37 (0.00) | 0.03 (0.00) |
| | Queue | Compound | 0.87 (0.00) | 0.33 (0.00) | 0.10 (0.00) |
| | LRU Cache | Cache | 0.93 (0.00) | 0.86 (0.02) | 0.56 (0.02) |
| | Priority Queue* | Compound | 0.38 (0.02) | 0.10 (0.00) | 0.01 (0.02) |
| Associative | Hashmap* | Compound | 0.28 (0.05) | 0.01 (0.02) | 0.00 (0.00) |
| | Trie | Compound | 0.31 (0.02) | 0.18 (0.02) | 0.03 (0.00) |
| | Suffix Tree | Construct | 0.00 (0.00) | 0.02 (0.02) | 0.00 (0.00) |
| | Skip List* | Compound | 0.16 (0.02) | 0.00 (0.00) | 0.03 (0.00) |
| Hierarchical | BST | Insert | 0.31 (0.02) | 0.27 (0.03) | 0.06 (0.04) |
| | | Remove | 0.63 (0.09) | 0.33 (0.03) | 0.13 (0.03) |
| | | In-Order Traversal | 0.87 (0.00) | 0.66 (0.02) | 0.71 (0.04) |
| | | Pre-Order Traversal | 1.00 (0.00) | 1.00 (0.00) | 0.93 (0.00) |
| | | Post-Order Traversal | 0.63 (0.00) | 0.17 (0.00) | 0.10 (0.00) |
| | | Depth | 0.13 (0.09) | 0.03 (0.00) | 0.00 (0.00) |
| | | Compound | 0.51 (0.05) | 0.12 (0.04) | 0.10 (0.00) |
| | Heap* | Compound | 0.32 (0.05) | 0.03 (0.00) | 0.02 (0.02) |
| | | Heapify | 0.23 (0.06) | 0.00 (0.00) | 0.00 (0.00) |
| | RB Tree* | Construct | 0.08 (0.02) | 0.00 (0.00) | 0.00 (0.00) |
| | | Compound | 0.43 (0.03) | 0.07 (0.00) | 0.00 (0.00) |
| | $B^+$ Tree* | Compound | 0.17 (0.00) | 0.13 (0.03) | 0.06 (0.05) |
| | K-D Tree* | Construct | 0.02 (0.02) | 0.00 (0.00) | 0.00 (0.00) |
| | K-D Heap* | Compound | 0.10 (0.00) | 0.03 (0.00) | 0.00 (0.00) |
| Network | Graph | Breadth-First Traversal | 0.10 (0.00) | 0.03 (0.00) | 0.00 (0.00) |
| | | Depth-First Traversal | 0.19 (0.02) | 0.00 (0.00) | 0.00 (0.00) |
| | DSU | Compound | 0.01 (0.02) | 0.00 (0.00) | 0.00 (0.00) |
| | Geom Graph* | Construct | 0.07 (0.03) | 0.00 (0.00) | 0.00 (0.00) |
| Hybrid | Bloom Filter* | Compound | 0.10 (0.00) | 0.03 (0.00) | 0.00 (0.00) |
| | DAWG* | Compound | 0.18 (0.02) | 0.00 (0.00) | 0.00 (0.00) |

## D.8. Performance of Claude-3.5-Sonnet

*Table 14.* Mean (± std) accuracy of **Claude-3.5-Sonnet** on all `DSR-Bench` tasks over three runs. Data structures marked with * are included in the `DSR-Bench-challenge` subset.

| Category | Data Structure | Operation | Short | Medium | Long |
|---|---|---|---|---|---|
| Linear | Array | Access | 1.00 (0.00) | 1.00 (0.00) | 1.00 (0.00) |
| | | Delete | 1.00 (0.00) | 1.00 (0.00) | 0.93 (0.00) |
| | | Insert | 1.00 (0.00) | 0.90 (0.00) | 0.90 (0.00) |
| | | Reverse | 1.00 (0.00) | 0.88 (0.02) | 0.72 (0.05) |
| | | Search | 1.00 (0.00) | 1.00 (0.00) | 1.00 (0.00) |
| Temporal | Stack | Compound | 1.00 (0.00) | 1.00 (0.00) | 0.98 (0.04) |
| | Queue | Compound | 0.87 (0.00) | 0.67 (0.03) | 0.79 (0.02) |
| | LRU Cache | Cache | 0.99 (0.02) | 0.90 (0.07) | 0.58 (0.13) |
| | Priority Queue* | Compound | 0.63 (0.00) | 0.27 (0.03) | 0.09 (0.02) |
| Associative | Hashmap | Compound | 0.37 (0.03) | 0.10 (0.00) | 0.00 (0.00) |
| | Trie* | Compound | 0.89 (0.04) | 0.50 (0.03) | 0.07 (0.00) |
| | Suffix Tree* | Construct | 0.21 (0.02) | 0.03 (0.00) | 0.00 (0.00) |
| | Skip List* | Compound | 0.77 (0.06) | 0.30 (0.07) | 0.20 (0.07) |
| Hierarchical | BST | Insert | 0.80 (0.06) | 0.50 (0.06) | 0.51 (0.05) |
| | | Remove | 0.96 (0.04) | 0.87 (0.00) | 0.77 (0.00) |
| | | In-Order Traversal | 0.97 (0.03) | 0.94 (0.02) | 0.94 (0.02) |
| | | Pre-Order Traversal | 1.00 (0.00) | 1.00 (0.00) | 0.99 (0.02) |
| | | Post-Order Traversal | 1.00 (0.00) | 0.69 (0.08) | 0.54 (0.02) |
| | | Depth | 1.00 (0.00) | 0.96 (0.02) | 0.78 (0.02) |
| | | Compound | 0.77 (0.07) | 0.28 (0.05) | 0.09 (0.02) |
| | Heap* | Compound | 0.78 (0.04) | 0.13 (0.00) | 0.11 (0.02) |
| | | Heapify | 0.53 (0.12) | 0.08 (0.04) | 0.00 (0.00) |
| | RB Tree* | Construct | 0.13 (0.00) | 0.00 (0.00) | 0.00 (0.00) |
| | | Compound | 0.44 (0.02) | 0.03 (0.00) | 0.00 (0.00) |
| | $B^+$ Tree* | Compound | 0.40 (0.00) | 0.28 (0.08) | 0.02 (0.02) |
| | K-D Tree* | Construct | 0.09 (0.02) | 0.00 (0.00) | 0.00 (0.00) |
| | K-D Heap* | Compound | 0.13 (0.00) | 0.02 (0.02) | 0.00 (0.00) |
| Network | Graph | Breadth-First Traversal | 0.17 (0.03) | 0.02 (0.02) | 0.00 (0.00) |
| | | Depth-First Traversal | 0.13 (0.03) | 0.02 (0.02) | 0.00 (0.00) |
| | DSU* | Compound | 0.07 (0.03) | 0.00 (0.00) | 0.00 (0.00) |
| | Geom Graph* | Construct | 0.10 (0.00) | 0.00 (0.00) | 0.00 (0.00) |
| Hybrid | Bloom Filter* | Compound | 0.10 (0.00) | 0.03 (0.00) | 0.00 (0.00) |
| | DAWG* | Compound | 0.20 (0.00) | 0.00 (0.00) | 0.00 (0.00) |

## D.9. Performance of DeepSeek-V3

*Table 15.* Mean (± std) accuracy of **DeepSeek-V3** on all `DSR-Bench` tasks over three runs. Data structures marked with * are included in the `DSR-Bench-challenge` subset.

| Category | Data Structure | Operation | Short | Medium | Long |
|---|---|---|---|---|---|
| Linear | Array | Access | 1.00 (0.00) | 0.97 (0.00) | 0.97 (0.00) |
| | | Delete | 1.00 (0.00) | 1.00 (0.00) | 1.00 (0.00) |
| | | Insert | 1.00 (0.00) | 1.00 (0.00) | 1.00 (0.00) |
| | | Reverse | 1.00 (0.00) | 0.92 (0.02) | 0.92 (0.02) |
| | | Search | 1.00 (0.00) | 0.97 (0.00) | 0.93 (0.00) |
| Temporal | Stack | Compound | 0.70 (0.03) | 0.49 (0.02) | 0.04 (0.02) |
| | Queue | Compound | 0.84 (0.02) | 0.38 (0.02) | 0.07 (0.03) |
| | LRU Cache | Compound | 0.94 (0.02) | 0.77 (0.06) | 0.76 (0.02) |
| | Priority Queue* | Compound | 0.53 (0.03) | 0.06 (0.04) | 0.00 (0.00) |
| Associative | Hashmap* | Compound | 0.04 (0.02) | 0.00 (0.00) | 0.00 (0.00) |
| | Trie | Compound | 0.00 (0.00) | 0.00 (0.00) | 0.00 (0.00) |
| | Suffix Tree | Construct | 0.06 (0.02) | 0.00 (0.00) | 0.00 (0.00) |
| | Skip List* | Compound | 0.06 (0.02) | 0.00 (0.00) | 0.00 (0.00) |
| Hierarchical | BST | Insert | 0.93 (0.03) | 0.62 (0.02) | 0.46 (0.05) |
| | | Remove | 0.84 (0.04) | 0.80 (0.03) | 0.66 (0.02) |
| | | In-Order Traversal | 0.97 (0.00) | 1.00 (0.00) | 1.00 (0.00) |
| | | Pre-Order Traversal | 1.00 (0.00) | 1.00 (0.00) | 1.00 (0.00) |
| | | Post-Order Traversal | 0.82 (0.02) | 0.53 (0.03) | 0.20 (0.03) |
| | | Depth | 0.67 (0.03) | 0.24 (0.02) | 0.07 (0.03) |
| | | Compound | 0.68 (0.02) | 0.12 (0.04) | 0.06 (0.02) |
| | Heap* | Compound | 0.23 (0.00) | 0.00 (0.00) | 0.00 (0.00) |
| | | Heapify | 0.59 (0.02) | 0.06 (0.02) | 0.00 (0.00) |
| | RB Tree* | Construct | 0.09 (0.02) | 0.00 (0.00) | 0.00 (0.00) |
| | | Compound | 0.30 (0.03) | 0.00 (0.00) | 0.00 (0.00) |
| | $B^+$ Tree* | Compound | 0.14 (0.05) | 0.10 (0.03) | 0.00 (0.00) |
| | K-D Tree* | Construct | 0.00 (0.00) | 0.00 (0.00) | 0.00 (0.00) |
| | K-D Heap* | Compound | 0.07 (0.00) | 0.03 (0.00) | 0.00 (0.00) |
| Network | Graph | Breadth-First Traversal | 0.29 (0.05) | 0.04 (0.02) | 0.00 (0.00) |
| | | Depth-First Traversal | 0.22 (0.02) | 0.03 (0.00) | 0.00 (0.00) |
| | DSU | Compound | 0.03 (0.00) | 0.00 (0.00) | 0.00 (0.00) |
| | Geom Graph* | Construct | 0.06 (0.02) | 0.00 (0.00) | 0.00 (0.00) |
| Hybrid | Bloom Filter* | Compound | 0.10 (0.00) | 0.03 (0.00) | 0.00 (0.00) |
| | DAWG* | Compound | 0.17 (0.00) | 0.00 (0.00) | 0.00 (0.00) |

## D.10. Performance of Llama-3.3

*Table 16.* Mean (± std) accuracy of **Llama-3.3** on all `DSR-Bench` tasks over three runs. Data structures marked with * are included in the `DSR-Bench-challenge` subset.

| Category | Data Structure | Operation | Short | Medium | Long |
|---|---|---|---|---|---|
| Linear | Array | Access | 1.00 (0.00) | 0.56 (0.04) | 0.38 (0.02) |
| | | Delete | 0.81 (0.08) | 0.68 (0.04) | 0.44 (0.05) |
| | | Insert | 0.76 (0.04) | 0.78 (0.02) | 0.29 (0.07) |
| | | Reverse | 0.91 (0.02) | 0.56 (0.02) | 0.34 (0.07) |
| | | Search | 0.97 (0.00) | 0.90 (0.00) | 0.93 (0.00) |
| Temporal | Stack | Compound | 0.09 (0.02) | 0.04 (0.05) | 0.00 (0.00) |
| | Queue | Compound | 0.58 (0.11) | 0.12 (0.02) | 0.06 (0.02) |
| | LRU Cache | Compound | 0.74 (0.08) | 0.44 (0.11) | 0.31 (0.11) |
| | Priority Queue* | Compound | 0.21 (0.05) | 0.01 (0.02) | 0.02 (0.02) |
| Associative | Hashmap* | Compound | 0.00 (0.00) | 0.00 (0.00) | 0.00 (0.00) |
| | Trie | Compound | 0.01 (0.02) | 0.00 (0.00) | 0.00 (0.00) |
| | Suffix Tree | Construct | 0.00 (0.00) | 0.00 (0.00) | 0.00 (0.00) |
| | Skip List* | Compound | 0.03 (0.00) | 0.00 (0.00) | 0.00 (0.00) |
| Hierarchical | BST | Insert | 0.37 (0.00) | 0.14 (0.02) | 0.03 (0.00) |
| | | Remove | 0.49 (0.04) | 0.30 (0.03) | 0.14 (0.04) |
| | | In-Order Traversal | 0.60 (0.06) | 0.61 (0.08) | 0.61 (0.04) |
| | | Pre-Order Traversal | 0.86 (0.11) | 0.81 (0.08) | 0.78 (0.11) |
| | | Post-Order Traversal | 0.31 (0.04) | 0.04 (0.02) | 0.00 (0.00) |
| | | Depth | 0.70 (0.00) | 0.38 (0.07) | 0.13 (0.06) |
| | | Compound | 0.26 (0.02) | 0.01 (0.02) | 0.00 (0.00) |
| | Heap* | Compound | 0.17 (0.03) | 0.03 (0.00) | 0.00 (0.00) |
| | | Heapify | 0.24 (0.05) | 0.00 (0.00) | 0.00 (0.00) |
| | RB Tree* | Construct | 0.00 (0.00) | 0.00 (0.00) | 0.00 (0.00) |
| | | Compound | 0.31 (0.02) | 0.00 (0.00) | 0.00 (0.00) |
| | $B^+$ Tree* | Compound | 0.02 (0.04) | 0.00 (0.00) | 0.00 (0.00) |
| | K-D Tree* | Construct | 0.00 (0.00) | 0.00 (0.00) | 0.00 (0.00) |
| | K-D Heap* | Compound | 0.02 (0.02) | 0.02 (0.02) | 0.00 (0.00) |
| Network | Graph | Breadth-First Traversal | 0.07 (0.06) | 0.00 (0.00) | 0.00 (0.00) |
| | | Depth-First Traversal | 0.04 (0.05) | 0.01 (0.02) | 0.00 (0.00) |
| | DSU | Compound | 0.00 (0.00) | 0.00 (0.00) | 0.00 (0.00) |
| | Geom Graph* | Construct | 0.07 (0.03) | 0.00 (0.00) | 0.00 (0.00) |
| Hybrid | Bloom Filter* | Compound | 0.00 (0.00) | 0.07 (0.00) | 0.00 (0.00) |
| | DAWG* | Compound | 0.02 (0.02) | 0.00 (0.00) | 0.00 (0.00) |

## D.11. Performance of Qwen3-8b-Instruct

*Table 17.* Mean (± std) accuracy of **Qwen3-8b-Instruct** on all `DSR-Bench` tasks over three runs. Data structures marked with * are included in the `DSR-Bench-challenge` subset.

| Category | Data Structure | Operation | Short | Medium | Long |
|---|---|---|---|---|---|
| Linear | Array | Access | 1.00 (0.00) | 1.00 (0.00) | 1.00 (0.00) |
| | | Delete | 0.99 (0.02) | 0.99 (0.02) | 0.96 (0.04) |
| | | Insert | 1.00 (0.00) | 1.00 (0.00) | 0.97 (0.03) |
| | | Reverse | 1.00 (0.00) | 1.00 (0.00) | 0.92 (0.02) |
| | | Search | 1.00 (0.00) | 1.00 (0.00) | 1.00 (0.00) |
| Temporal | Stack | Compound | 1.00 (0.00) | 0.99 (0.02) | 0.93 (0.06) |
| | Queue | Compound | 1.00 (0.00) | 0.98 (0.04) | 0.98 (0.02) |
| | LRU Cache | Compound | 0.93 (0.00) | 0.77 (0.09) | 0.49 (0.08) |
| | Priority Queue* | Compound | 0.71 (0.07) | 0.28 (0.08) | 0.07 (0.03) |
| Associative | Hashmap* | Compound | 0.59 (0.08) | 0.68 (0.10) | 0.20 (0.00) |
| | Trie | Compound | 0.02 (0.02) | 0.00 (0.00) | 0.01 (0.02) |
| | Suffix Tree | Construct | 0.00 (0.00) | 0.00 (0.00) | 0.00 (0.00) |
| | Skip List* | Compound | 0.08 (0.05) | 0.00 (0.00) | 0.00 (0.00) |
| Hierarchical | BST | Insert | 0.78 (0.07) | 0.30 (0.13) | 0.01 (0.02) |
| | | Remove | 0.63 (0.06) | 0.26 (0.04) | 0.01 (0.02) |
| | | In-Order Traversal | 0.84 (0.02) | 0.57 (0.07) | 0.23 (0.03) |
| | | Pre-Order Traversal | 0.87 (0.06) | 0.80 (0.09) | 0.84 (0.02) |
| | | Post-Order Traversal | 0.86 (0.12) | 0.76 (0.07) | 0.37 (0.07) |
| | | Depth | 1.00 (0.00) | 0.97 (0.03) | 0.89 (0.04) |
| | | Compound | 0.87 (0.03) | 0.24 (0.08) | 0.01 (0.02) |
| | Heap* | Compound | 0.34 (0.07) | 0.07 (0.03) | 0.02 (0.02) |
| | | Heapify | 0.79 (0.05) | 0.24 (0.04) | 0.01 (0.02) |
| | RB Tree* | Construct | 0.00 (0.00) | 0.00 (0.00) | 0.00 (0.00) |
| | | Compound | 0.22 (0.02) | 0.00 (0.00) | 0.00 (0.00) |
| | $B^+$ Tree* | Compound | 0.23 (0.15) | 0.08 (0.05) | 0.03 (0.03) |
| | K-D Tree* | Compound | 0.01 (0.02) | 0.00 (0.00) | 0.00 (0.00) |
| | K-D Heap* | Compound | 0.12 (0.05) | 0.03 (0.00) | 0.00 (0.00) |
| Network | Graph | Depth-First Traversal | 0.63 (0.03) | 0.04 (0.02) | 0.00 (0.00) |
| | | Breadth-First Traversal | 0.46 (0.13) | 0.01 (0.02) | 0.00 (0.00) |
| | DSU | Compound | 0.29 (0.04) | 0.01 (0.02) | 0.00 (0.00) |
| | Geom Graph* | Construct | 0.00 (0.00) | 0.00 (0.00) | 0.00 (0.00) |
| Hybrid | Bloom Filter* | Compound | 0.36 (0.10) | 0.00 (0.00) | 0.00 (0.00) |
| | DAWG* | Compound | 0.01 (0.02) | 0.00 (0.00) | 0.00 (0.00) |

## D.12. Performance of Phi-4-reasoning-14B

*Table 18.* Mean (± std) accuracy of **Phi-4-reasoning-14B** on all `DSR-Bench` tasks over three runs. Data structures marked with * are included in the `DSR-Bench-challenge` subset.

| Category | Data Structure | Operation | Short | Medium | Long |
|---|---|---|---|---|---|
| Linear | Array | Access | 0.97 (0.03) | 0.77 (0.05) | 0.73 (0.09) |
| | | Delete | 0.85 (0.02) | 0.65 (0.02) | 0.53 (0.07) |
| | | Insert | 0.85 (0.02) | 0.54 (0.02) | 0.24 (0.08) |
| | | Reverse | 0.68 (0.07) | 0.08 (0.07) | 0.03 (0.03) |
| | | Search | 0.97 (0.00) | 0.83 (0.00) | 0.70 (0.03) |
| Temporal | Stack | Compound | 0.15 (0.02) | 0.00 (0.00) | 0.00 (0.00) |
| | Queue | Compound | 0.33 (0.05) | 0.02 (0.02) | 0.00 (0.00) |
| | LRU Cache | Compound | 0.43 (0.05) | 0.02 (0.02) | 0.03 (0.00) |
| | Priority Queue* | Compound | 0.02 (0.02) | 0.00 (0.00) | 0.00 (0.00) |
| Associative | Hashmap* | Compound | 0.00 (0.00) | 0.00 (0.00) | 0.00 (0.00) |
| | Trie | Compound | 0.00 (0.00) | 0.00 (0.00) | 0.00 (0.00) |
| | Suffix Tree | Construct | 0.00 (0.00) | 0.00 (0.00) | 0.00 (0.00) |
| | Skip List* | Compound | 0.02 (0.02) | 0.00 (0.00) | 0.00 (0.00) |
| Hierarchical | BST | Insert | 0.31 (0.02) | 0.01 (0.02) | 0.00 (0.00) |
| | | Remove | 0.14 (0.05) | 0.02 (0.02) | 0.00 (0.00) |
| | | In-Order Traversal | 0.50 (0.05) | 0.37 (0.00) | 0.20 (0.07) |
| | | Pre-Order Traversal | 0.68 (0.07) | 0.62 (0.07) | 0.41 (0.07) |
| | | Post-Order Traversal | 0.13 (0.14) | 0.02 (0.02) | 0.00 (0.00) |
| | | Depth | 0.28 (0.02) | 0.17 (0.09) | 0.23 (0.00) |
| | | Compound | 0.10 (0.05) | 0.00 (0.00) | 0.00 (0.00) |
| | Heap* | Compound | 0.08 (0.02) | 0.00 (0.00) | 0.00 (0.00) |
| | | Heapify | 0.10 (0.05) | 0.00 (0.00) | 0.00 (0.00) |
| | RB Tree* | Construct | 0.00 (0.00) | 0.00 (0.00) | 0.00 (0.00) |
| | | Compound | 0.07 (0.00) | 0.00 (0.00) | 0.00 (0.00) |
| | $B^+$ Tree* | Compound | 0.07 (0.00) | 0.00 (0.00) | 0.00 (0.00) |
| | K-D Tree* | Compound | 0.00 (0.00) | 0.00 (0.00) | 0.00 (0.00) |
| | K-D Heap* | Compound | 0.03 (0.00) | 0.00 (0.00) | 0.00 (0.00) |
| Network | Graph | Depth-First Traversal | 0.07 (0.00) | 0.01 (0.02) | 0.00 (0.00) |
| | | Breadth-First Traversal | 0.07 (0.00) | 0.00 (0.00) | 0.00 (0.00) |
| | DSU | Compound | 0.00 (0.00) | 0.00 (0.00) | 0.00 (0.00) |
| | Geom Graph* | Construct | 0.00 (0.00) | 0.00 (0.00) | 0.00 (0.00) |
| Hybrid | Bloom Filter* | Compound | 0.02 (0.02) | 0.00 (0.00) | 0.00 (0.00) |
| | DAWG* | Compound | 0.00 (0.00) | 0.00 (0.00) | 0.00 (0.00) |

## D.13. Performance of Mixtral-8x7B

*Table 19.* Mean (± std) accuracy of **Mixtral-8x7B** on all `DSR-Bench` tasks over three runs. Data structures marked with * are included in the `DSR-Bench-challenge` subset.

| Category | Data Structure | Operation | Short | Medium | Long |
|---|---|---|---|---|---|
| Linear | Array | Access | 0.87 (0.00) | 0.55 (0.02) | 0.26 (0.04) |
| | | Delete | 0.88 (0.02) | 0.54 (0.05) | 0.30 (0.00) |
| | | Insert | 0.68 (0.02) | 0.63 (0.05) | 0.22 (0.04) |
| | | Reverse | 0.60 (0.03) | 0.03 (0.00) | 0.02 (0.02) |
| | | Search | 0.87 (0.00) | 0.48 (0.02) | 0.38 (0.02) |
| Temporal | Stack | Compound | 0.08 (0.02) | 0.00 (0.00) | 0.00 (0.00) |
| | Queue | Compound | 0.26 (0.02) | 0.01 (0.02) | 0.00 (0.00) |
| | LRU Cache | Compound | 0.04 (0.05) | 0.02 (0.02) | 0.00 (0.00) |
| | Priority Queue* | Compound | 0.06 (0.02) | 0.00 (0.00) | 0.00 (0.00) |
| Associative | Hashmap* | Compound | 0.00 (0.00) | 0.00 (0.00) | 0.00 (0.00) |
| | Trie | Compound | 0.00 (0.00) | 0.02 (0.02) | 0.00 (0.00) |
| | Suffix Tree | Construct | 0.00 (0.00) | 0.00 (0.00) | 0.00 (0.00) |
| | Skip List* | Compound | 0.00 (0.00) | 0.00 (0.00) | 0.00 (0.00) |
| Hierarchical | BST | Insert | 0.00 (0.00) | 0.00 (0.00) | 0.00 (0.00) |
| | | Remove | 0.00 (0.00) | 0.00 (0.00) | 0.00 (0.00) |
| | | In-Order Traversal | 0.45 (0.02) | 0.30 (0.05) | 0.11 (0.02) |
| | | Pre-Order Traversal | 0.33 (0.00) | 0.10 (0.03) | 0.07 (0.00) |
| | | Post-Order Traversal | 0.04 (0.02) | 0.00 (0.00) | 0.00 (0.00) |
| | | Depth | 0.28 (0.02) | 0.05 (0.02) | 0.00 (0.00) |
| | | Compound | 0.11 (0.02) | 0.00 (0.00) | 0.00 (0.00) |
| | Heap* | Compound | 0.00 (0.00) | 0.00 (0.00) | 0.00 (0.00) |
| | | Heapify | 0.04 (0.02) | 0.00 (0.00) | 0.00 (0.00) |
| | RB Tree* | Construct | 0.00 (0.00) | 0.00 (0.00) | 0.00 (0.00) |
| | | Compound | 0.00 (0.00) | 0.00 (0.00) | 0.00 (0.00) |
| | $B^+$ Tree* | Compound | 0.03 (0.00) | 0.00 (0.00) | 0.00 (0.00) |
| | K-D Tree* | Compound | 0.00 (0.00) | 0.00 (0.00) | 0.00 (0.00) |
| | K-D Heap* | Compound | 0.03 (0.00) | 0.00 (0.00) | 0.00 (0.00) |
| Network | Graph | Depth-First Traversal | 0.00 (0.00) | 0.03 (0.00) | 0.00 (0.00) |
| | | Breadth-First Traversal | 0.03 (0.00) | 0.00 (0.00) | 0.00 (0.00) |
| | DSU | Compound | 0.00 (0.00) | 0.00 (0.00) | 0.00 (0.00) |
| | Geom Graph* | Construct | 0.00 (0.00) | 0.00 (0.00) | 0.00 (0.00) |
| Hybrid | Bloom Filter* | Compound | 0.00 (0.00) | 0.00 (0.00) | 0.00 (0.00) |
| | DAWG* | Compound | 0.00 (0.00) | 0.00 (0.00) | 0.00 (0.00) |

# E. Accuracy by prompting methods across instruction-tuned models

This section presents additional accuracy tables for tasks in `DSR-Bench`, evaluating each instruction-tuned model across five prompting methods: **Stepwise**, **0-CoT**, **CoT**, **3-shot**, and **None**. The results are shown in Table 20 (GPT-4.1), Table 21 (Gemini-2.0-Flash), Table 22 (Claude-3.5-Sonnet), Table 23 (DeepSeek-V3), and Table 24 (Llama-3.3).

*Table 20.* Mean (± std) accuracy of **GPT-4.1** across prompting methods over three runs.

| Data structure | Task | Stepwise | 0-CoT | CoT | 3-shot | None |
|---|---|---|---|---|---|---|
| Array | Access | 1.00 (0.00) | 1.00 (0.00) | 1.00 (0.00) | 1.00 (0.00) | 1.00 (0.00) |
| | Delete | 1.00 (0.00) | 1.00 (0.00) | 1.00 (0.00) | 1.00 (0.00) | 1.00 (0.00) |
| | Insert | 1.00 (0.00) | 1.00 (0.00) | 1.00 (0.00) | 1.00 (0.00) | 0.91 (0.08) |
| | Reverse | 1.00 (0.00) | 1.00 (0.00) | 1.00 (0.00) | 0.99 (0.02) | 0.98 (0.02) |
| | Search | 1.00 (0.00) | 1.00 (0.00) | 1.00 (0.00) | 1.00 (0.00) | 1.00 (0.00) |
| Queue | Compound | 1.00 (0.00) | 1.00 (0.00) | 1.00 (0.00) | 1.00 (0.00) | 0.82 (0.04) |
| Stack | Compound | 1.00 (0.00) | 1.00 (0.00) | 1.00 (0.00) | 1.00 (0.00) | 0.97 (0.00) |
| LRU Cache | Cache | 1.00 (0.00) | 1.00 (0.00) | 1.00 (0.00) | 1.00 (0.00) | 0.94 (0.02) |
| Priority Queue | Compound | 0.94 (0.04) | 0.99 (0.01) | 0.91 (0.02) | 0.94 (0.02) | 0.63 (0.03) |
| Hashmap | Compound | 0.96 (0.02) | 0.99 (0.01) | 1.00 (0.00) | 1.00 (0.00) | 0.19 (0.07) |
| Trie | Compound | 0.82 (0.02) | 0.98 (0.00) | 0.77 (0.07) | 0.68 (0.02) | 0.39 (0.07) |
| Suffix Tree | Construct | 0.49 (0.07) | 0.87 (0.01) | 0.69 (0.04) | 0.28 (0.08) | 0.00 (0.00) |
| Skip List | Compound | 0.77 (0.03) | 0.94 (0.01) | 0.56 (0.10) | 0.84 (0.02) | 0.21 (0.02) |
| BST | Insert | 0.99 (0.02) | 1.00 (0.00) | 0.97 (0.00) | 0.99 (0.02) | 0.79 (0.04) |
| | Remove | 0.99 (0.02) | 1.00 (0.00) | 0.99 (0.02) | 1.00 (0.00) | 0.78 (0.04) |
| | In-Order Traversal | 0.98 (0.02) | 1.00 (0.00) | 1.00 (0.00) | 0.98 (0.02) | 1.00 (0.00) |
| | Pre-Order Traversal | 1.00 (0.00) | 1.00 (0.00) | 1.00 (0.00) | 1.00 (0.00) | 1.00 (0.00) |
| | Post-Order Traversal | 1.00 (0.00) | 1.00 (0.00) | 1.00 (0.00) | 1.00 (0.00) | 0.82 (0.02) |
| | Depth | 1.00 (0.00) | 0.99 (0.02) | 1.00 (0.00) | 1.00 (0.00) | 0.30 (0.04) |
| | Compound | 1.00 (0.00) | 1.00 (0.00) | 0.98 (0.02) | 1.00 (0.00) | 0.69 (0.02) |
| Heap | Compound | 0.77 (0.07) | 0.96 (0.01) | 0.87 (0.06) | 0.78 (0.14) | 0.58 (0.02) |
| | Heapify | 0.99 (0.02) | 1.00 (0.01) | 0.96 (0.06) | 0.93 (0.07) | 0.57 (0.03) |
| RB Tree | Construct | 0.40 (0.13) | 0.91 (0.02) | 0.40 (0.12) | 0.38 (0.05) | 0.12 (0.02) |
| | Compound | 0.77 (0.03) | 0.96 (0.01) | 0.37 (0.12) | 0.70 (0.07) | 0.31 (0.04) |
| B+ Tree | Compound | 0.71 (0.05) | 0.93 (0.01) | 0.77 (0.03) | 0.60 (0.06) | 0.27 (0.00) |
| Graph | Breadth-First Traversal | 0.90 (0.03) | 0.94 (0.02) | 0.83 (0.00) | 0.82 (0.07) | 0.31 (0.05) |
| | Depth-First Traversal | 0.86 (0.05) | 0.92 (0.02) | 0.83 (0.03) | 0.80 (0.03) | 0.50 (0.03) |
| DSU | Compound | 0.67 (0.03) | 0.93 (0.02) | 0.67 (0.07) | 0.62 (0.05) | 0.06 (0.02) |
| Bloom Filter | Compound | 0.36 (0.07) | 0.91 (0.02) | 0.26 (0.08) | 0.42 (0.07) | 0.10 (0.00) |
| DAWG | Compound | 0.20 (0.00) | 0.79 (0.01) | 0.21 (0.02) | 0.20 (0.00) | 0.16 (0.02) |

*Table 21.* Mean (± std) accuracy of **Gemini-2.0-Flash** across prompting methods over three runs.

| Data structure | Task | Stepwise | 0-CoT | CoT | 3-shot | None |
|---|---|---|---|---|---|---|
| Array | Access | 1.00 (0.00) | 1.00 (0.00) | 1.00 (0.00) | 1.00 (0.00) | 1.00 (0.00) |
| | Delete | 1.00 (0.00) | 1.00 (0.00) | 1.00 (0.00) | 1.00 (0.00) | 0.96 (0.02) |
| | Insert | 1.00 (0.00) | 1.00 (0.00) | 1.00 (0.00) | 1.00 (0.00) | 0.99 (0.02) |
| | Reverse | 0.67 (0.00) | 0.78 (0.19) | 0.56 (0.19) | 1.00 (0.00) | 0.96 (0.02) |
| | Search | 1.00 (0.00) | 1.00 (0.00) | 1.00 (0.00) | 1.00 (0.00) | 1.00 (0.00) |
| Queue | Compound | 0.91 (0.05) | 0.93 (0.03) | 0.94 (0.02) | 0.94 (0.02) | 0.87 (0.00) |
| Stack | Compound | 0.93 (0.07) | 0.92 (0.04) | 0.73 (0.09) | 0.97 (0.03) | 0.67 (0.00) |
| LRU Cache | Cache | 0.97 (0.00) | 0.96 (0.04) | 0.82 (0.04) | 0.87 (0.06) | 0.93 (0.00) |
| Priority Queue | Compound | 0.53 (0.12) | 0.62 (0.12) | 0.53 (0.12) | 0.72 (0.05) | 0.38 (0.02) |
| Hashmap | Compound | 0.42 (0.08) | 0.56 (0.11) | 0.67 (0.09) | 0.63 (0.07) | 0.28 (0.05) |
| Trie | Compound | 0.26 (0.04) | 0.27 (0.12) | 0.19 (0.07) | 0.33 (0.03) | 0.31 (0.02) |
| Suffix Tree | Construct | 0.13 (0.00) | 0.11 (0.05) | 0.22 (0.07) | 0.18 (0.05) | 0.00 (0.00) |
| Skip List | Compound | 0.18 (0.04) | 0.31 (0.08) | 0.27 (0.03) | 0.31 (0.08) | 0.16 (0.02) |
| BST | Insert | 0.62 (0.02) | 0.58 (0.13) | 0.66 (0.02) | 0.64 (0.08) | 0.31 (0.02) |
| | Remove | 0.64 (0.12) | 0.67 (0.07) | 0.73 (0.03) | 0.69 (0.08) | 0.63 (0.09) |
| | In-Order Traversal | 0.87 (0.03) | 0.86 (0.02) | 0.92 (0.02) | 0.80 (0.03) | 0.87 (0.00) |
| | Pre-Order Traversal | 1.00 (0.00) | 1.00 (0.00) | 1.00 (0.00) | 1.00 (0.00) | 1.00 (0.00) |
| | Post-Order Traversal | 0.86 (0.05) | 0.78 (0.04) | 0.86 (0.04) | 0.84 (0.08) | 0.63 (0.00) |
| | Depth | 1.00 (0.00) | 1.00 (0.00) | 1.00 (0.00) | 0.86 (0.02) | 0.13 (0.09) |
| | Compound | 0.66 (0.08) | 0.74 (0.08) | 0.80 (0.03) | 0.69 (0.08) | 0.51 (0.05) |
| Heap | Compound | 0.40 (0.09) | 0.37 (0.07) | 0.44 (0.13) | 0.39 (0.05) | 0.32 (0.05) |
| | Heapify | 0.51 (0.11) | 0.61 (0.05) | 0.43 (0.09) | 0.54 (0.08) | 0.23 (0.06) |
| RB Tree | Construct | 0.08 (0.07) | 0.07 (0.03) | 0.04 (0.02) | 0.03 (0.00) | 0.08 (0.02) |
| | Compound | 0.41 (0.08) | 0.44 (0.02) | 0.28 (0.04) | 0.31 (0.02) | 0.43 (0.03) |
| B+ Tree | Compound | 0.33 (0.09) | 0.39 (0.08) | 0.47 (0.03) | 0.50 (0.09) | 0.17 (0.00) |
| Graph | Breadth-First Traversal | 0.17 (0.03) | 0.24 (0.10) | 0.36 (0.07) | 0.20 (0.03) | 0.10 (0.00) |
| | Depth-First Traversal | 0.20 (0.09) | 0.27 (0.12) | 0.30 (0.03) | 0.08 (0.02) | 0.19 (0.02) |
| DSU | Compound | 0.29 (0.02) | 0.20 (0.00) | 0.17 (0.09) | 0.12 (0.02) | 0.01 (0.02) |
| Bloom Filter | Compound | 0.33 (0.07) | 0.20 (0.00) | 0.12 (0.04) | 0.29 (0.08) | 0.10 (0.00) |
| DAWG | Compound | 0.16 (0.02) | 0.18 (0.02) | 0.13 (0.00) | 0.14 (0.02) | 0.18 (0.02) |

*Table 22.* Mean (± std) accuracy of **Claude-3.5-Sonnet** across prompting methods over three runs.

| Data structure | Task | Stepwise | 0-CoT | CoT | 3-shot | None |
|---|---|---|---|---|---|---|
| Array | Access | 1.00 (0.00) | 1.00 (0.00) | 1.00 (0.00) | 1.00 (0.00) | 1.00 (0.00) |
| | Delete | 1.00 (0.00) | 1.00 (0.00) | 1.00 (0.00) | 1.00 (0.00) | 1.00 (0.00) |
| | Insert | 1.00 (0.00) | 1.00 (0.00) | 1.00 (0.00) | 1.00 (0.00) | 1.00 (0.00) |
| | Reverse | 1.00 (0.00) | 0.99 (0.02) | 0.96 (0.04) | 0.99 (0.02) | 1.00 (0.00) |
| | Search | 1.00 (0.00) | 1.00 (0.00) | 1.00 (0.00) | 1.00 (0.00) | 1.00 (0.00) |
| Queue | Compound | 1.00 (0.00) | 1.00 (0.00) | 1.00 (0.00) | 1.00 (0.00) | 0.87 (0.00) |
| Stack | Compound | 1.00 (0.00) | 1.00 (0.00) | 1.00 (0.00) | 1.00 (0.00) | 1.00 (0.00) |
| LRU Cache | Cache | 1.00 (0.00) | 1.00 (0.00) | 1.00 (0.00) | 1.00 (0.00) | 0.99 (0.02) |
| Priority Queue | Compound | 0.94 (0.02) | 0.96 (0.02) | 0.90 (0.03) | 0.94 (0.02) | 0.63 (0.00) |
| Hashmap | Compound | 0.89 (0.02) | 0.81 (0.04) | 1.00 (0.00) | 1.00 (0.00) | 0.37 (0.03) |
| Trie | Compound | 0.02 (0.02) | 0.11 (0.07) | 0.00 (0.00) | 0.11 (0.02) | 0.89 (0.04) |
| Suffix Tree | Construct | 0.29 (0.08) | 0.24 (0.07) | 0.56 (0.08) | 0.27 (0.03) | 0.21 (0.02) |
| Skip List | Compound | 0.82 (0.02) | 0.77 (0.03) | 0.64 (0.04) | 0.61 (0.02) | 0.77 (0.06) |
| BST | Insert | 1.00 (0.00) | 1.00 (0.00) | 0.96 (0.04) | 0.92 (0.05) | 0.80 (0.06) |
| | Remove | 1.00 (0.00) | 1.00 (0.00) | 0.97 (0.00) | 0.98 (0.02) | 0.96 (0.04) |
| | In-Order Traversal | 0.88 (0.05) | 0.97 (0.06) | 1.00 (0.00) | 0.98 (0.02) | 0.97 (0.03) |
| | Pre-Order Traversal | 1.00 (0.00) | 1.00 (0.00) | 1.00 (0.00) | 1.00 (0.00) | 1.00 (0.00) |
| | Post-Order Traversal | 0.99 (0.02) | 0.99 (0.02) | 1.00 (0.00) | 1.00 (0.00) | 1.00 (0.00) |
| | Depth | 1.00 (0.00) | 0.99 (0.02) | 1.00 (0.00) | 1.00 (0.00) | 1.00 (0.00) |
| | Compound | 0.89 (0.05) | 0.88 (0.05) | 0.88 (0.04) | 0.86 (0.05) | 0.77 (0.07) |
| Heap | Compound | 0.69 (0.02) | 0.69 (0.02) | 0.66 (0.04) | 0.68 (0.02) | 0.78 (0.04) |
| | Heapify | 0.73 (0.06) | 0.69 (0.04) | 0.33 (0.06) | 0.80 (0.07) | 0.53 (0.12) |
| RB Tree | Construct | 0.11 (0.08) | 0.08 (0.05) | 0.19 (0.07) | 0.21 (0.02) | 0.13 (0.04) |
| | Compound | 0.57 (0.00) | 0.60 (0.03) | 0.03 (0.00) | 0.13 (0.03) | 0.44 (0.02) |
| B+ Tree | Compound | 0.61 (0.02) | 0.69 (0.04) | 0.67 (0.03) | 0.44 (0.05) | 0.40 (0.00) |
| Graph | Breadth-First Traversal | 0.30 (0.09) | 0.32 (0.04) | 0.52 (0.02) | 0.26 (0.04) | 0.17 (0.03) |
| | Depth-First Traversal | 0.30 (0.09) | 0.24 (0.04) | 0.26 (0.05) | 0.23 (0.03) | 0.13 (0.03) |
| DSU | Compound | 0.53 (0.07) | 0.49 (0.05) | 0.76 (0.08) | 0.53 (0.09) | 0.07 (0.03) |
| Bloom Filter | Compound | 0.12 (0.04) | 0.12 (0.02) | 0.10 (0.00) | 0.10 (0.00) | 0.10 (0.00) |
| DAWG | Compound | 0.17 (0.06) | 0.19 (0.02) | 0.18 (0.02) | 0.19 (0.02) | 0.20 (0.00) |

*Table 23.* Mean (± std) accuracy of **DeepSeek-V3** across prompting methods over three runs.

| Data structure | Task | Stepwise | 0-CoT | CoT | 3-shot | None |
|---|---|---|---|---|---|---|
| Array | Access | 1.00 (0.00) | 1.00 (0.00) | 1.00 (0.00) | 1.00 (0.00) | 1.00 (0.00) |
| | Delete | 1.00 (0.00) | 1.00 (0.00) | 1.00 (0.00) | 1.00 (0.00) | 1.00 (0.00) |
| | Insert | 1.00 (0.00) | 1.00 (0.00) | 1.00 (0.00) | 1.00 (0.00) | 1.00 (0.00) |
| | Reverse | 1.00 (0.00) | 1.00 (0.00) | 0.99 (0.00) | 1.00 (0.00) | 1.00 (0.00) |
| | Search | 1.00 (0.00) | 1.00 (0.00) | 1.00 (0.00) | 1.00 (0.00) | 1.00 (0.00) |
| Queue | Compound | 1.00 (0.00) | 1.00 (0.00) | 1.00 (0.00) | 1.00 (0.00) | 0.84 (0.02) |
| Stack | Compound | 1.00 (0.00) | 1.00 (0.00) | 1.00 (0.00) | 1.00 (0.00) | 0.70 (0.03) |
| LRU Cache | Cache | 0.93 (0.00) | 0.97 (0.00) | 0.98 (0.02) | 0.90 (0.00) | 0.94 (0.02) |
| Priority Queue | Compound | 0.86 (0.04) | 0.79 (0.05) | 0.84 (0.02) | 0.82 (0.02) | 0.53 (0.03) |
| Hashmap | Compound | 1.00 (0.00) | 1.00 (0.00) | 0.90 (0.03) | 1.00 (0.00) | 0.04 (0.02) |
| Trie | Compound | 0.00 (0.00) | 0.00 (0.00) | 0.63 (0.00) | 0.64 (0.02) | 0.00 (0.00) |
| Suffix Tree | Construct | 0.19 (0.02) | 0.18 (0.02) | 0.39 (0.04) | 0.19 (0.02) | 0.06 (0.02) |
| Skip List | Compound | 0.08 (0.02) | 0.06 (0.02) | 0.19 (0.04) | 0.08 (0.02) | 0.06 (0.02) |
| BST | Insert | 0.94 (0.02) | 0.98 (0.02) | 0.91 (0.02) | 0.87 (0.06) | 0.93 (0.03) |
| | Remove | 0.92 (0.04) | 0.70 (0.03) | 0.84 (0.04) | 0.82 (0.05) | 0.84 (0.04) |
| | In-Order Traversal | 0.94 (0.04) | 0.92 (0.02) | 0.94 (0.04) | 0.93 (0.00) | 0.97 (0.00) |
| | Pre-Order Traversal | 1.00 (0.00) | 1.00 (0.00) | 1.00 (0.00) | 1.00 (0.00) | 1.00 (0.00) |
| | Post-Order Traversal | 1.00 (0.00) | 1.00 (0.00) | 1.00 (0.00) | 0.99 (0.02) | 0.82 (0.02) |
| | Depth | 0.67 (0.03) | 1.00 (0.00) | 1.00 (0.00) | 1.00 (0.00) | 0.82 (0.31) |
| | Compound | 0.81 (0.02) | 0.77 (0.03) | 0.96 (0.02) | 0.71 (0.04) | 0.68 (0.02) |
| RB Tree | Construct | 0.04 (0.04) | 0.02 (0.04) | 0.06 (0.02) | 0.07 (0.06) | 0.09 (0.02) |
| | Compound | 0.63 (0.03) | 0.67 (0.03) | 0.12 (0.02) | 0.59 (0.08) | 0.30 (0.03) |
| B+ Tree | Compound | 0.71 (0.04) | 0.66 (0.02) | 0.44 (0.08) | 0.38 (0.02) | 0.14 (0.05) |
| Heap | Compound | 0.83 (0.03) | 0.87 (0.03) | 0.87 (0.03) | 0.89 (0.05) | 0.23 (0.00) |
| | Heapify | 0.83 (0.06) | 0.83 (0.03) | 0.57 (0.03) | 0.81 (0.02) | 0.59 (0.02) |
| Graph | Breadth-First Traversal | 0.51 (0.02) | 0.51 (0.04) | 0.29 (0.08) | 0.00 (0.00) | 0.29 (0.05) |
| | Depth-First Traversal | 0.23 (0.03) | 0.39 (0.05) | 0.36 (0.05) | 0.00 (0.00) | 0.22 (0.02) |
| DSU | Compound | 0.34 (0.04) | 0.41 (0.05) | 0.30 (0.03) | 0.16 (0.02) | 0.03 (0.00) |
| Bloom Filter | Compound | 0.10 (0.00) | 0.10 (0.00) | 0.10 (0.00) | 0.10 (0.00) | 0.10 (0.00) |
| DAWG | Compound | 0.18 (0.02) | 0.18 (0.02) | 0.17 (0.00) | 0.18 (0.02) | 0.17 (0.00) |

*Table 24.* Mean (± std) accuracy of **Llama-3.3** across prompting methods over three runs.

| Data structure | Task | Stepwise | 0-CoT | CoT | 3-shot | None |
|---|---|---|---|---|---|---|
| Array | Access | 0.98 (0.02) | 1.00 (0.00) | 0.97 (0.03) | 0.99 (0.02) | 0.98 (0.02) |
| | Delete | 0.92 (0.05) | 0.96 (0.02) | 0.78 (0.05) | 0.82 (0.08) | 0.81 (0.08) |
| | Insert | 0.78 (0.02) | 0.88 (0.07) | 0.81 (0.04) | 0.89 (0.10) | 0.76 (0.04) |
| | Reverse | 0.96 (0.02) | 0.87 (0.06) | 0.50 (0.07) | 0.74 (0.02) | 0.91 (0.02) |
| | Search | 0.99 (0.02) | 0.99 (0.02) | 0.96 (0.02) | 0.98 (0.02) | 1.00 (0.00) |
| Skip List | Compound | 0.14 (0.04) | 0.10 (0.03) | 0.18 (0.02) | 0.20 (0.07) | 0.03 (0.00) |
| Queue | Compound | 0.93 (0.03) | 0.94 (0.05) | 0.94 (0.02) | 0.81 (0.05) | 0.58 (0.11) |
| Stack | Compound | 0.88 (0.02) | 0.86 (0.13) | 0.88 (0.10) | 0.68 (0.07) | 0.09 (0.02) |
| LRU Cache | Compound | 0.89 (0.02) | 0.92 (0.05) | 0.98 (0.04) | 0.82 (0.07) | 0.74 (0.08) |
| Priority Queue | Compound | 0.73 (0.03) | 0.73 (0.10) | 0.69 (0.02) | 0.76 (0.04) | 0.21 (0.05) |
| Hashmap | Compound | 0.39 (0.16) | 0.23 (0.07) | 0.32 (0.04) | 0.13 (0.00) | 0.00 (0.00) |
| Trie | Compound | 0.01 (0.02) | 0.00 (0.00) | 0.10 (0.09) | 0.14 (0.10) | 0.01 (0.02) |
| Suffix Tree | Construct | 0.01 (0.02) | 0.01 (0.02) | 0.07 (0.03) | 0.00 (0.00) | 0.00 (0.00) |
| BST | Insert | 0.53 (0.07) | 0.52 (0.16) | 0.38 (0.05) | 0.37 (0.09) | 0.37 (0.00) |
| | Remove | 0.57 (0.03) | 0.57 (0.15) | 0.40 (0.25) | 0.60 (0.06) | 0.49 (0.04) |
| | In-Order Traversal | 0.56 (0.08) | 0.51 (0.13) | 0.69 (0.02) | 0.60 (0.12) | 0.60 (0.06) |
| | Pre-Order Traversal | 0.60 (0.15) | 0.78 (0.10) | 0.67 (0.02) | 0.82 (0.07) | 0.86 (0.11) |
| | Post-Order Traversal | 0.54 (0.26) | 0.61 (0.05) | 0.90 (0.03) | 0.71 (0.04) | 0.31 (0.04) |
| | Depth | 0.70 (0.00) | 0.83 (0.03) | 0.82 (0.08) | 0.99 (0.02) | 0.93 (0.03) |
| | Compound | 0.54 (0.16) | 0.51 (0.05) | 0.50 (0.09) | 0.49 (0.10) | 0.27 (0.15) |
| Heap | Compound | 0.33 (0.06) | 0.33 (0.03) | 0.32 (0.05) | 0.41 (0.11) | 0.17 (0.03) |
| | Heapify | 0.29 (0.02) | 0.23 (0.12) | 0.08 (0.02) | 0.18 (0.05) | 0.24 (0.05) |
| RB Tree | Construct | 0.01 (0.02) | 0.01 (0.02) | 0.01 (0.02) | 0.07 (0.03) | 0.00 (0.00) |
| | Compound | 0.36 (0.04) | 0.30 (0.10) | 0.03 (0.00) | 0.28 (0.07) | 0.31 (0.02) |
| B+ Tree | Compound | 0.17 (0.03) | 0.18 (0.02) | 0.50 (0.00) | 0.23 (0.03) | 0.02 (0.04) |
| Graph | Breadth-First Traversal | 0.12 (0.02) | 0.12 (0.07) | 0.28 (0.08) | 0.06 (0.07) | 0.04 (0.05) |
| | Depth-First Traversal | 0.12 (0.08) | 0.14 (0.02) | 0.09 (0.05) | 0.10 (0.07) | 0.07 (0.06) |
| DSU | Construct | 0.08 (0.05) | 0.04 (0.02) | 0.37 (0.15) | 0.07 (0.00) | 0.00 (0.00) |
| Bloom Filter | Compound | 0.01 (0.02) | 0.01 (0.02) | 0.00 (0.00) | 0.01 (0.02) | 0.00 (0.00) |
| DAWG | Compound | 0.03 (0.03) | 0.02 (0.02) | 0.06 (0.02) | 0.07 (0.06) | 0.02 (0.02) |

## E.1. Additional analysis on CoT prompting

In Section 4.1.1, we analyzed various prompting strategies with instruction-tuned models. We further examined the CoT method by inspecting its reasoning outputs, which offers insights we hope will benefit practitioners and researchers:

- **CoT offers limited benefits for well-known tasks.** For familiar problems that models are likely to have encountered during pretraining (such as ARRAY, QUEUE, and BINARY SEARCH TREE), CoT prompting yields only marginal gains. This suggests that models already possess internalized procedures for these tasks and can execute them reliably without additional reasoning steps. Simpler prompts can be more cost efficient and effective.

- **Without careful design, CoT can hurt performance.** In more complex scenarios, such as the HASHMAP compound task, using CoT with structured JSON-style reasoning actually degraded accuracy. When the reasoning format was changed to natural language, performance recovered to match the baseline (0-CoT). This highlights that the effectiveness of CoT is highly sensitive to prompt design, and poorly chosen reasoning formats may introduce unnecessary complexity rather than aiding problem solving.

## E.2. Additional prompting strategies

We conduct experiments with additional prompting strategies: Plan-and-Solve (Wang et al., 2023b), Self-Consistency (Wang et al., 2023c) (3 rollouts), CodeEnforce (which is equivalent to Program-of-thought (Chen et al., 2023) without code interpreter), Least-to-Most (Zhou et al., 2023) (with manually designed decomposition).

As observed from Table 25, our claims about prompting in Section 4.1.1 remain robust and still hold: (i) Lightweight prompts (Stepwise, 0-CoT, Plan-and-Solve, Self-Consistency) consistently improve performance. (ii) Crafted prompts (CoT, 3-Shot, Least-to-Most) can be most effective but require careful design.

## E.3. Effect of "Answer the question in <number> tokens"

Following recommended practices in prompt engineering, we also append the instruction *"Answer the question in <number> tokens"* in our prompt templates to encourage concise outputs within a specified token budget. We conduct an ablation with

*Table 25.* Additional prompting methods on a selection of data structure tasks.

|  | None | Stepwise | 0-CoT | Plan-and-Solve | Self-Consistency | CodeEnforce | 3-Shot | CoT | Least-to-Most |
|---|---|---|---|---|---|---|---|---|---|
| Heap | 0.58 | 0.77 | 0.96 | 0.88 | 0.78 | 0.51 | 0.78 | 0.87 | 0.91 |
| Suffix Tree | 0.00 | 0.49 | 0.87 | 0.47 | 0.31 | 0.07 | 0.28 | 0.69 | 0.38 |
| RB Tree | 0.12 | 0.40 | 0.91 | 0.20 | 0.19 | 0.13 | 0.38 | 0.40 | 0.11 |
| DAWG | 0.16 | 0.20 | 0.79 | 0.53 | 0.36 | 0.13 | 0.20 | 0.21 | 0.49 |

*Table 26.* Results for removing "Answer the question in <number> tokens" from the prompt on the priority queue task.

| Condition | Short | Medium | Long |
|---|---|---|---|
| Answer in $N$ tokens | 0.84 | 0.44 | 0.28 |
| Remove instruction | 0.87 | 0.43 | 0.23 |

GPT-5 by removing the instruction and observe minimal impact on performance.

## E.4. Additional Analysis of Zero Scores

We conducted additional analysis to investigate the causes of zero scores by programmatically extracting error statistics and manually inspecting model outputs. We examine two settings: the same task across different models, and different tasks for the same model.

### E.4.1. SAME TASK ACROSS DIFFERENT MODELS

We first analyze the KD-TREE task, which received the largest number of zero scores, across four models: Claude-3.7-Sonnet, DeepSeek-V3, GPT-4.1, and Qwen3-8B.

**Common failure patterns.** Across models, zero scores are often caused by small but consequential errors early in the construction process. Because KD-TREE construction requires recursive partitioning in a $k$-dimensional space, an incorrect split near the root can invalidate the entire tree under exact-match evaluation. The most common source of such errors is incorrect median selection. Models frequently choose the element at position $\lfloor n/2 \rfloor \pm 1$ rather than the correct position $\lfloor n/2 \rfloor$, even when the input size is odd and no tie-breaking is required. When the number of points is even, models also sometimes fail to apply the tie-breaking rule specified in the prompt, leading to incorrect recursive partitions.

**Model-specific behaviors.** Although several models receive similarly near-zero scores, their failures are qualitatively different. GPT-4.1 tends to exhibit a rightward bias when selecting the median, often overshooting by $+2$ or $+3$ positions. By contrast, DeepSeek-V3 and Claude-3.7-Sonnet show more balanced median-selection offsets. DeepSeek-V3 also occasionally selects the minimum or maximum element instead of the median, a behavior not observed in the other analyzed models. Qwen3-8B fails in a different manner: rather than primarily making near-miss median errors, it often produces unparsable outputs, hallucinates points, or returns undersized trees.

### E.4.2. DIFFERENT TASKS FOR THE SAME MODEL

We next examine Qwen3-8B across four tasks: TRIE (associative), KD-HEAP (hierarchical), GEOM GRAPH (network), and DAWG (hybrid).

**Key observations.** Across these tasks, Qwen3-8B displays several consistent failure modes (see also Table 27). The model appears to retrieve surface-level knowledge of the data structures described in the prompts, but it often fails to execute the required construction or transformation correctly. In many cases, it drops elements from earlier states, suggesting difficulty maintaining context across multi-step reasoning processes. It also appears to rely on superficial pattern matching: for example, it occasionally outputs in $box\{\}$ format, which is common in some training data but is not required by the prompt, and it sometimes responds in Chinese despite receiving no Chinese input.

| Metric | Deepseek-V3 | Claude-3.7-Sonnet | GPT-4.1 | Qwen3-8B |
|---|---|---|---|---|
| Binary accuracy | 0.0 | 0.0 | 0.0 | 0.01 |
| Levenshtein distance | 0.5 | 0.66 | 0.53 | 0.10 |
| % format error | 0% | 0% | 0% | 43% |
| Hallucinated values | Never | Never | Never | Often |
| Wrong dimensionality | Seldomly | Never | Never | Often |
| Primary Error Mode | Median offset (balanced) | Median offset (balanced) | Median offset (bias right offset) | Format/comprehension failure |

*Table 27.* Summary of error modes and qualitative analysis on four models that scores 0 with the 0-1 binary metric on KD-Tree construct task.

## F. The `spatial` probe supplementary materials

Figure 8 presents example illustrations of the non-uniform input distributions: circles, moons, and blobs, which were adopted from `scikit-learn`. These synthetic patterns are used to evaluate whether models can adapt to irregular and non-uniform spatial distributions, an essential aspect of real-world data, as discussed in Section 4.2.

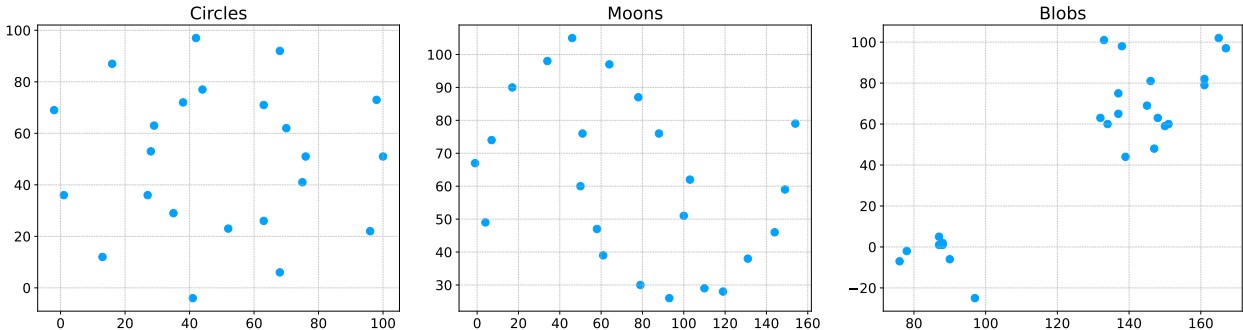

*Figure 8.* Example K-D Tree instances from three non-uniform distributions.

We observe a performance drop in KD-TREE tasks when the input data is non-uniform. At the problem level, uniform and non-uniform KD-TREE tasks appear equally difficult for a human reasoner. Across 30 questions per group, duplicated indices, which trigger tie-breaking, occur at similar rates on both axes (x: 32 vs. 33; y: 26 vs. 28), and the total number of median operations is comparable (156 vs. 145). This suggests the surface-level difficulty is well matched across distributions. A closer inspection of errors reveals two main causes:

**Non-uniform data often increases ambiguity near the global median.** Non-uniform distributions often produce clusters of similar or identical coordinates near the global median, making it harder for the model to select the correct split. Since the root node is determined by a median over the entire dataset, this ambiguity leads to more errors. Supporting this, we observed 13 out of 30 correct root nodes on uniform data, but only 5 out of 30 on circle (non-uniform) data.

**Even simple tasks, such as median calculation, can be challenging for LLMs.** On non-uniform data, models exhibited 28.6% more cases of violating the prompt's "latter-median" rule by instead choosing the "former-median," and 50% more instances of selecting a completely incorrect median (neither the former nor latter). Additionally, while no axis confusion errors occurred on uniform data, the model made three such errors on non-uniform data, mistakenly selecting the wrong axis before performing median calculation.

# G. The `realistic` probe supplementary materials

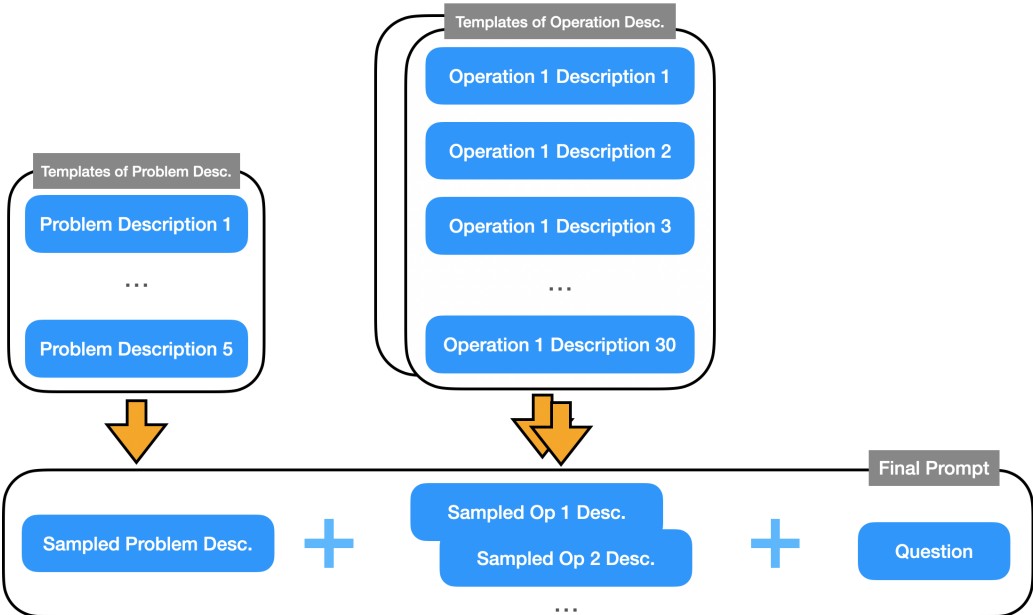

*Figure 9.* The pipeline for generating natural language prompts.

Figure 9 illustrates our generation process for natural language prompts in `DSR-Bench-realistic`. For each data structure task, we begin by manually writing an initial scenario narrative (see On a sunny afternoon... arrived in the queue" in Section G). This narrative is then paraphrased into five variants using `GPT-4o`. For each supported operation type (e.g., enqueue, dequeue), we generate 30 paraphrased operation templates in a similar manner (e.g., With money in hand, Yuki Lopez stepped into the queue." corresponds to "enqueue Yuki Lopez" in Section G). In total, each task is backed by a pool of five scenario descriptions and 30 templates per operation type.

We use the same input distribution as in the `DSR-Bench`, replacing synthetic values (e.g. "enqueue 5") with realistic values (e.g. "enqueue Yuki Lopez"). During prompt construction, we randomly sample one scenario description. For each data structure operation, we instantiate it with a synthetically generated value (e.g., a name) and sample one operation template to form a complete instance. All templates were reviewed by three human annotators to ensure clarity and unambiguous solvability.

**Queue (children buying ice cream)**   We construct a real-world scenario that implicitly models a `QUEUE`: children lining up to buy ice cream from a truck. The enqueue operation corresponds to a child joining the end of the line, while the dequeue operation represents a child being served from the front. The scenario explicitly enforces the FIFO discipline by stating that no skipping is allowed.

---

**Example prompt from the natural language extension for QUEUE.**

On a sunny afternoon in the neighborhood park, an ice cream truck rolled in, its cheerful tune drawing children from all directions. The children began to form a line. Each child joined at the end while the vendor served at the front. Coins jingled in pockets while the children eagerly discussed the different flavors. The children were served in the order they had arrived in the queue.

- Fatima Singh ran over from the swings and joined the ice cream line.
- With money in hand, Yuki Lopez stepped into the queue.
- Haruto Sanchez spotted the growing line and quietly joined.
- A cone was handed over, and the line moved on.
- After hearing about the ice cream truck from Fatima Singh, Isabella Miller decided to line up too.
- One more excited customer walked off with a cone in hand.
- Carlos Martinez joined the queue after Yuki Lopez mentioned how good the ice cream looked.

**Q**: What is the order of the remaining kids in line? Your answer should be a list of names.
Answer the question in 8000 tokens.

---

**BST (clinic appointments)**    We construct a scenario in which a clinic uses a BST to store patient appointments, ordered by appointment time and tie-broken by the patient's name. The insert operation adds a (name, time) appointment to the tree, while the delete operation corresponds to a patient canceling their appointment. To retrieve all records, a pre-order traversal of the tree is performed.

---

**Example prompt from the natural language extension for BST.**

A local clinic uses an appointment management system which maintains a binary search tree to store appointments. Each appointment (name, appointment time) is a tuple of two strings, e.g. ('Alice Baker', '10:30'), and is represented by a node in the tree. Order is maintained by appointment time. The alphabetical order of the patients' names is used to break ties. During data retrieval (i.e. to print out all of the appointments), a pre-order traversal is used, starting from the root node. Initially the tree is empty.

- Hassan Chen joined the list at 09:22 successfully.
- Knowing Hassan Chen has booked, Amelia Martinez was placed at 13:11.
- Hassan Chen hesitated a lot but still decided to cancel.
- As recommended by a friend Harper Young, Lucas Fernandez quickly booked at 09:18.
- Harper Young was scheduled for 15:48, slightly earlier than 16:01.

**Q**: What is the pre-order traversal of the appointment schedule following the binary search tree? Your answer should be a list of (name, appointment time) in the format of a tuple of two strings. Answer the question in 8000 tokens.

---

**Graph (galaxy traveling)**    We create a scenario set in a galaxy where planets are connected by space tunnels. The task is to navigate a starship to visit as many planets as possible using depth-first search, starting from a given planet and visiting neighbors in lexicographical order.

---

**Example prompt from the natural language extension for GRAPH.**

You pilot a Star Courier through a galaxy of planets. Your job is to travel to as many planets as possible via bidirectional space tunnels, starting from a source planet. The courier computes its route with depth-first search, and whenever multiple unvisited neighbors are available it selects the neighbor with the alphabetically earliest planet name.

- Star maps show a space tunnel running between Triton and Pulsar.
- A space tunnel links Ganymede and Wraith.
- The tunnel from Triton to Ganymede is well-known for its convenience.
- There's a tunnel between Fenrir and Vega.
- The tunnel linking Fenrir and Pulsar is a crucial route for all space dwellers.
- Long-range scans confirm a navigable tunnel between Triton and Orion.
- Though Vega is nearby, the space team decides to connect Nereus and Pulsar via a tunnel.
- Vega and Orion are part of the same local cluster, connected by a space tunnel.
- Vega and Ymir are directly linked by a tunnel monitored by the space police.
- The ancient network includes a direct tunnel between Wraith and Pulsar.
- Only Pulsar and Orion are reachable via this tunnel — not Nereus.

**Q**: What is the full DFS traversal order (as a list of planet names) starting from Fenrir? Answer the question in 8000 tokens.

---

## H. The `code` probe supplementary materials

Our benchmark is designed to evaluate *general (inherent) reasoning* in LLMs without relying on external tools such as code interpreters or formal solvers. The focus is on the underlying reasoning ability that drives broad problem-solving, beyond coding or domains where code is directly applicable. This follows recent evaluation practices, such as Gemini-Deep-Think and OpenAI's IMO assessments (Luong & Lockhart, 2025; Wei, 2025), which deliberately disallowed code or tool use (e.g., Lean) to test end-to-end reasoning. By using data structures as controlled, interpretable settings for structural reasoning, DSR-Bench differs from code-synthesis benchmarks that emphasize syntax or API knowledge and are often vulnerable to training data contamination.

To further investigate how code generation may support structural reasoning, we conducted an ablation across six models, including reasoning models with competitive performance on code-synthesis benchmarks. We cover seven data structures of varying difficulty (ARRAY, QUEUE, HASHMAP, HEAP, DAWG, GEOM GRAPH, GRAPH-NATURAL) and average results over three runs. Three code-generation modes were tested, with prompts shown below:

- *CodeOnly*: "Your answer should be a Python function 'def solution()' that takes no inputs, solves the problem, and returns the final solution in the expected format. Output the code only."
- *CodeEnforce*: "You should write code to solve the problem, then reason through its execution to explain what the output would be. Your final answer should be the output itself."
- *CodeMaybe*: "You can write code to help solve the problem, or solve it directly. If you use code, reason through its execution to explain what the output would be. Your final answer should be the output itself."

While GPT-4.1 and o4-mini generally perform well, we observe that Claude models also perform competitively, particularly in spatial data structure tasks such as GEOM GRAPH. The only exception is GRAPH-NATURAL, suggesting that Claude models struggle with handling natural language ambiguity. On the other hand, Gemini models tend to return final answers directly rather than code in *CodeOnly* mode, despite explicit instructions ("Your answer should be a Python function... Output the code only."). This indicates weaker instruction-following in code generation compared to other models and contributes to their low performance. We investigated this further and presented a new table showing (i) task accuracy when code is written (Code Accuracy) and (ii) the code-writing frequency of Gemini models (Code Frequency). We observe that Gemini models also perform on par when code is written.

*Table 28.* Individual scores with code generation by GPT-4.1 and o4-mini.

| | GPT-4.1 | | | | o4-mini | | | |
|---|---|---|---|---|---|---|---|---|
| **Structure** | None | CodeMaybe | CodeEnforce | CodeOnly | None | CodeMaybe | CodeEnforce | CodeOnly |
| Array | 1.00 | 1.00 | 1.00 | 1.00 | 1.00 | 1.00 | 1.00 | 1.00 |
| Queue | 0.82 | 0.84 | 0.89 | 0.98 | 1.00 | 1.00 | 1.00 | 1.00 |
| Hashmap | 0.19 | 0.11 | 0.09 | 1.00 | 0.89 | 0.89 | 0.86 | 0.94 |
| Heap | 0.58 | 0.53 | 0.53 | 1.00 | 0.44 | 0.73 | 0.70 | 0.53 |
| DAWG | 0.16 | 0.17 | 0.11 | 0.90 | 0.49 | 0.46 | 0.30 | 0.56 |
| Geom Graph | 0.03 | 0.02 | 0.04 | 0.93 | 0.83 | 0.98 | 0.97 | 0.99 |
| Graph-Natural | 0.01 | 0.00 | 0.00 | 0.86 | 0.43 | 0.26 | 0.39 | 0.69 |

*Table 29.* Individual scores with code generation by Gemini-2.0-Flash.

| **Structure** | None | CodeMaybe | CodeEnforce | CodeOnly | Accuracy | Frequency |
|---|---|---|---|---|---|---|
| Array | 1.00 | 1.00 | 1.00 | 0.01 | 1.00 | 0.01 |
| Queue | 0.87 | 0.86 | 0.87 | 0.13 | 0.93 | 0.14 |
| Hashmap | 0.28 | 0.49 | 0.49 | 0.62 | 0.93 | 0.67 |
| Heap | 0.23 | 0.28 | 0.34 | 0.83 | 1.00 | 0.83 |
| DAWG | 0.18 | 0.20 | 0.17 | 0.00 | 0.00 | 0.01 |
| Geom Graph | 0.07 | 0.03 | 0.03 | 0.68 | 0.76 | 0.90 |
| Graph-Natural | 0.00 | 0.00 | 0.00 | 0.82 | 0.82 | 1.00 |

**Models cannot reason over the code they write.** Performance in *CodeMaybe* and *CodeEnforce* is on par with our original setup, indicating that writing code does not improve reasoning when models must internally simulate it. This reinforces our central claim: LLMs still struggle with structural reasoning, even when guided by their own code. To get a better understanding of the failure modes, we conduct additional qualitative analysis, and found a few major sources of failures.

- **Task understanding failure.** Generated code is often plausible and executable, but fails to understand core algorithmic rules, especially in uncommon tasks. For example, in DAWG, errors arise from incorrect minimization logic that uses object identity (id(child)) instead of structural signatures, violating equivalence and preventing subtree merging.

- **Brittle mapping in narrative language.** In GRAPH-NATURAL, models rely on brittle pattern matching (e.g., mapping "A space tunnel links planet1 and planet2" to "G.add_edge(planet1, planet2)"), but often fail to cover all phrasing variations or misinterpret descriptions (e.g., missing "Couriers frequently travel the tunnel that connects planet1 to planet2"). This highlights a key limitation that models still struggle to apply structural reasoning to ambiguous natural language scenarios, even with external tools.

- **Failure to internally reason over code.** When code is written correctly, but models are required to reason over it internally, they still make reasoning mistakes, e.g., DFS tie-breaking/backtracking mistakes.

**Code helps when tasks align with memorized patterns.** In *CodeOnly*, models perform well on GEOM GRAPH (k-dimensional graphs embedded in geometric space, which is a standard data structure widely used in computer graphics), whose code implementation is more available online. However, they struggle with the less familiar DAWG (directed acyclic word graph), where we define it with customized constraints to enforce a unique output. This suggests that performance may reflect memorization rather than true reasoning.

**Ablation with ChatGPT web UI.** Our benchmark was evaluated via APIs. In contrast, the ChatGPT web UI has a built-in code interpreter that executes code automatically when needed. We tested o4-mini on ChatGPT with DAWG+*CodeMaybe*. The model scored 0.70, generating and executing code in 20 out of 30 cases. We observed that the model typically uses code for complex algorithmic components, while relying on natural language reasoning for the rest.

# I. Auxiliary metric for `DSR-Bench`

In this section, we summarize a few auxiliary metrics we implemented in `DSR-Bench` that compliments the 0-1 scoring in the main paper.

*Table 30.* Individual scores with code generation by Gemini-2.5-Pro. Frequency measures how often the model outputs code as intended in CodeOnly, and accuracy measures the accuracy when code is generated.

| Structure | None | CodeMaybe | CodeEnforce | CodeOnly | Accuracy | Frequency |
|---|---|---|---|---|---|---|
| Array | 1.00 | 1.00 | 1.00 | 0.83 | 1.00 | 0.83 |
| Queue | 1.00 | 1.00 | 1.00 | 0.99 | 1.00 | 0.99 |
| Hashmap | 0.58 | 0.17 | 0.13 | 0.57 | 0.95 | 0.60 |
| Heap | 0.36 | 0.37 | 0.50 | 0.37 | 0.42 | 0.87 |
| DAWG | 0.61 | 0.47 | 0.45 | 0.33 | 0.90 | 0.37 |
| Geom Graph | 0.19 | 0.77 | 0.53 | 0.64 | 0.98 | 0.65 |
| Graph-Natural | 0.12 | 0.10 | 0.09 | 0.24 | 0.56 | 0.43 |

*Table 31.* Individual scores with code generation by Claude-3.5-Sonnet and Claude-3.7-Sonnet.

| | Claude-3.5-Sonnet | | | | Claude-3.7-Sonnet | | | |
|---|---|---|---|---|---|---|---|---|
| **Structure** | None | CodeMaybe | CodeEnforce | CodeOnly | None | CodeMaybe | CodeEnforce | CodeOnly |
| Array | 1.00 | 1.00 | 1.00 | 1.00 | 1.00 | 1.00 | 1.00 | 1.00 |
| Queue | 0.87 | 0.86 | 0.84 | 1.00 | 1.00 | 1.00 | 1.00 | 1.00 |
| Hashmap | 0.37 | 0.27 | 0.27 | 1.00 | 0.71 | 0.73 | 0.65 | 1.00 |
| Heap | 0.53 | 0.63 | 0.63 | 0.93 | 0.89 | 0.85 | 0.82 | 0.98 |
| DAWG | 0.20 | 0.17 | 0.13 | 0.20 | 0.17 | 0.18 | 0.15 | 0.89 |
| Geom Graph | 0.10 | 0.09 | 0.10 | 0.98 | 0.04 | 0.23 | 0.20 | 0.96 |
| Graph-Natural | 0.00 | 0.00 | 0.00 | 0.10 | 0.01 | 0.02 | 0.01 | 0.23 |

## I.1. Levenshtein distance

In addition to the binary (0/1) accuracy reported in the main text, `DSR-Bench` includes an optional evaluation metric based on *Levenshtein distance*, which measures the minimum number of single-character insertions, deletions, or substitutions needed to transform one string into another. As a continuous metric, it captures degrees of error that binary accuracy flattens. For instance, given the correct output `[1,3,6]`, the prediction `[3,1,6]` is clearly closer than `[0,0,0]`, and Levenshtein distance reflects that nuance.

However, this granularity can also blur important semantic distinctions. A syntactically well-formed but semantically incorrect output may still receive a high score, especially when the expected answer is long or formatted. When averaged over 30 test cases, models with large gaps in binary accuracy can appear deceptively similar under Levenshtein. For example, the SKIP LIST and DSU compound tasks yield the same Levenshtein score (0.75), despite the former achieving more than triple the binary accuracy (Table 32).

Output length further complicates cross-task comparison. Tasks with short, single-token outputs (e.g., BINARY SEARCH TREE depth) tend to show similar binary and Levenshtein scores (e.g., 0.66), while longer, multi-token outputs (e.g., GRAPH BFS) inflate Levenshtein scores (0.82) even when binary accuracy remains low (0.31). Relying on Levenshtein distance alone may thus give a misleading impression—for example, that the model performs well on BFS but poorly on tree depth—when binary accuracy indicates the opposite.

*Table 32.* Mean (± std) binary accuracy vs. Levenshtein distance scores across tasks using GPT-4.1.

| Data structure | Operation | Binary | Levenshtein |
|---|---|---|---|
| Array | Access | 1.00 (0.00) | 1.00 (0.00) |
| Queue | Compound | 0.82 (0.04) | 0.96 (0.01) |
| Stack | Compound | 0.97 (0.00) | 0.99 (0.00) |
| LRU Cache | Cache | 0.94 (0.02) | 1.00 (0.00) |
| Priority Queue | Compound | 0.63 (0.03) | 0.89 (0.01) |
| Hashmap | Compound | 0.19 (0.07) | 0.75 (0.02) |
| Trie | Compound | 0.39 (0.07) | 0.90 (0.02) |
| Suffix Tree | Construct | 0.00 (0.00) | 0.49 (0.01) |
| Skip List | Compound | 0.21 (0.02) | 0.75 (0.01) |
| BST | Insert | 0.79 (0.04) | 0.97 (0.01) |
| | Remove | 0.78 (0.04) | 0.95 (0.01) |
| | Post-Order Traversal | 0.82 (0.02) | 0.94 (0.01) |
| | Depth | 0.66 (0.05) | 0.66 (0.05) |
| | Compound | 0.69 (0.02) | 0.88 (0.01) |
| Heap | Compound | 0.58 (0.02) | 0.87 (0.01) |
| | Heapify | 0.57 (0.03) | 0.94 (0.01) |
| RB Tree | Construct | 0.12 (0.02) | 0.87 (0.00) |
| | Compound | 0.31 (0.04) | 0.88 (0.02) |
| B+ Tree | Compound | 0.27 (0.00) | 0.79 (0.00) |
| Graph | Breadth-First Traversal | 0.31 (0.05) | 0.82 (0.01) |
| | Depth-First Traversal | 0.50 (0.03) | 0.85 (0.01) |
| DSU | Compound | 0.06 (0.02) | 0.75 (0.01) |
| Bloom Filter | Compound | 0.10 (0.00) | 0.84 (0.01) |
| DAWG | Compound | 0.16 (0.02) | 0.74 (0.02) |

For these reasons, we report all results using binary accuracy and relegate Levenshtein evaluation to the toolkit. The implementation remains publicly available, as the metric can still offer a useful secondary perspective, particularly when comparing models on the same task and output length.

## I.2. Tree-edit Distance and Graph-edit Distance

For tree-based and graph-based data structures, `DSR-Bench` includes edits distances (ED) as an optional evaluation metric, which measures the minimum node operations required to transfer a tree or a graph to another tree or graph. We report normalized edit distance for trees. More formally, we report

$$\texttt{score} = 1 - \frac{\texttt{TED}(T_1, T_2)}{|T_1| + |T_2|},$$

where $|T|$ is the number of node in a tree, and $\texttt{TED}(T_1, T_2)$ is the tree edit distance (TED) between two trees $T_1$ and $T_2$. For graphs, exact graph edit distance is generally intractable in practice due to the hardness of graph isomorphism-related matching; however, in our setting, the node set is fixed, so we can directly compare edge sets. We therefore use the normalized edge disagreement, i.e.

$$\texttt{score} = 1 - \frac{|E_1 \cup E_2| - |E_1 \cap E_2|}{|E_1| + |E_2|}.$$

We conduct numerical study on a subset of `DSR-Bench` tasks. Results are shown in Table 33. Some key observations include:

- ED adds complementary information to 0-1 score: Models often recover substantial portions of the target structure even when the exact match is zero.

- ED distribution reveals the consistency of model outputs. For example, models produce more consistent outputs in KD-Tree, with no large errors (0% with ED > 0.5), even though the mean matches RB-tree. RB-Tree has less consistent outputs, with more large errors (13.3%) and greater variability (std 0.19 vs. 0.13). See also Figure 10.

*Table 33.* Summary statistics of edit distance (ED) scores. ED score are calculated using 1 minus normalized edit distance, therefore higher (↑) means better performance. % ED > 0.5 measures the percentage of tasks that is normalized edit distance 0.5 away from the ground truth, so lower (↓) means better performance. IQR stands for inter-quartile range, and STD stands for standard deviation.

| Task | 0–1 score ↑ (avg over 3 reps) | ED score ↑ (avg over 3 reps) | % ED > 0.5 ↓ | ED IQR (over 30 tasks) | ED STD (over 30 tasks) |
|---|---|---|---|---|---|
| RB-tree | 0.14 (0.03) | 0.66 (0.01) | 13.3 | [0.20, 0.44] | 0.19 |
| KD-tree | 0.04 (0.02) | 0.66 (0.01) | 0.0 | [0.28, 0.41] | 0.13 |
| Geom Graph | 0.00 (0.00) | 0.67 (0.00) | 10.0 | [0.25, 0.4] | 0.13 |

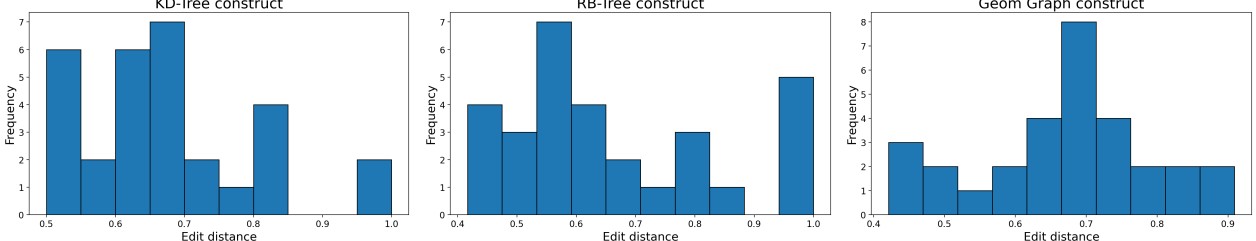

*Figure 10.* Distribution partial scores measured using 1- normalized edit distance for KD-Tree, RB-Tree, and Geom Graph. Distribution varies despite similar mean scores.

## J. Failure rates of JSON parsing via Structured Output

*Table 34.* Failure rates of JSON parsing across models and data structures.

| Model | Array | Priority Queue | Hashmap | RB Tree | Geom Graph | Bloom Filter |
|---|---|---|---|---|---|---|
| Llama3.3 | 0/30 | 0/30 | 0/30 | 0/30 | 0/30 | 0/30 |
| GPT-4.1 | 0/30 | 0/30 | 0/30 | 0/30 | 0/30 | 0/30 |
| DeepSeek-Chat | 0/30 | 0/30 | 0/30 | 0/30 | 0/30 | 0/30 |
| DeepSeek-R1 | 0/30 | 0/30 | 0/30 | 0/30 | 0/30 | 0/30 |
| o4-mini | 0/30 | 0/30 | 0/30 | 0/30 | 0/30 | 0/30 |
| Claude-3.5-Sonnet | 0/30 | 0/30 | 0/30 | 0/30 | 0/30 | 0/30 |
| Claude-3.7-Sonnet | 0/30 | 0/30 | 0/30 | 0/30 | 0/30 | 0/30 |
| Gemini-2.0-Flash | 0/30 | 0/30 | 0/30 | 0/30 | 0/30 | 0/30 |
| Gemini-2.5-Pro | 0/30 | 0/30 | 0/30 | 0/30 | 0/30 | 0/30 |

## K. Ablation on paraphrased prompt templates

In the `realistic` probe, we use 5 paraphrased templates for problem statements and 30 for operations (e.g., "A space tunnel connects planet1 and planet2") to smooth out ambiguity in natural language descriptions. However, in the `main` and `challenge` suites, we use formalized prompt templates to enable large-scale evaluation. We conducted an ablation to test how prompt template variations affect performance. Prompting GPT-4o with "Paraphrase the following description," we generated five paraphrases and found that overall performance trends remained consistent.

## L. Preliminary study on strategy adaptation with `DSR-Bench`

In Section 5, we note a future direction is to study whether models can select appropriate data structures given task requirements and switch dynamically. `DSR-Bench` already implicitly evaluates this adaptivity by not predefining strategies in prompts, requiring models to adjust their reasoning to the task. This emerges in (i) contrasting but related structure

*Table 35.* Performance on default and paraphrased prompt templates.

| Task | o4-mini | | GPT-4.1 | |
|---|---|---|---|---|
| | Default | Paraphrased | Default | Paraphrased |
| Priority Queue | 0.99 | 0.84 | 0.39 | 0.36 |
| Trie Tree | 0.89 | 0.90 | 0.63 | 0.61 |
| Geom Graph | 0.37 | 0.37 | 0.20 | 0.14 |

pairs (e.g., BFS/DFS, queue/priority queue), (ii) length generalization that demands adaptive complexity management, (iii) distribution shifts (Section 4.2) requiring robustness across input patterns, and (iv) natural language scenarios that demand transferring strategies across contexts (Section 4.3).

Beyond implicit evaluation, `DSR-Bench`'s modular design (data generation, prompt templates, automated evaluation, and schema-based verification) readily supports dynamic strategy switching. A preliminary study is shown in Table 37 and Table 36. We mask structure names or add constraints to `DSR-Bench`'s prompts, and test whether models can, (i) select BFS vs DFS for graph tasks, or (ii) choose between K-D tree and array for nearest-neighbor queries under varying complexities.

This extension underscores `DSR-Bench`'s broader value: it serves as a fundamental, extensible framework for probing not only structural reasoning but also more advanced reasoning abilities.

*Table 36.* Dynamic strategy switching on graph algorithms: selecting the most appropriate traversal method. Results show number of correct selections out of 10.

| Task | GPT-4.1 | o4-mini |
|---|---|---|
| Shortest Path | 9.5/10 (BFS) | 10/10 (BFS) |
| Cycle Detection | 10/10 (DFS) | 10/10 (DFS) |
| Connectivity Detection | 10/10 (Both equal; 8 BFS, 2 DFS) | 10/10 (Both equal; 6 BFS, 4 DFS) |

*Table 37.* Dynamic strategy switching on nearest-neighbor queries: selecting the most appropriate data structure under different sizes and query numbers.

| Task | GPT-4.1 | o4-mini |
|---|---|---|
| 1 query, n=20–30 | 9/10 (array) | 10/10 (array) |
| 1 query, n=40–60 | 6/10 (array) | 10/10 (array) |
| n queries, n=5–10 | 10/10 (array) | 10/10 (array) |
| n queries, n=20–30 | 10/10 (kd-tree) | 10/10 (kd-tree) |

# M. Performance correlation with real-world benchmarks

We conduct additional analysis to provide more evidence of the real-world practicality of DSR-Bench through its performance correlation with existing benchmarks for real-world applications. We collect scores from public benchmark leaderboards and consider the models that overlap with those in our evaluation. The number of overlapping models for each benchmark is recorded in the "# Models" column within the tables.

We cover a broad range of application-oriented benchmarks, including LiveBench (White et al., 2025) (reasoning, math, data analysis, etc.), SWE-Bench (Jimenez et al., 2024) and LiveCodeBench (Jain et al., 2025) (software engineering and coding), GSO Bench (Shetty et al., 2026) (SWE agent), AgentBench (Liu et al., 2024b) (agentic tasks such as web shopping and household tasks), and TableBench (Wu et al., 2025) (real-world table question answering). We use the following metrics

- Spearman's rank correlation coefficient: measures how well two benchmarks preserve the relative ranking of models.

- Pearson correlation coefficient: measures how strongly the scores on two benchmarks move together in a roughly linear way across models.

- Pairwise ranking accuracy: measures the fraction of model pairs for which the two benchmarks agree on which model performs better.

**Overall performance with real-world benchmarks**   As shown in Table 38, DSR-Bench shows strong positive alignment with several existing benchmarks for real-world applications, especially LiveBench, TableBench, SWE-Bench, and Live-CodeBench. This provides further evidence of DSR-Bench's practicality in the real world. The weaker correlation with more agentic benchmarks (GSO Bench, AgentBench) is also expected and informative: these tasks depend more heavily on tool use and API calls, rather than reasoning.

*Table 38.* Performance correlation between DSR-Bench and real-world benchmarks.

| Dataset | Spearman | Pearson | Pairwise | #Models |
|---|---|---|---|---|
| LiveBench | 0.90 | 0.94 | 0.89 | 10 |
| TableBench | 0.90 | 0.99 | 0.93 | 6 |
| SWE-Bench | 0.90 | 0.83 | 0.93 | 6 |
| LiveCodeBench | 0.97 | 0.99 | 1.00 | 5 |
| GSO Bench | 0.63 | 0.60 | 0.80 | 4 |
| AgentBench | 0.30 | 0.48 | 0.60 | 5 |

**Fine-grained performance with TableBench**   Beyond overall benchmark correlation, we also performed a finer-grained analysis on TableBench (Wu et al., 2025) in Table 39. We chose TableBench due to its focused domain on real-world table question answering, making it suitable to interpret the types of reasoning required. We found its strongest alignment with DSR-Bench's temporal, linear, hierarchical, and network categories. In our interpretation, this is consistent with the main challenges stated in TableBench's paper, namely multi-step reasoning (hierarchical), trend forecasting (temporal, linear), and chart generation (graph/network).

*Table 39.* Performance correlation between TableBench and DSR-Bench relationship categories.

| Relationship category | Spearman | Pearson | Pairwise |
|---|---|---|---|
| Temporal | 1.00 | 0.98 | 1.00 |
| Linear | 0.94 | 0.88 | 1.00 |
| Hierarchical | 0.94 | 0.95 | 0.93 |
| Network | 0.94 | 1.00 | 0.93 |
| Associative | 0.83 | 0.95 | 0.87 |
| Hybrid | 0.66 | 0.93 | 0.80 |

At the data-structure level, trie, heap, and graph show stronger correlation, while bloom filter, KD heap, and skip list show weaker correlation in Table 40. In our interpretation, this is consistent with the demands of TableBench, which may involve entity matching, ranking/comparison, and multi-hop reasoning rather than specialized operations such as probabilistic membership testing or high-dimensional indexing.

*Table 40.* Performance correlation between TableBench and DSR-Bench data structure tasks.

| Data structure task | Spearman | Pearson | Pairwise |
|---|---|---|---|
| Trie | 0.94 | 0.89 | 0.93 |
| Heap | 0.94 | 0.93 | 0.93 |
| Graph | 0.94 | 0.94 | 0.93 |
| . . . | | | |
| Bloom Filter | 0.60 | 0.91 | 0.73 |
| KD Heap | 0.58 | 0.96 | 0.75 |
| Skip List | 0.49 | 0.92 | 0.67 |

We note that real-world benchmarks are high-level and domain-specific, and often combine structural reasoning with other abilities (e.g., knowledge retrieval, instruction-following, language understanding). Our goal in proposing DSR-Bench is to isolate and evaluate structural reasoning itself, a fundamental requirement for real-world problem-solving.

