# OpenReview forum: "Can LLMs Reason Structurally? Benchmarking via the lens of Data Structures"
_ICML.cc/2026/Conference — ICML 2026 regular_

### Official Review · Reviewer_F1tJ · 2026-02-24

**Soundness:** 2
**Presentation:** 3
**Significance:** 2
**Originality:** 2
**Overall Recommendation:** 4
**Confidence:** 3

**Summary:**

This paper introduces DSR-Bench to systematically evaluate the structural reasoning capabilities of large language models. The evaluation revealed that current LLMs struggle with fundamental aspects of structural reasoning, particularly multi-attribute and multi-hop relationships, and show significant performance drops when tasks are presented in natural language or involve high-dimensional spatial data.

**Compliance With Llm Reviewing Policy:**

Affirmed.

**Final Justification:**

I appreciate the author's detailed rebuttal, which has addressed most of my concerns. Therefore, I raised the score to "4: weak accept". The only concern I have is that the structural data studied in this article may have rather ambiguous correlations in real scenarios. This aspect needs to be strengthened; otherwise, the application scope will be greatly reduced.

**Key Questions For Authors:**

1. Where does the rationality of the difficulty level setting come from? For example, is there any reference or experimental basis for the selection of the number of nodes? Meanwhile, this method of quantifying difficulty is overly simplistic. A more effective approach would be to provide other objective indicators for graph complexity.
2. Regarding model discrimination: As shown in Table 2, the more complex the relationship, the benchmark on that relationship is more capable of distinguishing SOTA models. I observed that the performance of the first 7 models was not significantly different in the linear case, but the differences became more obvious when moving to the hybrid relationship. So, how can performance be differentiated within the same relationship?
3. In Line 213, the author claims that these operations are for evaluating the multi-step structural reasoning of the model. But what's the difference between this and the instructions-following ability?
4. Are there any cases of reasoning on real-world data structures, which are also line, hybrid, graph, etc.? The case provided in Figure 1 is too formal.

**Limitations:**

The article does not explicitly state the limitations and broader impacts of the proposed DSR-Bench.

**Strengths And Weaknesses:**

**Strengths:**

1. Comprehensive evaluation scope: DSR-Bench contains 20 data structures, 35 operations, and 4,140 problem instances.
2. DSR-Bench features hierarchical task organization, fully automated generation and evaluation, and fine-grained diagnostics.
3. The observations are valuable, highlighting limitations of current LLMs.


**Weaknesses:**

1. The benchmarking based solely on data structures may deviate from the real-world user application scenarios.
2. Limited contribution: As stated in Section 2, there are many overlaps with the existing algorithmic benchmarks. It is difficult to determine whether the coverage is broader or more granular compared to the previous benchmark.

---

> ### Author Rebuttal · Authors · 2026-03-31
>
> We thank the reviewer for highlighting our evaluation as “comprehensive,” our observations as “valuable,” and for recognizing our three key strengths: hierarchical task organization, fully automated generation and evaluation, and fine-grained diagnostics. We address the concerns below.
> ### **[W1] Real-world practicality**:
> We clarify that real-world applicability is a core objective of our benchmark design, which we address across multiple dimensions:
> - **Canonical data structure tasks (the “main” set) underpin many real-world systems** (as motivated in l.82-91); we list each task’s applications in Table 5 (Appendix B). E.g., B+ trees are widely used in databases and file systems for efficient lookups and range queries.
> - **The “spatial” probe evaluates on high-dimensional data common in practice**. For example, K-D trees are used to accelerate nearest-neighbor search over 128-dimensional SIFT descriptors in computer vision (l.374–377).
> - **The “realistic” probe embeds these tasks in context-rich, real-world scenarios**. It assesses whether LLMs can handle narrative noise and still apply structural reasoning, reflecting their readiness for real-world deployment (l.357-367).
> ### **[W2] Overlap with existing algorithmic benchmarks**:
> We respectfully disagree with the concern and appreciate the opportunity to clarify. Task-level overlap is minimal, and our benchmark introduces novelty across multiple axes:
> - **Only 2 out of 35 tasks overlap (“main”)**: In Sec 2 (l.149–160), we detail the task-level distinctions: no overlap with NLGraph/GraphQA, and only 2 tasks (BFS, DFS) overlap with CLRS-Text, with substantially different prompt designs.
> - **Our specialized probes are novel (“spatial”, “realistic”, “code”)**. Existing algorithmic benchmarks primarily focus on canonical tasks (our “main” suite) rather than on settings with real-world utility goals.
> - **We address key limitations of existing algorithmic benchmarks**. Table 1 (l.139-145) highlights our differences. In particular, we address the key challenge (l. 55–69): design tasks that are *atomic* enough for precise error attribution while remaining *expressive* to cover a broad range of tasks.
> - **Our contribution is not limited to algorithmic benchmarks**. (l.18–54) We address limitations in existing *reasoning* and *coding* benchmarks by isolating algorithmic reasoning from domain-specific complexity.
>
> **[Q1]** Multi-axis difficulty design:
>
> We clarify that length is only one dimension in our multi-axis difficulty design:
> - **Length is a standard difficulty metric** in the literature [6,7,11], as long reasoning chains require tracking intermediate states and handling error propagation, making it a practical and interpretable metric.
> - **DSR-Bench includes 4 levels of difficulty measures**:
> |Difficulty measure|Set|Explanation|
> |---|---|---|
> |Length|main|Short/medium/long|
> |Relationship category|main,challenge|Linear/hybrid/etc with different structural complexity|
> |Dimensionality|spatial|Difficulty grows with data dimensionality|
> |Narrative noise|realistic|Difficulty from context-rich language|
>
> **[Q2]** Performance differentiation:
>
> We explain how performance can be differentiated within the same relationship:
> - **Scores for each relationship are averaged across all tasks in that category**. “Linear” only includes one simple task (array).
> - **Fine-grained differences emerge at the individual task level**. For example, in “temporal,” both GPT-5 and DeepSeek-R1 score perfectly on “queue,” but diverge on “LRU” (1.00 vs. 0.16), revealing R1’s weakness in inferred temporal dependencies.
>
> **[Q3]** Term usage clarification:
> - **Multi-step reasoning**: The ability to execute a sequence of operations over intermediate states [14]. For this, we design atomic operations (“What is the tree after insert 7?”) and compound operations (“What is the final tree after insert 4, insert 5, delete 3?”).
> - **Instruction-following**: The ability to produce outputs in a specified format [9]. We use Structured Output and observe no scheme violations (Table 30 in Appendix J).
>
> **[Q4]** Formal vs. real-world:
>
> We clarified the real-world utility of our benchmark in [W1]. Furthermore:
> - **Prompts in “realistic” probe are less formal**: Prompts in Figure 1 are from the “main” suite with canonical tasks in formal language. Prompts in “realistic” probes use narrative, context-rich language embedded in the real world.
> - **DSR-Bench is easily extensible to real-world data**: Our modular codebase supports easy integration with real-world data structures. To demonstrate this, we conducted additional experiments using graphs drawn from real-world distributions (see **Table 5** of the supplement).
>
> **Supplementary PDF** (Table 5, references used here): [link](https://anonymous.4open.science/api/repo/re-1330/file/re.pdf).
>
> We hope we have clarified the practical utility and contributions of our benchmark. Please let us know if any questions remain. Thank you for your time and consideration.

---

> > ### Author Rebuttal · Reviewer_F1tJ · 2026-04-01
> >
> > Thank you for the author's reply. My questions have been largely resolved. I still have the following questions:
> > 1. In "Length is a standard difficulty metric in the literature [6,7,11]", What do these references represent? They are not mentioned in the rebuttal, nor are there any ID identifiers in the original paper.
> > 2. Can you provide direct evidence about the real-world practicality with performance correlation between DSR-Bench and existing benchmarks for real-world applications? In Sec. 4.3, only the deliberately-constructed real-world scenarios were tested.
> > 3. Besides, during the rebuttal stage, it seems that external links should not be allowed to be added, as this would be considered unfair.

---

> > > ### Author Response · Authors · 2026-04-06
> > >
> > > We are glad that most questions have been resolved, and we appreciate the opportunity to clarify the remaining ones.
> > > 1. We apologize for the confusion. We have added the references at the end of this rebuttal.
> > > 2. Following the reviewer’s suggestion, we conducted additional analysis on DSR-Bench's performance correlation with existing benchmarks for real-world applications.
> > > - **Setup**: We collect scores from public benchmark leaderboards and consider the models that overlap with those in our evaluation. We report the number of overlapping models for each benchmark in “# Models”.
> > > - **Benchmarks**: We cover a broad range of benchmarks, including LiveBench (reasoning, math, data analysis, etc.), SWE-Bench and LiveCodeBench (software engineering and coding), GSO Bench (SWE agent), AgentBench (agentic tasks such as web shopping and household tasks), and TableBench (real-world table question answering).
> > > - **Metrics**:
> > >   - Spearman’s rank correlation coefficient: measures how well two benchmarks preserve the relative ranking of models
> > >   - Pearson correlation coefficient: measures how strongly the two sets of scores move together in a roughly linear way across models
> > >   - Pairwise ranking accuracy: measures the fraction of model pair rankings for which the two benchmarks agree on
> > > ### Overall performance with real-world benchmarks
> > > DSR-Bench shows **positive alignment with several benchmarks for real-world applications**, especially LiveBench, TableBench, SWE-Bench, and LiveCodeBench. This provides further evidence of DSR-Bench’s practicality in the real world. The weaker correlation with more agentic benchmarks (GSO Bench, AgentBench) is also expected and informative: these tasks depend more heavily on tool use and API calls, rather than reasoning.
> > > |Dataset|Spearman|Pearson|Pairwise|#Models|
> > > |---|---|---|---|---|
> > > |LiveBench|0.90|0.94|0.89|10|
> > > |TableBench|0.90|0.99|0.93|6|
> > > |SWE-Bench|0.90|0.83|0.93|6|
> > > |LiveCodeBench|0.97|0.99|1.00|5|
> > > |GSO Bench|0.63|0.60|0.80|4|
> > > |AgentBench|0.30|0.48|0.60|5|
> > > ### Fine-grained performance with TableBench
> > > Beyond overall benchmark correlation, we also performed a finer-grained analysis on TableBench. We chose TableBench due to its focused domain, which makes it suitable for interpreting the types of reasoning required. We found its strongest alignment with DSR-Bench’s temporal, linear, hierarchical, and network categories. In our interpretation, this is **consistent with the main challenges stated in TableBench’s paper**, namely multi-step reasoning (hierarchical), trend forecasting (temporal, linear), and chart generation (graph/network).
> > > |Relationship category|Spearman|Pearson|Pairwise|
> > > |---|---|---|---|
> > > |Temporal|1.00|0.98|1.00|
> > > |Linear|0.94|0.88|1.00|
> > > |Hierarchical|0.94|0.95|0.93|
> > > |Network|0.94|1.00|0.93|
> > > |Associative|0.83|0.95|0.87|
> > > |Hybrid|0.66|0.93|0.80|
> > >
> > > At the data-structure level, trie, heap, and graph show the strongest correlation, while bloom filter, KD heap, and skip list show the weakest correlation. In our interpretation, this is **consistent with the demands of TableBench**, which may involve entity matching, ranking/comparison, and multi-hop reasoning rather than specialized operations such as probabilistic membership testing or high-dimensional indexing.
> > > |Data structure task|Spearman|Pearson|Pairwise|
> > > |---|---|---|---|
> > > |Trie|0.94|0.89|0.93|
> > > |Heap|0.94|0.93|0.93|
> > > |Graph|0.94|0.94|0.93|
> > > |...||||
> > > |Bloom Filter|0.60|0.91|0.73|
> > > |KD Heap|0.58|0.96|0.75|
> > > |Skip List|0.49|0.92|0.67|
> > >
> > > We note that real-world benchmarks are high-level and domain-specific, and often combine structural reasoning with other abilities (e.g., world knowledge, commonsense reasoning). Our goal in proposing DSR-Bench is to evaluate structural reasoning itself more directly, a fundamental requirement for real-world problem-solving. The positive correlations above support this view.
> > >
> > > 3. We understand the reviewer’s concern. We gently note that anonymous links are permitted under the authors’ instructions. However, we understand that reviewers are not required to follow them. We apologize for any inconvenience this may have caused.
> > >
> > > We are grateful for the reviewer’s quick reply, which allowed us to conduct additional analysis. We are thankful for the reviewer’s suggestion on the correlation analysis, which we found informative and will add to the revision. Given this additional evidence, we hope the concern about real-world practicality can be substantially addressed. We sincerely thank the reviewer for their time and consideration.
> > >
> > > [6] Markeeva et al., The CLRS-text algorithmic reasoning language benchmark, 2024.
> > > [7] Modarressi et al., Nolima: Long-context evaluation beyond literal matching. ICML 2025.
> > > [9] Ouyang et al., Training language models to follow instructions with human feedback. NeurIPS 2022.
> > > [11] Wang et al., Can language models solve graph problems in natural language? NeurIPS 2023.
> > > [14] Wei et al., Chain-of-thought prompting elicits reasoning in large language models. NeurIPS 2022.

---

### Official Review · Reviewer_ferX · 2026-02-26

**Soundness:** 3
**Presentation:** 3
**Significance:** 3
**Originality:** 3
**Overall Recommendation:** 4
**Confidence:** 4

**Summary:**

This paper introduces DSR-Bench, the first benchmark for systematically evaluating the structural reasoning capabilities of LLMs through the lens of data structures. It comprises 20 data structures and 4,140 problem instances, and features four probing components (challenge, spatial, realistic, code) that enable comprehensive diagnosis across complex structures, high-dimensional spaces, context-rich scenarios, and code generation.

**Compliance With Llm Reviewing Policy:**

Affirmed.

**Final Justification:**

Thanks to the authors' responses, my concerns have been addressed. I will keep my score.

**Key Questions For Authors:**

1. It remains unclear whether the occurrence of zero scores stems from the models' inherent inability to handle certain datasets, or from issues such as poorly designed prompts or noise introduced during dataset construction that hinder model performance.

2. Hope authors provide case studies, rather than focusing solely on prompt design.

3. For the “Answer in N tokens” instruction in prompt, how often do models deviate from N? Would removing this sentence affect the model performance?

**Limitations:**

Lack discussion regarding the potential impact of inherent hallucinations and biases in large language models on structural reasoning.

**Strengths And Weaknesses:**

Strengths：

1. Using data structures as the organizing lens for “structural reasoning” is well-motivated and distinct from prior algorithmic and coding benchmarks, enabling atomic-yet-expressive probes and interpretable failures.

2. Broad model coverage (13 models across open/closed, instruction-tuned/reasoning) and multi-run evaluation with prompting ablations provide a robust empirical picture.

3. The categorization into Linear/Temporal/Associative/Hierarchical/Network/Hybrid clarifies what capability each task probes.

Weakness：

1. In Tables 2 and 3, certain models score 0 on Network, Hybrid, and K-D Heap tasks. Although the experimental section mentions that model performance degrades significantly on some datasets, it does not explicitly discuss or analyze the reasons behind these 0 scores, which can be regarded as anomalous performance.

2. Lack case studies, particularly regarding the outputs of LLMs that received a score of 0.

3. Lack discussion regarding the potential impact of inherent hallucinations and biases in LLMs on structural reasoning.

---

> ### Author Rebuttal · Authors · 2026-03-31
>
> We appreciate the reviewer’s positive assessment of our work, including recognizing our use of data structures as “well-motivated” and “distinct” from prior work, our evaluation as having “broad model coverage” that provides “a robust empirical picture,” and the usefulness of the relationship categories in our benchmark. We address the concerns below.
> ### **[W1] Error analysis of 0 scores**
> Thank you for the observation. We conduct additional analysis to investigate the causes of zero scores by programmatically extracting error statistics and manually inspecting model outputs, under two settings.
> 1. **Same task across different models:** KD-Tree (task with the most 0 scores) on Claude-3.7-Sonnet, DeepSeek-V3, GPT-4.1, and Qwen3-8B (see **Table 4** in supplement for error breakdown).
>     - **Common failure patterns**
>       - **Early errors are highly detrimental**: KD-Tree construction requires recursive partitioning in k-dimensional space, so a small early mistake can invalidate the entire tree under exact-match evaluation.
>       - **Incorrect median selection is common**: Models often select position n//2 ±1 instead of n//2, even when the input size is odd with no tie-breaking.
>       - **Difficulty following tie-breaking rules**: Models sometimes fail to select the correct middle element when the number of points is even, despite the prompt specifying the desired behavior.
>     - **Models exhibit qualitatively different behaviors despite similar near-zero scores**
>       - GPT tends to show a rightward bias when selecting the median, often overshooting by +2 or +3 positions, whereas the offsets of DeepSeek and Claude are more balanced.
>       - DeepSeek occasionally selects the min or max instead of the median, a behavior not observed in the other two models.
>       - Qwen fails in qualitatively different ways: it often produces unparsable outputs, hallucinated points, and sometimes returns undersized trees.
> 2. **Different tasks for the same model**: Trie (associative), KD-Heap (hierarchical), Geom Graph (network), DAWG (hybrid) with Qwen3-8B.
>     - **Failure to execute intended reasoning**: Across four tasks, Qwen appears to retrieve surface-level knowledge of the data structure in our prompts but fails to follow through with correct execution.
>     - **Frequent element dropping**: Qwen often drops elements from earlier states, indicating difficulty maintaining context across steps.
>     - **Reliance on pattern matching**: Qwen occasionally outputs in \box{} format (common in training data) despite not being required, and occasionally responds in Chinese despite no Chinese input.
> ### **[W2] Case studies**
> We provided an in-depth analysis of 0 scores in [W1]. We also highlight that our paper includes **additional case studies and qualitative analyses** in the Appendix, summarized below:
> - CoT error analysis: Appendix E
> - Spatial probe error analysis: Appendix F
> - Code probe error analysis: Appendix H
> - Prompt paraphrase ablation: Appendix K
> - Case study on dynamic algorithmic selection: Appendix L
> ### **[W3] Impact of LLM hallucination and bias**
> We appreciate the opportunity to clarify our designs to mitigate the impact of these two known limitations of LLMs:
> - **No reliance on external knowledge to reduce hallucination impact**: As described in Sec. 3.3, all necessary information is provided in the prompt, including descriptions of the data structures and operations. There is no requirement for factual knowledge retrieval.
> - **Intermediate hallucinations treated as reasoning errors**: If a model introduces hallucination during reasoning (e.g., referencing a node not in the input), it is counted as a reasoning failure.
> - **Unambiguous task specifications to mitigate model bias**: We provide precise definitions of tasks and operations (e.g., tie-breaking rules and hash functions) to eliminate ambiguity across different model interpretations.
> - **Evaluation based on final outputs**: We score only final answers, allowing flexibility in intermediate reasoning since models may adopt different valid strategies.
> - **Robustness to prompt variation**: In Appendix K, we evaluate five paraphrased prompt templates across three tasks on o4-mini and GPT-4.1. We observe consistent performance trends, indicating robustness to prompt variations.
>
> **[Q1]** Error analysis on 0 scores: See [W1]
>
> **[Q2]** Case studies: See [W2].
>
> **[Q3]** “Answer in N tokens”: The “Answer in N tokens” instruction follows “Six strategies for getting better results with prompt engineering” (OpenAI, 2025). We conducted an ablation by removing this instruction on GPT-5 and observed no significant change in performance (see **Table 3** of the supplement).
>
> **Supplementary PDF** (tables): [link](https://anonymous.4open.science/api/repo/re-1330/file/re.pdf).
>
> We thank the reviewer for the thoughtful feedback. We will add [W3] to limitation discussion and experiments [W1, Q3] to the revision. Please let us know if any questions remain. Thank you again.

---

> > ### Author Rebuttal · Reviewer_ferX · 2026-04-01
> >
> > Thanks to the authors' responses, my concerns have been addressed.

---

> > > ### Author Response · Authors · 2026-04-06
> > >
> > > We gratefully thank the reviewer for their constructive feedback on our paper. We appreciate the reviewer’s suggestion on zero-score case studies, which provided additional insights that we will incorporate into our revision. We will also add the discussion on hallucination and bias to the Limitations to strengthen our arguments. Thank you again for your support. We deeply appreciate the time and consideration you invested in our paper.

---

### Official Review · Reviewer_LaHT · 2026-03-09

**Soundness:** 3
**Presentation:** 4
**Significance:** 3
**Originality:** 3
**Overall Recommendation:** 5
**Confidence:** 4

**Summary:**

This paper discusses an important topic of how LLMs can reason about data structures. The authors propose a new benchmark DSR-Bench to evaluate if code is needed for LLMs to reason about algorithms, and how good LLMs are at reasoning about different data structures.

**Compliance With Llm Reviewing Policy:**

Affirmed.

**Final Justification:**

I am happy with the paper and rebuttal and will maintain my score of Accept.

**Key Questions For Authors:**

See weaknesses. Additionally:

1. How does this benchmark differ from LeetCode problems, which also vary in difficulty from easy to hard? Why did you not leverage LeetCode problems in the benchmark creation?
2. How exactly was the synthetic data generated, and how was its validity verified?

**Limitations:**

yes

**Strengths And Weaknesses:**

**Strengths**

1. The authors account for data contamination by generating synthetic data rather than relying on existing benchmarks.
2. Results are designed to be interpretable and not only reward/rank-driven. As the authors note, data structures have well-defined semantics and easily computable ground truth, which allows interpretable analysis of where/how models fail.
3. The evaluation is extensive and thorough, including different scenarios, ablations (with/no code).
4. Code is publicly available.

**Weaknesses**

1. Hierarchical task organisation may introduce unfair penalisation. If a model lacks knowledge of a specific data structure, e.g., a dequeue, it will fail all tasks (easy, medium, and hard).
2. Prompts contain hints that are too obvious to the model.
    a. The example on p. 4 includes an explanation of what a queue is (item (i)). Since LLMs are expected to possess this in parametric knowledge, providing such explanations can help the LLM and increase its performance. A user would not normally explain what a queue is. The prompt always includes a concise description of the data structure, which is counterintuitive for a benchmark designed to test reasoning about data structures. Furthermore, probe prompts leak information about the relevant data structures or algorithms.  In Appendix G, in the second case, the question is “What is the pre-order traversal of the appointment schedule following the binary search tree?” and in the third case task description is “The courier computes its route with depth-first search, and whenever multiple unvisited…”. Both contain a direct hint about the data structure. This allows models to identify the data structure/algorithm without any genuine reasoning.
3. The way data was synthetically generated is not discussed in detail (l. 213-216, c.1). It is unclear exactly how data is generated and how synthetic data is validated, such that it fits the problem/data structure.
4. Token budget restriction may lead to better/worse results for particular models. It would be informative to report performance without token budget restrictions as an ablation.
5. Binary scoring penalises partially correct responses where an LLM completes all but one step of a task. Evaluating intermediate steps and partial results, especially for probe scenarios and multi-hop reasoning, would allow a more informative assessment.
6. The probe scenarios in the paper appear synthetic and lack scalability. Existing HackerRank/LeetCode problems could be used since they contain a good collection of tasks that allow evaluating data structure reasoning with different difficulties. Many LeetCode problems have easy/medium/hard versions with additional constraints. Leveraging LeetCode problems would enable a scalable and realistic way to create probe scenarios.

**Minor**
1. "natural-language benchmark" should be written without a hyphen

2. Several references take multiple pages, e.g., the citation for Gemini 2.5. Please fix this with "et al.".

---

> ### Author Rebuttal · Authors · 2026-03-31
>
> We thank the reviewer for the careful reading and thoughtful feedback, and for highlighting our benchmark’s strengths in addressing “data contamination,” “interpretable analysis,” and “extensive and thorough” evaluation. We address the concerns below.
> ### **[W1] Fairness**
> We clarify the meaning of “hierarchical organization” and how fairness is addressed in our design:
> - **Distinguish the two terms**
>   - Hierarchical organization: Tasks follow a natural hierarchy, where simpler tasks (e.g., queues) serve as prerequisites for more complex ones (e.g., BFS on graphs), enabling fine-grained failure localization.
>   - Difficulty levels (short/medium/long): Problem lengths are varied within the same task (e.g., graphs with 5 vs. 30 nodes), probing length generalization.
> - **Scores provide an averaged quantitative view**. While dependencies exist across both hierarchy and length, the atomicity of our tasks enables precise error attribution. We can further perform granular, qualitative analysis comparing reasoning traces with ground truth.
> - **Multiple design choices to promote fairness across models**
>   - We provide detailed task descriptions in prompts to prevent prior-knowledge biases (see [W2]).
>   - Scores are computed solely from final outputs, allowing different models to adopt any valid strategy.
> ### **[W2] Explicit task description**
> We address this concern in two parts:
> - *Why detailed task specification is necessary*
>   - **Mitigate model bias**: All necessary information is provided in the prompt to reduce bias arising from differences in prior knowledge across models (e.g., different distance metrics for geometric graphs), so that evaluation focuses on structural reasoning rather than knowledge retrieval.
>   - **Eliminate ambiguity to ensure a single ground truth**: Explicit definitions of tie-breaking rules, hash functions, etc., are required to ensure a unique ground truth, enabling automated evaluation without subjective human or LLM judging (Sec. 3.4).
> - *Why we explicitly name data structures or algorithms*
>   - **Alignment with our research focus**: whether LLMs can perform the required structural reasoning for specified tasks. Explicit naming does not remove the challenge, since models must still execute the correct algorithmic behavior (e.g., track states during tree traversal).
>   - **Data structure/algorithm identification is a distinct research question**, though valuable, lies beyond the main scope of this work. That said, our benchmark is extensible to evaluate such settings: Appendix L includes a preliminary study on such an identification challenge.
> ### **[W3] Data generation and evaluation procedures**
> Details of data generation, evaluation pipeline, and scoring system can be found in Sec. 3.4 (l. 230-260, c.1). We are happy to clarify further if questions remain.
> ### **[W4] Token budget restriction**
> Our token budgets balance fairness and practicality. Some instruction-tuned models (e.g., Gemini-2.0-Flash) support ~8k output tokens, so we use a uniform 8k limit. For reasoning models, based on pilot experiments and cost, we set limits of 15k/30k/45k for short/medium/long problems.
> ### **[W5] Intermediate step and partial scores**
> We do not quantitatively score intermediate steps to allow free thinking (see [W1]). Our pilot studies indicate that requiring parsable intermediate states tends to harm performance.
>
> For final outputs, we complement the binary metric with partial-credit metrics:
> - **Levenshtein string edit distance** (included Appendix I).
> - **Tree and graph edit distances** for suitable tasks: **Additional experiments** in our response to Reviewer zRHx [Q3].
> ### **[W6] Differences from LeetCode problems**
> Thanks for the suggestion. We view LeetCode tasks as a valuable future direction requiring careful design, but beyond the scope of this paper for several reasons:
> - **Contamination risk**. LeetCode problems are widely available in pretraining data, making it difficult to isolate genuine reasoning from memorized patterns.
> - **Different notion of difficulty**. LeetCode’s easy/medium/hard labels reflect the difficulty of human coding. Our benchmark has multiple difficulty axes: length, structural complexity, dimensionality, and narrative noise.
> - **Different evaluation goal**. LeetCode tasks assess more advanced skills (e.g., problem decomposition, algorithm selection, complexity analysis) beyond the fundamental abilities we target, but are valuable for future research.
> ### **[W7, W8] Typos**:
> We sincerely thank the reviewer for pointing out the typos and reference formatting issues! We will correct these in the revision.
>
> **[Q1]** See [W6].
>
> **[Q2]** See [W3].
>
> **Supplementary PDF** (additional experiment results): [link](https://anonymous.4open.science/api/repo/re-1330/file/re.pdf).
>
> We thank the reviewer for the constructive feedback. We will include these clarifications [W1, W2, W5, W6] in the revised manuscript. Please let us know if any questions remain. Thank you again.

---

> > ### Author Rebuttal · Reviewer_LaHT · 2026-04-01
> >
> > Thanks, my concerns were addressed, and I will keep the score.

---

> > > ### Author Response · Authors · 2026-04-06
> > >
> > > We sincerely thank the reviewer for their careful reading and detailed feedback, which has been helpful in strengthening our paper. We are grateful for their positive support and will ensure our responses are incorporated in the revision. Thank you again for your engagement throughout the rebuttal period.

---

### Official Review · Reviewer_zRHx · 2026-03-11

**Soundness:** 2
**Presentation:** 3
**Significance:** 3
**Originality:** 3
**Overall Recommendation:** 3
**Confidence:** 4

**Summary:**

This paper proposes DSR-Bench. This dataset consists of 4,140 problem instances spanning across 20 data structures, and 35 operations. This dataset is useful in terms of evaluating reasoning capabilities of LLMs on algorithmic tasks. This paper presents a systematic pipeline to generate data and evaluate LLMs without relying on autoraters. The authors evaluate 13 LLMs and show that they are still underperforming on the DSR-Bench challenge version. This indicates that these models are still not able to reason structurally. Based on experiments, there are several important and interesting findings presented in this paper.

**Compliance With Llm Reviewing Policy:**

Affirmed.

**Key Questions For Authors:**

- Many of the simple data structure questions can be solved using what LLMs has memorized about them. How do you ensure that success/failures are due to not being able to reason structurally?
- Do the authors have any study or validation on the quality of generated narratives for “realistic” sets of instances from DSR-Bench?
- Have the authors considered using structure-aware partial credit metrics, such as tree-edit distance or graph edit distance, instead of string-based Levenshtein distance?
- Could the authors elaborate on why code generation with external execution fails on non-standard structures like DAWG?

**Limitations:**

Yes

**Strengths And Weaknesses:**

Strengths:

- Having a benchmark to understand and evaluate structural reasoning capabilities of recent LLMs is valuable and useful. Authors create this benchmark very systematically with focus on evaluating some of the targeted reasoning capabilities.
- Creation of various versions of the benchmark is quite interesting. I specifically like the idea of focusing on atomic tasks to really understand LLMs’ reasoning behind solving them.
- Evaluation on various LLMs is systematic and comprehensive. There are various interesting findings from the study.
- Paper is well written and well organized.
- Selection of models to evaluate is diverse and inclusive of various styles of models which make conclusions robust.

Weaknesses:

- I agree with the importance of the data structures in understanding algorithmic reasoning capabilities of LLMs. However, I have a fundamental problem with claiming that synthetically generated data has minimal risk of contamination. The concepts of data structures are well studied in literature and hence they are part of vast pre-training data. I think it is very important that authors define what they mean by minimum contamination, and also provide some evidence for the same.
- Another major issue I have is limited coverage of prompting techniques for evaluation. I know that CoT is a de-facto method for reasoning, however, plan-and-solve, program of thoughts, and many more similar ones are also well-studied and known for improving structured reasoning. Benchmarking on such methods (at least on a subset of models) can give a better idea about the robustness of claims made in the paper.
- Authors claim that performance drops on more “realistic” tasks, but it might be due to noise that is coming with text narrative. For instance, a model might know the concept of Array, but not be able to apply or do it efficiently due to narrative bias. I think discussion on the reason behind the drop is really a structured reasoning or not is very important.
- There is qualitative analysis missing from some part of the paper where it might be important. For instance, authors state that “LLMs remain limited in their ability to perform structural reasoning, even when guided by their own code”, but having some qualitative study on instances and showing error analysis establishing why it fails might be more interesting.

Since this benchmark can be useful, I am willing to revise my assessment if my comments and questions are answered.

---

> ### Author Rebuttal · Authors · 2026-03-31
>
> We thank the reviewer for recognizing our benchmark as “valuable and useful” and appreciating our core motivation to evaluate structural reasoning through atomic tasks. We also appreciate the acknowledgment of our specialized probes, our evaluation as “systematic and comprehensive,” leading to “several important and interesting findings”, and that the paper is “well written and well organized.” We address the concerns below.
> ### **[W1] Minimum contamination**
> We point out two levels of “contamination”, following [15]:
> - **Test-set contamination**: This is the standard definition in prior work [3,4,5,8,10,15]: test prompts appearing in the training corpus. Our benchmark minimizes this risk by generating new instances from hand-crafted prompt templates and randomized generators.
> - **Task-level "contamination"**: This type of contamination appears closer to the concern raised in the review: evaluating models on task families that may already be represented in pretraining. We note that this issue is difficult to eliminate entirely at the scale of modern LLM training, but generally accepted. For example, many frontier models trained on competitive math data are still meaningfully evaluated on new problems from the same distribution [1]. Even so, we take steps to reduce the risk by introducing customized constraints (e.g., hash functions) that may deviate the test data from common distributions.
> ### **[W2] More prompting strategies**
> We appreciate the suggestion and conducted experiments with GPT-5 (see **Table 1** in supplement).
> - **Additional methods**: Plan-and-Solve, Self-Consistency, Code-Enforce, Least-to-Most.
> - **Our claims (Sec 4.1.1) remain robust**: (i) Lightweight prompts (Stepwise, 0-CoT, Plan-and-Solve, Self-Consistency) consistently improve performance. (ii) Crafted prompts (CoT, 3-Shot, Least-to-Most) can be most effective but require careful design.
> ### **[W3] Narrative noise**
> We clarify that narrative noise in the “realistic” probe is deliberate.
> - **Narrative ambiguity is the target of evaluation**. The goal is to test whether LLMs can navigate narrative noise (as in real-world scenarios) while still applying structural reasoning (as they do with formal descriptors). The performance drop reflects a gap in real-world deployment readiness.
> - **We conducted additional qualitative analysis**.
>   - **Natural language ambiguity**: Many errors stem from failures to map narrative to the correct operations; e.g., not recognizing “Another child was served and happily walked away” as implying a dequeue.
>   - **Tie-breaking difficulty**: Errors also arise from more complex tie-breaking rules expressed in narrative language, e.g., lexicographical order (“Kelvin” before “Krypton”) can be harder than simple numeric values (1 vs. 2).
>   - **Hallucination**: The model generates more hallucination errors, generating names or timestamps that are absent from the input.
> ### **[W4] Qualitative analysis on “code” probe**
> We summarize key error types, in addition to the qualitative analysis in Appendix H:
> - **Task understanding failure**: Generated code is often plausible and executable, but the model fails to understand core algorithmic rules, especially in uncommon tasks. For example, in DAWG, errors arise from incorrect minimization logic that uses object identity (id(child)) instead of structural signatures, violating equivalence.
> - **Brittle mapping in narrative language**: For “realistic” tasks, models rely on brittle pattern matching (e.g., mapping “A tunnel links A and B” to “G.add_edge(A, B)”) but fail to handle all paraphrases.
> - **Failure to internally reason over code**: When code is written correctly, but models are required to reason over it internally, they still make reasoning mistakes, e.g., DFS backtracking errors.
>
> **[Q1]** Memorization: We clarified “contamination” in [W1]. Our setup evaluates generalization by minimizing test-set contamination.
>
> **[Q2]** Validation for “realistic” probe: As noted in l.381–384, prompts are human-written and paraphrased by GPT-4o, then reviewed by three human verifiers for clarity. Each verifier also solves three random samples per task for solvability.
>
> **[Q3]** Structure-aware partial credit metrics: Thank you for the suggestion. We ran additional experiments using tree and graph edit distances across three tasks (see **Fig. 1 and Table 2** in supplement). These metrics provide complementary insights and highlight model consistency. We will add the metrics to our codebase.
>
> **[Q4]** Error analysis on “code” probe: We covered error analysis for DAWG in [W4].
>
> **Supplementary PDF** (tables, figures, references used here): [link](https://anonymous.4open.science/api/repo/re-1330/file/re.pdf).
>
> We thank the reviewer for their time and consideration, especially for acknowledging the usefulness of our benchmark for the community. We will add the additional discussion [W1] and experiments [W2, W3, W4, Q3] to the revision. Please let us know if any questions remain.

---

> > ### Author Rebuttal · Reviewer_zRHx · 2026-04-04
> >
> > Thank you for the rebuttal. I appreciate the substantial effort put into the new experiments (GPT-5 prompting baselines, structure-aware metrics) and qualitative analyses. Please include these additions in the updated version.
> >
> > However, my fundamental concerns regarding what the benchmark is actually measuring remain unresolved:
> >
> > W1: Authors acknowledge task-level contamination, and I agree that this issue is hard to eliminate. Because these specific data structures are heavily represented in pre-training data, it remains very difficult to prove the benchmark measures true reasoning rather than pattern matching of memorized concepts.
> >
> > W3: Your qualitative analysis confirms my suspicion: models are failing on the "realistic" probe largely due to natural language mapping failures and hallucinations, not necessarily a lack of structural reasoning. This conflicts with the main claim of the paper about evaluating algorithmic reasoning.
> >
> > Because of these, I keep my current score.

---

> > > ### Author Response · Authors · 2026-04-06
> > >
> > > Thank you for the follow-up. We believe the remaining concerns primarily require **a clearer statement of the scope of our claims**, which we were not able to fully elaborate in the previous response due to space constraints. We appreciate the opportunity to clarify them below.
> > > ### [W1] Contamination
> > > We agree with the reviewer that, because knowledge about canonical data structures is likely present in pretraining, DSR-Bench cannot rule out task-level exposure; indeed, this is beyond the scope of the present work. We’ll revise l.57 of Section 1 to explicitly state that **DSR-Bench is designed to reduce test-set contamination** (in line with existing literature and benchmarks, e.g., [1]). We will add an explicit “Limitations” section to clarify the distinction between task-level and test-set contamination, and that we minimize the latter but not the former. We’ll discuss possible extensions of DSR-Bench that explicitly rule out task-level exposure as a promising future direction (e.g., by devising entirely novel families of structural tasks rather than canonical textbook ones).
> > >
> > > We’ll provide **clearer distinctions among “memorization”, “generalization”, “reasoning”, and “pattern matching”** in Section 5 (Discussion and conclusion), using the extra page in the camera-ready. In this context, *memorization* corresponds to success driven by prior exposure to the same (or nearly identical) instances, which is precisely why minimizing test-set contamination is important. Meanwhile, *generalization* refers to the ability to apply learned computational operations to unseen instances. In this sense, DSR-Bench is best interpreted as evaluating *structural generalization* capabilities: whether models can correctly perform specific structural computations under controlled changes in lengths, constraints, distributions, and linguistic forms.
> > >
> > > As the reviewer suggests, we’ll clarify that **DSR-Bench doesn’t assume that success or failure can be cleanly partitioned into “true reasoning” or “pattern matching”**, as this distinction itself is not well-defined and remains actively debated in research. Our paper uses “*structural reasoning*” in an operational, behavioral sense: whether a model can correctly maintain intermediate states and follow task-specific rules. We’ll state this explicitly in l.80 of Section 1. Empirically, surface-level pattern matching alone does not seem sufficient for success on DSR-Bench: our results also show that models fail on very common data structures (e.g., heaps, priority queues) with likely high task-level exposure. Moreover, we view some failures due to potentially brittle pattern matching, as discussed in Sec. 4.2 on controlled changes in data distribution.
> > >
> > > Furthermore, we will clarify that **we did not intend to imply that DSR-Bench measures “reasoning” in the broadest sense**. Rather, it evaluates whether models can solve tasks by correctly performing explicit structural computations in an operational sense. We view this ability as a necessary but not sufficient prerequisite for *broader algorithmic reasoning*. We’ll sharpen the conclusions in Section 5 to position DSR-Bench as an initial step toward evaluating this capability.
> > >
> > > ### [W3] Narrative noise in the realistic probe
> > >
> > > We agree with the reviewer that the “realistic” probe is not a pure measure of structural reasoning in isolation. We’ll make this explicit by adding a clarification in the opening of Section 4.3 that this is **an auxiliary transfer probe** that tests whether a model’s ability to execute explicit structural computations transfers from formal prompts to context-rich language. The probe introduces an additional step not present in the formal setting: when structural reasoning problems are presented in natural language, models must first **(i) recover the relevant structure from context-rich language** before they can **(ii) apply the correct structural execution**. We will state the two-step requirement explicitly.
> > >
> > > As the reviewer suggests, we agree that the performance drop cannot be attributed solely to structural reasoning failure. In particular, we will add a sentence in l.397 to clarify that **errors in “realistic” mainly have two types**. One type is the same underlying reasoning failures already seen in canonical data structures (e.g., multi-hop, backtracking failures). The other type arises from the additional difficulty of recovering and maintaining the relevant structures from narrative language (e.g., language mapping, hallucination).
> > >
> > > Thank you for reading our response. We believe the remaining concerns are best resolved by stating our claims more precisely, and we appreciate the opportunity to clarify them here. We sincerely thank the reviewer for their engagement and the time and consideration they kindly offered us.
> > >
> > > [1] White et al., LiveBench: A Challenging, Contamination-Limited LLM Benchmark. ICLR 2025 Spotlight.

---

### Decision · Program_Chairs · 2026-04-30

**Decision:**

Accept (regular)

**Comment:**

This paper focuses an important topic of reasoning capability on data structures. The authors propose a new benchmark DSR-Bench to evaluate how good LLMs are at reasoning about different data structures. Experimental results show that existing models struggle with  structural reasoning, especially with multi-attribute and multi-hop relationships, and significant performance drops are observed when tasks are presented in natural language.

Reviewers acknowledge that this is an important problem and the DSR-Bench dataset is valuable to the community. Some issues raised by reviewers in the review comments have been well addressed in the response.

Based on these, this paper will be included in ICML'26. Please include these additions in the camera ready version.